# Learning Hierarchical Polynomials of Multiple Nonlinear Features

**Hengyu Fu** [*]  **Zihao Wang** [†]  **Eshaan Nichani** [‡]  **Jason D. Lee** [‡]

## Abstract

In deep learning theory, a critical question is to understand how neural networks learn hierarchical features. In this work, we study the learning of hierarchical polynomials of *multiple nonlinear features* using three-layer neural networks. We examine a broad class of functions of the form $f^\star = g^\star \circ \mathbf{p}$, where $\mathbf{p} : \mathbb{R}^d \to \mathbb{R}^r$ represents multiple quadratic features with $r \ll d$ and $g^\star : \mathbb{R}^r \to \mathbb{R}$ is a polynomial of degree $p$. This can be viewed as a nonlinear generalization of the multi-index model (Damian et al., 2022), and also an expansion upon previous work that focused only on a single nonlinear feature, i.e. $r = 1$ (Nichani et al., 2023; Wang et al., 2023).

Our primary contribution shows that a three-layer neural network trained via layerwise gradient descent suffices for

- complete recovery of the space spanned by the nonlinear features
- efficient learning of the target function $f^\star = g^\star \circ \mathbf{p}$ or transfer learning of $f = g \circ \mathbf{p}$ with a different link function

within $\widetilde{\mathcal{O}}(d^4)$ samples and polynomial time. For such hierarchical targets, our result substantially improves the sample complexity $\Theta(d^{2p})$ of the kernel methods, demonstrating the power of efficient feature learning. It is important to highlight that our results leverage novel techniques and thus manage to go beyond all prior settings such as single-index and multi-index models as well as models depending just on one nonlinear feature, contributing to a more comprehensive understanding of feature learning in deep learning.

## 1 Introduction

Deep neural networks have achieved remarkable empirical success across numerous domains of artificial intelligence (Krizhevsky et al., 2012; He et al., 2016). This success can be largely attributed to their ability to extract latent features from real-world data and decompose complex targets into hierarchical representations, which improves test accuracy (He et al., 2016) and allows efficient transfer learning (Devlin, 2018). These feature learning capabilities are widely regarded as a core strength of neural networks over non-adaptive approaches such as kernel methods (Wei et al., 2020; Bai and Lee, 2020).

Despite these empirical achievements, the feature learning capabilities of neural networks are less well understood from a theoretical point of view. Previous work on feature learning has shown that two-layer neural networks can learn *multiple* linear features of the input (Damian et al., 2022), that is, multi-index models. However, the two-layer architecture inherently limits the network's ability to represent and learn nonlinear features (Daniely, 2017). Given that many real-world scenarios involve diverse and nonlinear features, recent studies have shifted focus to investigating the learning of *nonlinear* features using deeper neural networks. Safran and Lee (2022); Ren et al. (2023); Nichani et al. (2023); Wang et al. (2023) have demonstrated that three-layer networks, when trained via gradient descent, can efficiently learn *hierarchical targets* of the form $h = g \circ p$, where $p$ represents certain types of features such as the norm $|\mathbf{x}|$ or a quadratic form $\mathbf{x}^\top \mathbf{A} \mathbf{x}$. However, these studies are limited to relatively simple hierarchical functions and mainly focus on targets of a single feature. It remains unclear whether neural networks can efficiently learn a wider range of

---

[*]Peking University. Email: `fhy2021@stu.pku.edu.cn`

[†]Stanford University. Email: `zihaow@stanford.edu`

[‡]Princeton University. Email: {`eshnich`,`jasonlee`}`@princeton.edu`

hierarchical functions, particularly those that depend on multiple nonlinear features. This leads us to the following central question:

*Can neural networks adaptively identify **multiple nonlinear features** from the hierarchical targets by gradient descent, thereby allowing an efficient learning for such targets?*

## 1.1 Main Contributions

In this paper, we provide strong theoretical evidence that three-layer neural networks have the ability to learn multiple hidden nonlinear features. Specifically, we study the problem of learning any hierarchical polynomial with multiple quadratic features using a three-layer network trained via layer-wise gradient descent. Our main contributions are summarized as follows:

- **A Novel Analytic Framework for Multi-Nonlinear Feature Learning.** We demonstrate that when the target function belongs to a broad class of the form $f^\star = g^\star \circ \mathbf{p}$, where $\mathbf{p} : \mathbb{R}^d \to \mathbb{R}^r$ represents $r$ quadratic (nonlinear) features and $g^\star$ is a link function, the first step of gradient descent efficiently learns and recovers the space spanned by these nonlinear features $\mathbf{p}$ within only $\widetilde{\mathcal{O}}(d^4)$ samples. We remark that our proof techniques are also applicable to general nonlinear features. The core technical novelty is that we develop a novel and general universality argument (Lemma 1) that bridges multi nonlinear feature models to multi-index models, which allows for an accurate reconstruction of the features through a simple linear transformation on the learned representations with small approximation error (Proposition 1)

- **Improved Sample Complexity and Efficient Transfer Learning.** Leveraging the learned features in the first GD step, we prove that when the link function $g^\star$ is a polynomial of degree $p$, the gradient descent on the outer layer can achieves a vanishing generalization error with a small outer width and at most $\mathcal{O}(r^{\mathcal{O}(p)})$ additional training samples, removing the dependence on $d$ (Theorem 1). This significantly improves upon the sample complexity of kernel methods, which require $\Theta(d^{2p})$ samples. Moreover, our analysis enables efficient transfer learning for any other target function of the form $f = g \circ \mathbf{p}$ with a different link function $g$, which also only requires $\mathcal{O}(r^{\mathcal{O}(p)})$ additional samples.

## 1.2 Related Works

**Kernel Methods.** Earlier research links the behavior of gradient descent (GD) on the entire network to its linear approximation near the initialization. In this scenario, neural networks act as kernels, known as the Neural Tangent Kernel (NTK). This connection bridges neural network analysis with established kernel theory and offers initial learning guarantees for neural networks (Jacot et al., 2018; Soltanolkotabi et al., 2018; Du et al., 2018; Chizat et al., 2019; Arora et al., 2019). However, kernel theory fails to explain the superior empirical achievements of neural networks over kernel methods (Arora et al., 2019; Lee et al., 2020; E et al., 2020). Networks in the kernel regime **fail to learn features** (Yang and Hu, 2021), not adaptable to hierarchical structures of real world targets. Ghorbani et al. (2021) proves that for uniformly distributed data on the sphere, the NTK method requires $\widetilde{\Omega}(d^k)$ samples to learn any polynomials of degree $k$ in $d$ dimensions, which is impractical when $k$ is large. Thus, a central question is how neural networks can detect and capture the underlying hierarchies in the target functions, which allows for a better generalization behavior versus kernel methods.

**Learning Linear Features.** Recent studies have demonstrated neural networks' capability to learn hierarchical functions of linear features more efficiently than kernel methods. Specifically, Bietti et al. (2022); Ba et al. (2022) establish the efficient learning of single-index models, i.e., $f^\star(\mathbf{x}) = g(\langle \mathbf{u}, \mathbf{x} \rangle)$. Furthermore, recent works Damian et al. (2022); Abbe et al. (2023); Dandi et al. (2023a); Bietti et al. (2023) further demonstrate that for isotropic data, two-layer or three-layer neural networks can effectively learn multi-index models of the form $f^\star(\mathbf{x}) = g(\mathbf{U}\mathbf{x})$. These studies adopt certain modified training algorithms, such as layer-wise training. With sufficient feature learning, these networks can learn low-rank polynomials with a benign sample complexity of $\mathcal{O}(d^{\mathcal{O}(1)})$, which does not scale with the degree of the polynomial $g$. Empirically, fully connected networks trained via gradient descent on image classification tasks also capture low-rank features (Lee et al., 2007; Radhakrishnan et al., 2022). More recently, the learning of single-index and multi-index models is analyzed with more advanced algorithm framework or specified data structure. Mousavi-Hosseini et al. (2024) considers learning general multi-index models with two-layer neural networks

through a mean-field Langevin dynamics, Dandi et al. (2024b); Lee et al. (2024) goes beyond the traditional Correlational Statistical Query (CSQ) setting and consider algorithms that reuse samples for feature learning. Mousavi-Hosseini et al. (2023); Ba et al. (2023); Wang et al. (2024) considers learning linear features with structured data (such as data with a spiked covariance) rather than the commonly considered isotropic one. Cui et al. (2024); Dandi et al. (2024a) study the spectral structure revealed in the learned features with one huge gradient step through a spiked random feature model to understand the mechanism of feature learning in neural networks.

**Learning Nonlinear Features.** Previous studies indicate that neural networks can effectively learn specific hierarchies of nonlinear features. Safran and Lee (2022) shows that GD can efficiently learn functions such as $\mathbf{1}_{\|\mathbf{x}\| \geqslant \lambda}$ with a three-layer network. Ren et al. (2023) demonstrates that $\mathrm{ReLU}(1 - \|\mathbf{x}\|)$ can be learned by a multi-layer mean-field network. Moniri et al. (2024) studies the nonlinear feature learning capabilities of two-layer neural networks with one step of gradient descent. Allen-Zhu and Li (2019; 2020) explore learning target functions of the form $p + \alpha g \circ p$ with $p$ being the underlying feature through a three-layer residual network, though they either need $\alpha = o_d(1)$ or cannot reach vanishing error. More recent works have addressed a broader class of nonlinear features compared with the previous research and demonstrate that three-layer neural networks can learn these hidden features efficiently. Specifically, Nichani et al. (2023) demonstrates that a three-layer network trained with layer-wise GD algorithm effectively learns $g \circ p$ for a quadratic feature $p(\mathbf{x}) = \mathbf{x}^\top \mathbf{A} \mathbf{x}$ with an improved sample complexity of $\widetilde{\Theta}(d^4)$. Wang et al. (2023) further demonstrates that such a network can in fact efficiently learn $g \circ p$ for $p$ within a broad subclass of degree $k$ polynomials and optimizes the sample complexity to $\widetilde{\mathcal{O}}(d^k)$. However, all of these studies focus on a *single nonlinear feature*, limiting their applicability to scenarios involving multiple features. Our work addresses this gap by establishing the first theoretical guarantee for efficiently learning hierarchical polynomials of *multiple nonlinear features*, which significantly broadens the learnable function class and advances towards a better understanding of feature learning.

## 2 Preliminaries

### 2.1 Notations

We use bold letters to denote vectors and matrices. For a vector $\mathbf{v}$, we denote its Euclidean norm by $\|\mathbf{v}\|_2$. For a matrix $\mathbf{A}$, we denote its operator and Frobenius norm as $\|\mathbf{A}\|_2$ and $\|\mathbf{A}\|_\mathrm{F}$, respectively. For any positive integer $n$, we denote $[n] = \{1, 2, \ldots, n\}$. Moreover, for any indexes $i$ and $j$, we denote $\delta_{ij} = 1$ if $i = j$ and $0$ otherwise. We use $\mathcal{O}$, $\Theta$ and $\Omega$ to hide absolute constants. In addition, we denote $f \lesssim g$ when there exists some positive absolute constant $C$ with $f \leqslant Cg$. We use $\widetilde{\mathcal{O}}$, $\widetilde{\Theta}$ and $\widetilde{\Omega}$ to ignore logarithmic terms. For a function $f : \mathcal{X} \to \mathbb{R}$ and a distribution $v$ on $\mathcal{X}$, we denote $\|f\|_{L^p(\mathcal{X}, v)} = (\mathbb{E}_{\mathbf{x} \sim v}[|f(\mathbf{x})|^p])^{1/p}$. When the domain is clear from context, we write $\|f\|_{L^p}$ for simplicity. Finally, we write $\mathbb{E}_{\mathbf{x}}$ as the shorthand for $\mathbb{E}_{\mathbf{x} \sim v}$ sometimes.

### 2.2 Problem Setup

**Data distribution** Our aim is to learn the target function $f^\star : \mathcal{X} \to \mathbb{R}$, with $\mathcal{X} \subseteq \mathbb{R}^d$ being the input space. Throughout the paper, we assume $\mathcal{X} = \mathbb{S}^{d-1}(\sqrt{d})$, that is, the sphere with radius $\sqrt{d}$ in $d$ dimensions. Also, we consider the data distribution to be the uniform distribution on the sphere, i.e., $\mathbf{x} \sim \mathrm{Unif}(\mathcal{X})$, and we draw two independent datasets $\mathcal{D}_1, \mathcal{D}_2$, each with $n_1$ and $n_2$ i.i.d. samples, respectively. Thus, we draw $n_1 + n_2$ samples in total.

**Target function** For the target function $f^\star : \mathbb{R}^d \to \mathbb{R}$, we assume they are hierarchical functions of $r$ quadratic features

$$f^\star(\mathbf{x}) = g^\star(\mathbf{p}(\mathbf{x})) = g^\star\left(\mathbf{x}^\top \mathbf{A}_1 \mathbf{x}, \mathbf{x}^\top \mathbf{A}_2 \mathbf{x}, \ldots, \mathbf{x}^\top \mathbf{A}_r \mathbf{x}\right).$$

This structure represents a broad class of functions where $\mathbf{p}(\mathbf{x}) = [\mathbf{x}^\top \mathbf{A}_1 \mathbf{x}, \mathbf{x}^\top \mathbf{A}_2 \mathbf{x}, \ldots, \mathbf{x}^\top \mathbf{A}_r \mathbf{x}]^\top$ represents $r$ quadratic features, and $g^\star : \mathbb{R}^r \to \mathbb{R}$ is a link function. Here we consider the case $r \ll d$. To simplify our analysis while maintaining generality, we make the following assumptions:

**Assumption 1** (Orthogonal quadratic features). *For any $i, j \in [r]$, we suppose*

$$\mathbb{E}_{\mathbf{x}}\left[\mathbf{x}^\top \mathbf{A}_i \mathbf{x}\right] = 0, \quad \mathbb{E}_{\mathbf{x}}\left[(\mathbf{x}^\top \mathbf{A}_i \mathbf{x})(\mathbf{x}^\top \mathbf{A}_j \mathbf{x})\right] = \delta_{ij} \quad and \quad \|\mathbf{A}_i\|_{op} \leq \frac{\kappa_1}{\sqrt{d}}.$$

*Here we assume $\kappa_1 = \mathrm{poly}(\log d)$.*

The first assumption is equivalent to $\mathrm{tr}(\mathbf{A}_i) = 0$ for any $i \in [r]$. For $\mathbf{A}_i$ such that $\mathrm{tr}(\mathbf{A}_i) \neq 0$, we could simply subtract the mean of the feature to $\mathbf{A}_i' = \mathbf{A}_i - (\mathrm{tr}(\mathbf{A}_i)/d) \cdot \mathbf{I}$ so

$$\mathbf{x}^\top \mathbf{A}_i' \mathbf{x} = \mathbf{x}^\top (\mathbf{A}_i - (\mathrm{tr}(\mathbf{A}_i)/d) \cdot \mathbf{I})\mathbf{x} = \mathbf{x}^\top \mathbf{A}_i \mathbf{x} - \mathrm{tr}(\mathbf{A}_i).$$

The second assumption on the feature orthonormality can be attained via linear transformation on the features, preserving the overall function class. The third assumption on the operator norm bound ensures that the features are balanced, which is common in the non-linear feature learning literature (Nichani et al., 2023; Wang et al., 2023). Moreover, we note that when the entries of $\mathbf{A}_i$ are sampled i.i.d., the assumption is satisfied with high probability by standard random matrix arguments.

**Assumption 2** (Well-conditioned link function). *For the link function $g^\star$, we assume $g^\star$ is a degree-$p$ polynomial with $\mathbb{E}_{\mathbf{z}}\left[g^\star(\mathbf{z})^2\right] = \Theta(1)$, where $\mathbf{z} \sim \mathcal{N}(\mathbf{0}, \mathbf{I}_r)$ and $p \in \mathbb{N}$ is a constant. Moreover, we assume the expected Hessian $\mathbf{H} = \mathbb{E}_{\mathbf{z}}\left[\nabla^2 g^\star(\mathbf{z})\right] \in \mathbb{R}^{r \times r}$ is well-conditioned, i.e., there exists a constant $C_H$ such that $\lambda_{\min}(\mathbf{H}) \geqslant \frac{C_H}{\sqrt{r}}$.*

This assumption ensures the link function adequately emphasizes all $r$ features, preventing degeneracy to a lower-dimensional subspace. The second-moment condition is achievable through simple normalization.

**Assumption 3** (Prepocessed target function). *For the entire target function $f^\star$, we assume $\mathcal{P}_0(f^\star) = \mathbb{E}_{\mathbf{x}}[f^\star(\mathbf{x})] = 0$ and $\|\mathcal{P}_2(f^\star)\|_{L^2} \leq \kappa_2/\sqrt{d}$. Here $\mathcal{P}_k$ is the projection onto the function space of degree $k$ spherical harmonics on the sphere $\mathbb{S}^{d-1}(\sqrt{d})$, and $\kappa_2$ satisfies $\kappa_2 = \mathrm{poly}(r, \log d)$.*

We will give a rigorous definition of $\mathcal{P}_k$ in Section 2.3.1. This assumption is analogous to a pre-processing procedure conducted in Damian et al. (2022), which subtracts out the mean and linear component of the features from the target. The zero-mean condition ensures the network focuses on learning the function's variability rather than a constant offset. While Nichani et al. (2023); Wang et al. (2023) assume the link function $g$ has non-zero linear component, we rather assume $g$ has a *nearly zero linear component*, which prevents the target function from being dominated by a single linear combination of the quadratic features and keeps the learned representation space from collapsing to the one-dimensional space of that certain linear combination. This is an essential difference between single-feature and multi-feature learning, because our assumptions ensure that the network genuinely learns to *represent and distinguish all $r$ features* rather than conflate them, while assumptions in Nichani et al. (2023); Wang et al. (2023) represent a *degenerate case* that neural network may only learn the dominant linear combination of the $r$ features. We provide examples and counterexamples as follows.

**Remark 1.** These assumptions accommodate a wide range of target functions. For instance, $f^\star(\mathbf{x}) = \frac{1}{\sqrt{r}} \sum_{k=1}^r \left(\mathbf{x}^\top \mathbf{A}_k \mathbf{x}\right)^2 - \sqrt{r}$ satisfies Assumption 3 with $\kappa_2 \lesssim \sqrt{r}\kappa_1$ for any $\{\mathbf{a}_k\}_{k \in [r]}$ under Assumption 1. Moreover, for diagonal $\mathbf{A}_k$ with $\mathbf{A}_k = \mathrm{diag}(\mathbf{a}_k)$, where $\mathbf{a}_1, \mathbf{a}_2, \ldots, \mathbf{a}_r$ are orthogonal zero-sum vectors with entries $a_{k,i} \in \{\pm c/\sqrt{d}\}$, we can achieve $\kappa_2 = 0$. Here $c = \Theta(1)$ is a normalizing constant. Notably, linear combinations of features like $f(\mathbf{x}) = \frac{1}{\sqrt{r}} \sum_{k=1}^r \left(\mathbf{x}^\top \mathbf{A}_k \mathbf{x}\right)$ violate our assumptions, since it represents a degenerate case with $\|\mathcal{P}_2(f)\|_{L^2} = \|f\|_{L^2} = \Theta(1)$.

**Three-layer neural network** We adopt a standard three-layer neural network for learning the target functions. Let $m_1$, $m_2$ be the two hidden layer widths, and $\sigma_1$, $\sigma_2$ be two activation functions. Our learner is a three-layer neural network parameterized by $\theta = (\mathbf{a}, \mathbf{W}, \mathbf{b}, \mathbf{V})$, where $\mathbf{a} \in \mathbb{R}^{m_1}$, $\mathbf{W} \in \mathbb{R}^{m_1 \times m_2}$, $\mathbf{b} \in \mathbb{R}^{m_1}$, and $\mathbf{V} \in \mathbb{R}^{m_2 \times d}$. The network $f(\mathbf{x}; \theta)$ is defined as

$$f(\mathbf{x}; \theta) = \frac{1}{m_1} \sum_{i=1}^{m_1} a_i \sigma_1(\langle \mathbf{w}_i, \sigma_2(\mathbf{V}\mathbf{x}) \rangle + b_i) = \frac{1}{m_1} \sum_{i=1}^{m_1} a_i \sigma_1\left(\langle \mathbf{w}_i, \mathbf{h}^{(0)}(\mathbf{x}) \rangle + b_i\right). \quad (1)$$

Here, $\mathbf{w}_i \in \mathbb{R}^{m_2}$ is the $i$-th row of $\mathbf{W}$, and $\mathbf{h}^{(0)}(\mathbf{x}) := \sigma_2(\mathbf{V}\mathbf{x}) \in \mathbb{R}^{m_2}$ is the random feature embedding lying in the innermost layer. We initialize each row of $\mathbf{V}$ to be drawn uniformly on the sphere of radius $\sqrt{d}$, i.e., $\mathbf{v}_i^{(0)} \sim \mathrm{Unif}(\mathbb{S}^{d-1}(\sqrt{d}))$. For $\mathbf{a}$, $\mathbf{b}$ and $\mathbf{W}$, we use a symmetric initialization so that $f(\mathbf{x}; \theta^{(0)}) = 0$ (Chizat et al., 2019). Explicitly, we assume that $m_1$ is an even number and for any $j \in [m_1/2]$, we initialize the paramters as

$$a_j^{(0)} = -a_{m_1-j}^{(0)} \sim \mathrm{Unif}(\{-1, 1\}), \ \ \mathbf{w}_j^{(0)} = \mathbf{w}_{m_1-j}^{(0)} \sim \mathcal{N}(0, \epsilon \mathbf{I}_{m_2}), \ \text{and} \ b_j^{(0)} = b_{m_1-j}^{(0)} = 0.$$

Here $\epsilon > 0$ is a hyperparameter to control the magnitude of the initial neurons. Different from Nichani et al. (2023) where the weights $\mathbf{w}_j$ are initialized at zeros, we require a *random initialization*, which enables the learned weights to capture the multiple features in all directions instead of converging to a specific direction like the previous results for learning a single feature.

For the activation functions $\sigma_1$ and $\sigma_2$, we have the following assumptions:

**Assumption 4** (Activation Function). *We take the outer activation function $\sigma_1$ and the inner activation function $\sigma_2$ as*

$$\sigma_1(t) = \begin{cases} 2\left|t\right| - 1, & \left|t\right| \geqslant 1, \\ t^2, & \left|t\right| < 1. \end{cases} \quad \text{and} \quad \sigma_2(t) = \sum_{i=2}^{\infty} c_i Q_i(t), \tag{2}$$

*where $Q_i(t)$ is the $i$-th degree Gegenbauer polynomial in the $d$-dimensional space. Moreover, we assume there exist constants $C_\sigma$, $\alpha_\sigma$ such that $|\sigma_2(t)| \leq C_\sigma$ for $|t| \leq d$, and $\mathbb{E}_{\mathbf{x}}\left[\sigma_2^k(\mathbf{x}^\top \mathbf{1}_d)\right] \leq d^{-k}C_k$ for $k = 2, 4$. We assume $c_2 = \Theta(1)$, and $C_2$, $C_4$ and $\{c_i\}_{i=2}^{\infty}$ are all constants independent of $n$, $d$, $m_1$ and $m_2$.*

We remark the outer activation $\sigma_1$ is a slightly modified version of the absolute value function $|t|$, smoothed around the origin. The assumptions on $\sigma_2$ are based on the Gegenbauer expansion, often considered in the spherical analysis (introduced in Section 2.3.2). Compared to standard inner activations, we remove the constant term ($Q_0(t) = 1$) and the linear term ($Q_1(t) = t/d$) to focus on learning nonlinear features without low-order interference. Importantly, these assumptions on activation functions maintain significant generality. The assumptions on magnitude and moments are satisfied by many common activation functions with appropriate scaling. The core assumption in the Gegenbauer expansion is the non-zero component of $Q_2$, i.e., $c_2 = \Theta(1)$, which we rely on for a subspace recovery of the $r$ quadratic features while other assumptions are made to simplify our analysis since other components in inner activation will lead to useless noises or biases in the weights after training. Moreover, if we consider higher degree nonlinear features such as degree $q$ polynomials, we expect that $\sigma_2$ has sufficient emphasis on $Q_q$ for efficient feature learning.

**Remark 2.** $\sigma_2(t) = Q_2(t) = \frac{t^2 - d}{d(d-1)}$ is an example of the inner activation function.

**Training Algorithm** Following Nichani et al. (2023), our network is trained via layer-wise gradient descent with sample splitting. Throughout the training process, we freeze the innermost layer weights $\mathbf{V}$. In the first stage, the second layer weights $\mathbf{W}$ are trained for one step with a specified learning rate $\eta_1$ and weight decay $\lambda_1$. In the second stage, we reinitialize the bias $\mathbf{b}$ and train the outer layer weights $\mathbf{a}$ for $T - 1$ steps.

**Transfer Learning** We remark that our algorithm allows transfer learning of a different target function $f$ that shares the same features of the original target:

$$f^\star(\mathbf{x}) \to f(\mathbf{x}) = g\left(\mathbf{x}^\top \mathbf{A}_1 \mathbf{x}, \mathbf{x}^\top \mathbf{A}_2 \mathbf{x}, \ldots, \mathbf{x}^\top \mathbf{A}_r \mathbf{x}\right) \qquad \text{(transferred target)}$$

In this case, we switch the target function from $f^\star = g^\star(\mathbf{p})$ to $f = g(\mathbf{p})$ in the second training stage. For the loss function, we use the standard squared loss:

$$\hat{\mathcal{L}}^{(1)}(\theta) = \frac{1}{n_1} \sum_{\mathbf{x} \in \mathcal{D}_1} (f(\mathbf{x};\theta) - f^\star(\mathbf{x}))^2, \quad \hat{\mathcal{L}}^{(2)}(\theta) = \begin{cases} \frac{1}{n_2} \sum_{\mathbf{x} \in \mathcal{D}_2} (f(\mathbf{x};\theta) - f^\star(\mathbf{x}))^2 & \text{(original)}, \\ \frac{1}{n_2} \sum_{\mathbf{x} \in \mathcal{D}_2} (f(\mathbf{x};\theta) - f(\mathbf{x}))^2 & \text{(transferred)}. \end{cases}$$

This layer-wise training approach, combined with the ability to perform transfer learning, provides a powerful framework for learning and adapting to hierarchical functions with hidden features (Kulkarni and Karande, 2017; Damian et al., 2022; Nichani et al., 2023). The pseudocode for the entire training procedure is presented in Algorithm 1.

## 2.3 TECHNICAL BACKGROUND: ANALYSIS OVER THE SPHERE

We briefly introduce spherical harmonics and Gegenbauer polynomials, which forms the foundation of our analysis over the sphere $\mathbb{S}^{d-1}(\sqrt{d})$. For more details, see Appendix B.5.

### 2.3.1 SPHERICAL HARMONICS

Let $\tau_{d-1}$ be the uniform distribution on $\mathbb{S}^{d-1}(\sqrt{d})$. Consider functions in $L^2(\mathbb{S}^{d-1}(\sqrt{d}), \tau_{d-1})$, with scalar product and norm denoted as $\langle \cdot, \cdot \rangle_{L^2}$ and $\|\cdot\|_{L^2}$. For $\ell \in \mathbb{Z}_{\geqslant 0}$, let $V_{d,\ell}$ be the linear space of

---

**Algorithm 1** Layer-wise training algorithm

---

**Input:** Learning rates $\eta_1, \eta_2$, weight decay $\lambda_1, \lambda_2$, parameter $\epsilon$, number of steps $T$

1: **initialize a, b, W** *and* **V**.
2: **train W on dataset** $\mathcal{D}_1$
3: $\quad \Big| \quad \mathbf{W}^{(1)} \leftarrow \mathbf{W}^{(0)} - \eta_1[\nabla_{\mathbf{W}}\hat{\mathcal{L}}^{(1)}(\theta) + \lambda_1 \mathbf{W}^{(0)}]$
4: **end**
5: **re-initialize**
6: $\quad \Big| \quad b_i^{(1)} \sim \text{Unif}([-3,3]), \ i \in [m_1]$
$\quad \quad \quad \mathbf{a}^{(1)}, \mathbf{V}^{(1)} \leftarrow \mathbf{a}^{(0)}, \mathbf{V}^{(0)}$
$\quad \quad \quad \theta^{(1)} \leftarrow (\mathbf{a}^{(1)}, \mathbf{W}^{(1)}, \mathbf{b}^{(1)}, \mathbf{V}^{(0)})$
7: **end**
8: **train a on dataset** $\mathcal{D}_2$
9: $\quad \Big| \quad$ **for** $t = 2$ **to** $T$ **do**
10: $\quad \quad \Big| \quad \mathbf{a}^{(t)} \leftarrow \mathbf{a}^{(t-1)} - \eta_2[\nabla_{\mathbf{a}}\hat{\mathcal{L}}^{(2)}(\theta^{(t-1)}) + \lambda_2 \mathbf{a}^{(t-1)}]$
11: $\quad \quad$ **end**
12: **end**
13: **return** Prediction function $f(\cdot; \theta^{(T)})$: $\mathbf{x} \to \frac{1}{m_1}\langle \mathbf{a}^{(T)}, \sigma_1(\mathbf{W}^{(1)}\mathbf{h}^{(0)}(\mathbf{x}) + \mathbf{b}^{(1)})\rangle$

---

homogeneous harmonic polynomials of degree $\ell$ restricted on $\mathbb{S}^{d-1}(\sqrt{d})$. The set $\{V_{d,\ell}\}_{\ell \geqslant 0}$ forms an orthogonal basis of the $L^2$ space, with dimension $\dim(V_{d,\ell}) = \Theta(d^\ell)$. For each $\ell \in \mathbb{Z}_{\geqslant 0}$, the spherical harmonics $\{Y_{\ell,j}\}_{j \in [B(d,\ell)]}$ form an orthonormal basis of $V_{d,\ell}$. Moreover, we denote by $\mathcal{P}_k$ the orthogonal projections to $V_{d,k}$, which can be written as

$$\mathcal{P}_k(f)(\mathbf{x}) = \sum_{\ell=1}^{B(d,k)} \langle f, Y_{k,\ell}\rangle_{L^2} Y_{k,\ell}(\mathbf{x}).$$

We also define $\mathcal{P}_{\leq \ell} \equiv \sum_{k=0}^{\ell} \mathcal{P}_k$, $\mathcal{P}_{>\ell} \equiv \mathbf{I} - \mathcal{P}_{\leq \ell}$, $\mathcal{P}_{<\ell} \equiv \mathcal{P}_{\leq \ell-1}$, and $\mathcal{P}_{\geq \ell} \equiv \mathcal{P}_{>\ell-1}$.

### 2.3.2 GEGENBAUER POLYNOMIALS

Corresponding to the degree $\ell$ spherical harmonics in the $d$-dimension space, the $\ell$-th Gegenbauer polynomial $Q_\ell : [-d, d] \to \mathbb{R}$ is a polynomial of degree $\ell$. The set $\{Q_\ell\}_{\ell \geqslant 0}$ forms an orthogonal basis on $L^2([-d, d], \widetilde{\tau}_{d-1})$, where $\widetilde{\tau}_{d-1}$ is the distribution of $\sqrt{d}\langle \mathbf{x}, \mathbf{e}_1\rangle$ when $\mathbf{x} \sim \tau_{d-1}$. In particular, these polynomials are normalized so that $Q_\ell(d) = 1$. We present the explicit forms of Gegenbauer polynomials of degree no more than 2:

$$Q_0(t) = 1, \quad Q_1(t) = \frac{t}{d}, \quad \text{and} \quad Q_2(t) = \frac{t^2 - d}{d(d-1)}.$$

Gegenbauer polynomials are directly related to spherical harmonics, leading to a number of elegant properties. We provide further details on these properties in Appendix B.5.

## 3 MAIN RESULTS

The following is our main theorem, which bounds the population absolute loss of Algorithm 1:

**Theorem 1.** *Suppose* $n_1 = \widetilde{\Omega}(d^4)$ *and* $m_2 = \widetilde{\Omega}(d^6)$. *Let* $\hat{\theta}$ *be the output of Algorithm 1 after* $T = \text{poly}(n_1, n_2, m_1, m_2, d)$ *steps. Then, there exists a set of hyper-parameters* $(\epsilon, \eta_1, \eta_2, \lambda_1, \lambda_2)$ *such that, with high probability over the initialization of parameters and draws of* $\mathcal{D}_1, \mathcal{D}_2$, *we have*

$$\mathbb{E}_{\mathbf{x}}\Big[\big|f(\mathbf{x}; \hat{\theta}) - f^\star(\mathbf{x})\big|\Big] = \widetilde{\mathcal{O}}\left(\underbrace{\sqrt{\frac{r^p \kappa_2^{2p}}{\min(m_1, n_2)}}}_{\text{Complexity of } g^\star} + \underbrace{\sqrt{\frac{d^6 r^{p+1}}{m_2}} + \sqrt{\frac{d^2 r^{p+1}}{n_1}} + \frac{r^{p+2}}{d^{1/6}}}_{\text{Feature Learning Error}}\right).$$

*Moreover, for any other degree $p$ polynomial $g : \mathbb{R}^r \to \mathbb{R}$ with $\|g\|_{L^2} \lesssim 1$, by substituting the target function $f^\star = g^\star \circ \mathbf{p}$ by $f = g \circ \mathbf{p}$ in the second training stage, we can achieve the same result for learning the new target function.*

The full proof is provided in Appendix E.1. To interpret the results, we provide the following discussion of Theorem 1.

**Feature learning error** This terms quantifies the requirements on the first-stage sample complexity and the inner width to sufficiently capture the non-linear features. Given $d \gg r$, if the width $m_2 = \widetilde{\Omega}(d^6 r^{p+1})$ and the sample size $n_1 = \widetilde{\Omega}(d^4 + d^2 r^{p+1})$, we can fully capture the underlying feature information and approximate any degree $p$ polynomials of the features. We will demonstrate how Algorithm 1 learns these features through the learned representations in Proposition 1 and express hierarchical polynomials in Proposition 2.

**Complexity of $g^\star$** This term is the second-stage sample (and width) complexity given that the $r$ features have been fully captured in the first stage. Moreover, for a sufficiently preprocessed target function, i.e., $\kappa_2 = \mathcal{O}(1)$, we achieve the standard results of $\widetilde{\mathcal{O}}(r^p)$ complexity in learning a degree-$p$ polynomial in the $r$-dimensional space in the kernel regime.

**Transfer learning** Leveraging the two-stage structure of training, we can learn a different target function in the second stage that shares the same features with the original target. This also supports the fact that we have fully captured the information of the $r$ nonlinear features in the first stage, making it possible for the efficient learning with a different polynomial head $g$. Moreover, by viewing the first stage as a pre-training process with $\widetilde{\Omega}(d^4 + d^2 r^{p+1})$ samples, only additional $\widetilde{\mathcal{O}}(r^p \kappa_2^{2p})$ samples are required to learn any degree $p$ polynomial of the features, which gets rid of the polynomial dependence on the ambient dimension of $d$.

**Comparison with previous works** Compared with the sample complexity of $\widetilde{\Omega}(d^2 r + dr^p)$ in Damian et al. (2022) for learning multi-index models, we have a similar polynomial dependence on $r$, and the dependence on $d$ increases from $d^2$ to $d^4$ because of the increased complexity of quadratic features rather than linear ones. Moreover, our approach significantly improves upon the $\Theta(d^{2p})$ sample complexity required by kernel methods to learn degree $p$ polynomials of quadratic features (i.e., degree $2p$ polynomials of the input). Crucially, our polynomial dependence on $d$ in the overall sample complexity is independent of the degree $p$ of the link function $g$.

**Near optimality of the sample complexity** We remark that our sample complexity of $\widetilde{\mathcal{O}}(d^4)$ is nearly optimal with respect to $d$ for all algorithms that use one step of gradient descent for feature learning. Our assumptions on the target functions imply that the leap index[1] of our target functions are basically 4 (more specifically, the second order information of $g \circ \mathbf{p}$, where $\mathbf{p}$ are quadratic features), and we also utilize $\mathcal{P}_4(f)$ for recovering the subspace of the $r$ quadratic features, which will be discussed in details in Section 4. Dandi et al. (2023b) indicates that $\Omega(d^4)$ samples are required for an efficient learning of terms in $\mathcal{P}_4(f^\star)$, which substantiates the near optimality of our result.

## 4 PROOF ROADMAP OF THEOREM 1

The proof of Theorem 1 unfolds in two training stages. First, by a novel universality argument (Lemma 1), we show that after the first training stage, with sufficient training samples, the network learns to fully extract out the hidden features $\mathbf{p}$ (Proposition 1). Next, we show that during the second stage, the network is capable of expressing the link function with a mild outer width $m_1$ (Proposition 2). We conclude the proof through standard Rademacher complexity analysis to quantify the generalization error of the second-stage model (detailed in Appendix E.1).

### 4.1 STAGE 1: LEARNING THE FEATURES

We provide a brief analysis on the learned representations after the first training stage. Denote $\mathbf{w}_j = \epsilon^{-1} \mathbf{w}_j^{(0)} \sim \mathcal{N}(0, \mathbf{I}_{m_2})$. According to Algorithm 1, by setting $\epsilon$ sufficiently small, after one-step gradient descent on $\mathbf{W}$, we know for each $j \in [m_1]$,

$$\eta_1 \nabla_{\mathbf{w}_j^{(0)}} \mathcal{L}(\theta^{(0)}) = -\eta_1 \frac{a_j^{(0)}}{m_1} \cdot \frac{1}{n_1} \sum_{\mathbf{x} \in \mathcal{D}_1} f^*(\mathbf{x}_i) \mathbf{h}^{(0)}(\mathbf{x}_i) \sigma_1' \left( \langle \epsilon \mathbf{w}_j, \mathbf{h}^{(0)}(\mathbf{x}_i) \rangle \right)$$

$$\underset{\epsilon \to 0}{\to} -\frac{2\epsilon \eta_1}{m_1} a_j^{(0)} \cdot \frac{1}{n_1} \sum_{\mathbf{x} \in \mathcal{D}_1} f^*(\mathbf{x}_i) \mathbf{h}^{(0)}(\mathbf{x}_i) \mathbf{h}^{(0)}(\mathbf{x}_i)^\top \mathbf{w}_j.$$

---

[1] The leap index of a target function $f^\star$ is the first integer $\ell$ that $\mathcal{P}_\ell f^\star \neq 0$. Our assumptions imply a diminishing $\mathcal{P}_{<4}(f^\star)$ and a non-degenerate $\mathcal{P}_4(f^\star)$ as $d \to \infty$.

By taking $\eta_1 = \frac{m_1}{2\epsilon m_2} \cdot \eta$ for some $\eta > 0$ to be chosen later and $\lambda_1 = \eta_1^{-1}$, we have

$$\mathbf{w}_j^{(1)} = \mathbf{w}_j^{(0)} - \eta_1 \left[ \nabla_{\mathbf{w}_j^{(0)}} \mathcal{L}(\theta^{(0)}) + \lambda_1 \mathbf{w}_j^{(0)} \right] = \frac{\eta a_j^{(0)}}{m_2} \cdot \frac{1}{n_1} \sum_{\mathbf{x} \in \mathcal{D}_1} f^*(\mathbf{x}_i) \mathbf{h}^{(0)}(\mathbf{x}_i) \mathbf{h}^{(0)}(\mathbf{x}_i)^\top \mathbf{w}_j.$$

Then for any second-stage training sample $\mathbf{x}' \in \mathcal{D}_2$, the inner-layer representation becomes

$$\left\langle \mathbf{w}_j^{(1)}, \sigma_2(\mathbf{V}\mathbf{x}') \right\rangle = \frac{\eta a_j^{(0)}}{m_2} \left\langle \frac{1}{n_1} \sum_{\mathbf{x} \in \mathcal{D}_1} f^*(\mathbf{x}_i) \mathbf{h}^{(0)}(\mathbf{x}_i) \mathbf{h}^{(0)}(\mathbf{x}_i)^\top \mathbf{w}_j, \mathbf{h}^{(0)}(\mathbf{x}') \right\rangle$$

$$= \eta a_j^{(0)} \cdot \left\langle \mathbf{w}_j, \underbrace{\frac{1}{n_1 m_2} \sum_{\mathbf{x} \in \mathcal{D}_1} f^\star(\mathbf{x}_i) \langle \mathbf{h}^{(0)}(\mathbf{x}_i), \mathbf{h}^{(0)}(\mathbf{x}') \rangle \mathbf{h}^{(0)}(\mathbf{x}_i)}_{\mathbf{h}^{(1)}(\mathbf{x}')} \right\rangle.$$

Our main contribution in this part is that the first-step trained presentations representations $\mathbf{h}^{(1)}(\mathbf{x})$ approximately spans the space of the target features $(\mathbf{x}^\top \mathbf{A}_1 \mathbf{x}, \mathbf{x}^\top \mathbf{A}_2 \mathbf{x}, \dots, \mathbf{x}^\top \mathbf{A}_r \mathbf{x})$. Thus, the target features $\mathbf{p}(\mathbf{x})$ can be reconstructed through a linear transformation from the learned representations $\mathbf{h}^{(1)}(\mathbf{x})$, which is formalized in the following proposition.

**Proposition 1** (Reconstruct the feature). *Suppose $m_2, n_1 = \widetilde{\Omega}(d^4)$. With high probability jointly on $\mathbf{V}$, $\mathcal{D}_1$ and $\mathcal{D}_2$, there exists a matrix $\mathbf{B}^\star \in \mathbb{R}^{r \times m_2}$ such that for any $\mathbf{x} \in \mathcal{D}_2$, we have*

$$\left\| \mathbf{B}^\star \mathbf{h}^{(1)}(\mathbf{x}) - \mathbf{p}(\mathbf{x}) \right\|_2 = \widetilde{\mathcal{O}} \left( \frac{d^3 r}{\sqrt{m_2}} + \frac{dr}{\sqrt{n_1}} + \frac{r^{\frac{p+5}{2}}}{d^{1/6}} \right). \tag{3}$$

The proof is provided in Appendix C.3. We summarize the main idea of the proof as follows.

**Universality of features** The foundation of the proof lies in the universality result that the joint distribution of the multiple features $\mathbf{p}$ is approximately multivariate standard Gaussian:

$$\left( \mathbf{x}^\top \mathbf{A}_1 \mathbf{x}, \mathbf{x}^\top \mathbf{A}_2 \mathbf{x}, \dots, \mathbf{x}^\top \mathbf{A}_r \mathbf{x} \right) \overset{\mathrm{d}}{\approx} \mathcal{N}(\mathbf{0}_r, \mathbf{I}_r), \quad d \gg r.$$

It is worth mentioning that we provide a general universality theory that quantifies the difference between the distribution of any $r$-dimensional function (not limited in quadratic forms) and the $r$-dimensional Gaussian distribution, which is presented in Lemma 1.

**Lemma 1** (Universality of vector-valued functions). *Suppose $\mathbf{X} \sim \mathcal{N}(\mathbf{0}, \mathbf{I}_d)$ is an $d$-dimensional standard Gaussian variable. If a function $\mathbf{p} : \mathbb{R}^d \to \mathbb{R}^r$ satisfies $\mathbb{E}_{\mathbf{X}}[\mathbf{p}(\mathbf{X})] = \mathbf{0}_r$ and $\mathrm{Cov}(\mathbf{p}(\mathbf{X}), \mathbf{p}(\mathbf{X})) = \mathbf{I}_r$, then we have*

$$W_1(\mathrm{Law}(\mathbf{p}(\mathbf{X})), \mathcal{N}(\mathbf{0}, \mathbf{I}_r)) \le \frac{4}{\sqrt{\pi}} \left( \sum_{i=1}^r \mathbb{E}\left[ \|\nabla p_i(\mathbf{X})\|_2^4 \right]^{1/4} \right) \left( \sum_{j=1}^r \mathbb{E}\left[ \|\nabla^2 p_j(\mathbf{X})\|_{\mathrm{op}}^4 \right]^{1/4} \right).$$

*Here $\mathbf{p}(\mathbf{x}) = [p_1(\mathbf{x}), p_2(\mathbf{x}), \dots, p_r(\mathbf{x})]^\top$ and $W_1$ denotes the Wasserstein-1 distance.*

The proof is provided in Appendix B.2. This lemma extends the previous universality results of univariate Gaussian approximation theory (Chatterjee, 2007) to the multivariate version and could be of independent interest for the field of high dimensional probability theory. As a corollary, when we take $\mathbf{p}$ to be $r$ quadratic features satisfying Assumption 1, we ensure the $W_1$ distance is bounded by $\widetilde{\mathcal{O}}(r^2/\sqrt{d})$ (see Lemma 16 in the appendix for the formal statement). This approximation error finally contributes to third term in the error bound of Proposition 1 (Equation (3)).

**Utilizing the second-order information of $g^\star$** Lemma 1 establishes a crucial link between our model and the multi-index model studied by Damian et al. (2022). This connection allows us to simplify the analysis on non-linear features and utilize the second-order information of the link function $g^\star$ to fully recover the feature space. In the context of multi-index models where $f^\star(\mathbf{x}) = g^\star(\mathbf{p}(\mathbf{x}))$ with $\mathbf{p}(\mathbf{x}) = \mathbf{U}\mathbf{x}$, it has been shown that for a prepossessed target with a non-degenerate expected Hessian $\mathbf{H} = \mathbb{E}_{\mathbf{z}}[\nabla^2 g^\star(\mathbf{z})]$, the learned representations, dominated by the degree 2 component of

$f^*$ which takes form $\mathbb{E}_{\mathbf{x}}\left[f^\star(\mathbf{x})\mathbf{x}^{\otimes 2}\right] \approx \mathbf{U}^\top \mathbf{H}\mathbf{U}$, are spanned by $\{\mathbf{u}_i \otimes \mathbf{u}_j\}_{i,j\in[r]}$. Extending this to our setting with quadratic features and applying the universality argument from Lemma 1, we demonstrate that the degree 4 component of our $f^*$, namely $\mathbb{E}_{\mathbf{x}}\left[f^*(\mathbf{x})\mathbf{Y}_2(\mathbf{x})^{\otimes 2}\right]$, is approximately spanned by the quantities $\{\mathbf{A}_i \otimes \mathbf{A}_j\}_{i,j\in[r]}$, which is formalized in Proposition 3 in Appendix C.1. Here $\mathbf{Y}_2(\mathbf{x})$ represents the tensorized quadratic spherical harmonics. Under Assumption 3, it turns out that after the first step of GD (Stage 1 of Algorithm 1), the learned representations are dominated by this degree 4 component (Proposition 4 in Appendix C.2). This domination enables efficient recovery of the "span" of the hidden features $\mathbf{p}$. For a visual representation of our proof strategy, we also present our main idea of the proof in Figure 1. Remarkably, we find that the reconstruction matrix admits a surprisingly simple form of $\mathbf{B}^\star \propto \mathbf{H}^{-1}[\mathbf{p}(\mathbf{v}_1), \mathbf{p}(\mathbf{v}_2), \ldots, \mathbf{p}(\mathbf{v}_{m_2})]$. We provide empirical support for the effectiveness of this reconstruction through experiments in Section A.

Figure 1: The proof idea of Proposition 1. Block 1 characterizes the constant and linear terms of $g^\star$, which is approximately equivalent to the low-order terms $\mathcal{P}_{<4}(f^\star)$ by our universality theory and results into biases in the learned weights $\mathbf{h}^{(1)}(\mathbf{x}')$ after Stage 1. This bias is vanishing with $d \to \infty$ by our assumptions on $\mathcal{P}_0(f^\star)$ and $\mathcal{P}_2(f^\star)$. Block 2 describes the second-order information of $g^\star$ (approximately $\mathcal{P}_4(f^\star)$), which is of the greatest importance and captured by the quadratic component $c_2 Q_2(\cdot)$ in the inner activation $\sigma_2(\cdot)$ and converted into quantities spanned by the $r$ quadratic features $\mathbf{p}$. Block 3 represents the remaining terms of $f^\star$, which leads to high-order nuisance in the learned weights, but still dominated by the second term due to Assumption 2 when $d$ is large, which enables us to utilize the terms in blue (resulted from Block 2) to reconstruct the features efficiently.

### 4.2 STAGE 2: LEARNING THE LINK FUNCTION

By the deduction above, after the first training stage, the model becomes a random-feature model (Rahimi and Recht, 2007):

$$f(\mathbf{x}';\theta) = \frac{1}{m_1}\sum_{j=1}^{m_1} a_j \sigma_1\left(\eta a_j^{(0)}\langle \mathbf{w}_j, \mathbf{h}^{(1)}(\mathbf{x}')\rangle + b_j^{(1)}\right). \tag{4}$$

Here $\theta = (\mathbf{a}, \mathbf{W}^{(1)}, \mathbf{b}^{(1)}, \mathbf{V})$, with $\mathbf{a} = [a_1, a_2, \ldots, a_{m_1}]^\top \in \mathbb{R}^{m_1}$ being the trainable parameters in the second stage. Leveraging the construction in Proposition 1, we can construct a corresponding weight vector $\mathbf{a}$ in the outer layer to express the polynomial $g(\mathbf{B}^\star \mathbf{h}^{(1)}(\mathbf{x})) \approx g(\mathbf{p}(\mathbf{x}))$.

**Proposition 2** (Expressivity of the second-stage model). *Suppose $g$ is a degree $p$ polynomial with $\|g\|_{L^2} \lesssim 1$. Then there exists a learning rate $\eta$ such that, with high probability over $\mathcal{D}_1, \mathcal{D}_2, \mathbf{W}$ and $\mathbf{V}$, there exists $\mathbf{a}^\star \in \mathbb{R}^{m_1}$ such that the parameter $\theta^\star = (\mathbf{a}^\star, \mathbf{W}^{(1)}, \mathbf{b}^{(1)}, \mathbf{V})$ achieves a small empirical loss:*

$$\frac{1}{n_2}\sum_{\mathbf{x}\in\mathcal{D}_2}(f(\mathbf{x};\theta^\star) - g(\mathbf{p}(\mathbf{x})))^2 = \widetilde{\mathcal{O}}\left(\frac{\|\mathbf{a}^\star\|_2^2}{m_1^2} + \frac{d^6 r^{p+1}}{m_2} + \frac{d^2 r^{p+1}}{n_1} + \frac{r^{2p+4}}{d^{1/3}}\right).$$

*Here $\mathbf{a}^\star$ satisfies $\|\mathbf{a}^\star\|_2^2 = \widetilde{\mathcal{O}}\left(m_1 r^p \kappa_2^{2p}\right)$.*

The proof is provided in Appendix D.1. We provide following discussions.

**Error propagation**  To explain the increased polynomial dependence on $r$, we remark that the approximation error in Proposition 1 gets multiplied by the averaged Lipschitz smoothness of the link function $g$, which is upper bounded by $\mathcal{O}(r^{\frac{p-1}{2}})$. This product is then squared due to the use of squared loss in Proposition 2.

**Reduced complexity of a**  Moreover, we remark that the complexity of $\mathbf{a}$, i.e., $\|\mathbf{a}\|_2$, gets rid of the polynomial dependence on $d$, which is greatly reduced compared with a naive random-feature model that requires $\|\mathbf{a}\|_2^2 = \Theta(m_1 d^{2p})$. This directly saves the second-stage sample complexity $n_1$ and the outer width $m_1$, since $n_1, m_2 = \Theta(m_1^{-1}\|\mathbf{a}^\star\|_2^2)$ is required for efficient approximation and generalization (Ghorbani et al., 2021). We also examine this reduced dependency by comparing our model with a naive random feature model in learning hierarchical target functions in Section A.

**Arbitrariness of $g$**  Thanks to the two-stage architecture and the sufficient learning of the features, the choice on the link function $g$ can be an arbitrary degree $p$ polynomial, not limited to the truth target $g^\star$. This allows us to conduct transfer learning tasks in Stage 2 of Algorithm 1.

Finally, by standard Rademacher complexity analysis on the random feature model presented in Appendix E.1, we conclude our proof.

## 5  Conclusions and Discussions

**Comparison with Nichani et al. (2023); Wang et al. (2023)**  As discussed under assumptions 3 and the initialization of our neural networks, our work differs significantly in the targets of interests, the parametrization of neural networks, the mathematical strategies, and the intuitions behind the results. Our assumptions ensure a nearly zero linear component and a non-degenerate second order term of the link function $g$ which significantly contrasts the assumptions posed in Nichani et al. (2023); Wang et al. (2023) that emphasize the linear component. Our random initialization (rather than a deterministic initialization used in the aforementioned two works) in the weights of the three-layer neural networks allows the learned weights to capture multiple features in all directions simultaneously after training rather than converge to a single direction. We develop a novel universality result to relate multiple nonlinear features to multivariate Gaussian, while these two works adopt existing result of the approximate Stein's lemma which only applies to single nonlinear feature. Most importantly, subspace recovery is completely different from and also significantly harder than single feature recovery considered in Nichani et al. (2023); Wang et al. (2023).

**Conclusions**  In this work, we have shown the provable capabilities of three-layer networks in efficiently learning targets of multiple quadratic features. Leveraging a novel universality result, we have shown that one gradient step suffices for a full recovery of the subspace spanned by multiple quadratic features. In addition, leveraging the learned features, we have demonstrated the transfer learning capabilities of this three-layer neural network with a constant polynomial sample complexity guarantee. To the best of our knowledge, this is the first theoretical result of efficiently learning such a board target function class of multiple nonlinear features with neural networks. We have made a great improvement on the sample complexity by highlighting feature learning compared to kernel methods.

**Future works**  First, it may be possible that the sample complexity bound of $\widetilde{\mathcal{O}}(d^4)$ could be improved to the information-theoretic optimal sample complexity $\mathcal{O}(d^2)$ in learning general hierarchical polynomials of quadratic features. We think that this result may be achieved when we consider more advanced algorithms that utilize the samples more thoroughly such as using multiple steps of GD, which could be a great future extension of our work. Moreover, our methodology is not inherently limited to quadratic features. The principles shown in Figure 1 and techniques developed here give a foundation for understanding the learning of even more complex function classes. Another natural future direction of our work is to understand whether and when our results can be generalized to learning multiple high-degree features.

## 6 ACKNOWLEDGMENTS

The authors would like to thank Alex Damian and Yunwei Ren for the many insightful discussions. JDL acknowledges support of Open Philanthropy, NSF IIS 2107304, NSF CCF 2212262, NSF CA-REER Award 2144994, and NSF CCF 2019844.

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

# Appendix

# A  NUMERICAL EXPERIMENTS

We empirically verify Theorem 1 and Proposition 1. We consider learning functions with $r = 3$ quadratic features. Regarding the target function, we choose the target functions to be of the form

$$f_{d,p}^{\star}(\mathbf{x}) = \frac{f_{d,p}(\mathbf{x}) - \mathbb{E}\left[f_{d,p}(\mathbf{x})\right]}{\sqrt{\mathrm{Var}[f_{d,p}(\mathbf{x})]}}, \quad \text{with } f_{d,p}(\mathbf{x}) = \sum_{i=1}^{r} \left(\mathbf{x}^{\top}\mathbf{A}_1\mathbf{x}\right)^p, \ p \in \mathbb{N}. \tag{5}$$

For the underlying features, we take $\mathbf{p}(\mathbf{x}) = [\mathbf{x}^{\top}\mathbf{A}_1\mathbf{x}, \mathbf{x}^{\top}\mathbf{A}_2\mathbf{x}, \mathbf{x}^{\top}\mathbf{A}_3\mathbf{x}]^{\top}$ with $\mathbf{A}_k = \mathrm{diag}\left(c \cdot \mathbf{a}_k\right)$, and $c > 0$ is a normalizing constant. To ensure the orthogonality of the features and $\mathrm{tr}(\mathbf{A}_k) = 0$, we choose the ambient dimension $d$ to be divisible by $4$ and take $\mathbf{a}_k$ to be

$$\mathbf{a}_1 = \mathrm{Vec}\left([\mathbf{1}, \mathbf{1}, -\mathbf{1}, -\mathbf{1}]\right), \ \mathbf{a}_2 = \mathrm{Vec}\left([\mathbf{1}, -\mathbf{1}, \mathbf{1}, -\mathbf{1}]\right), \ \text{and} \ \mathbf{a}_3 = \mathrm{Vec}\left([\mathbf{1}, -\mathbf{1}, -\mathbf{1}, \mathbf{1}]\right).$$

Here $\mathbf{1}$ is a vector of ones in $d/4$ dimensions, and $c = \sqrt{\frac{d+2}{2d^2}}$ to ensure that $\mathbb{E}_{\mathbf{x}}\left[(\mathbf{x}^{\top}\mathbf{A}_k\mathbf{x})^2\right] = 1$ for each $k = 1, 2, 3$.

For the network architecture, we choose $\sigma_1$ as per (2) and $\sigma_2 = Q_2$, with network sizes set to $m_1 = 10000$ and $m_2 = 20000$. We compare our proposed model (4) (given by Algorithm 1) against the naive random-feature model defined as

$$f^{\mathrm{RF}}(\mathbf{x}'; \theta) = \frac{1}{m_1}\sum_{j=1}^{m_1} a_j \sigma_1\left(\eta a_j^{(0)}\langle \mathbf{w}_j, \mathbf{h}^{(0)}(\mathbf{x}')\rangle + b_j^{(1)}\right), \tag{6}$$

where $\mathbf{a}$ is the only trainable parameter throughout the training process. Our experiments involve learning $f_{d,p}^{\star}$ with $p = 4$ and $d \in \{8, 16, 32\}$. To examine our model's transfer learning capabilities, we also train the model on an initial target function $f_{d,2}^{\star}$ with $d = 16$ and $n_1 = 2^{16}$ in the first stage, then transfer to targets $f_{d,p}^{\star}$ with $p = 4, 6, 8$. For each task, we explore a range of sample sizes from $2^8$ to $2^{16}$. The results of these experiments are presented in Figure 2.

**Improved sample complexity and Polynomial dependence on** $d$   The left panel of Figure 2 demonstrates that our model outperforms the naive random-feature model across all dimensions. As the dimension $d$ increases, both models show larger test errors, but our model exhibits less sensitivity to $d$. This aligns with our theoretical analysis in Theorem 1 that the sample complexity of kernel methods should be $\Omega(d^{2p-4})$ times greater than that of our model. Moreover, we redraw Figure 2 by plotting the test error against $\log_d n$. As shown in Figure 3, the loss curves for our model (Algorithm 1) align closely for different values of $d$, indicating that it achieves low error rates with only $\widetilde{\mathcal{O}}(d^4)$ samples. In stark contrast, the naive random feature model exhibits significant separation between curves for different $d$ values, requiring more than $\widetilde{\mathcal{O}}(d^4)$ samples to achieve comparable error rates. This graphical evidence powerfully demonstrates how our approach eliminates the dependence on dimension $\Theta(d^{2p})$ presented in kernel methods, resulting in substantially improved sample complexity in high-dimensional settings.

**Efficient transfer learning**   The right panel of Figure 2 showcases our algorithm's strong transfer learning capabilities. our algorithm successfully learns all three transferred target functions with benign second-stage sample complexity. Notably, as the degree $p$ increases, the test error grows no faster than $r^p$, which is significantly slower than $d^{2p}$. This supports our theoretical result that the second-stage sample complexity depends on the number of features $r$ rather than the ambient dimension $d$, underscoring our model's strong transfer learning capabilities.

**Accurate reconstruction of quadratic features**   To further demonstrate our model's feature learning capabilities, we extract the learned features $\mathbf{h}^{(1)}$ after the first training stage of Algorithm 1, using $f_{16,2}$ as the target. We then reconstruct these features using a linear transformation $\mathbf{B}^{\star} \in \mathbb{R}^{r \times m_2}$, as described in Proposition 1. We examine how reconstruction accuracy changes with first-stage sample sizes. Figure 4 shows the correlation between true and reconstructed features for each sample size. As $n_1$ increases, all features are better approximated simultaneously. Notably, $d^4$ samples prove sufficient to reconstruct the features with high accuracy, supporting our model's effective feature learning ability.

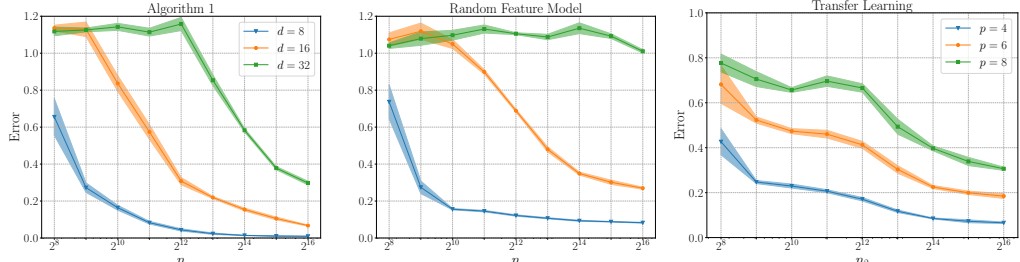

(a) Comaprison between Algorithm 1 and the random-feature model    (b) Performance of transfer learning

Figure 2: For the left panel, Algorithm 1 uses two equally sized datasets, while the random feature model uses the full dataset. For the right panel, we conduct transfer learning with $n_1 = 2^{16}$ pretraining samples and plot the dependence on $n_2$. The figure reports the mean and normalized standard error of the test error using $10,000$ fresh samples, based on $5$ independent experimental instances.

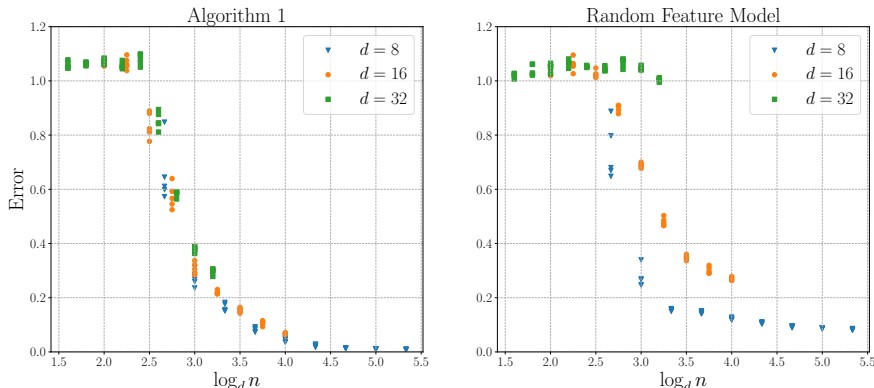

Figure 3: Test error of Algorithm 1 and the naive random feature models with x-axis being the relative sample complexity $(\log_d n)$. We plot the test error of $5$ independent instances for each $d \in \{8, 16, 32\}$.

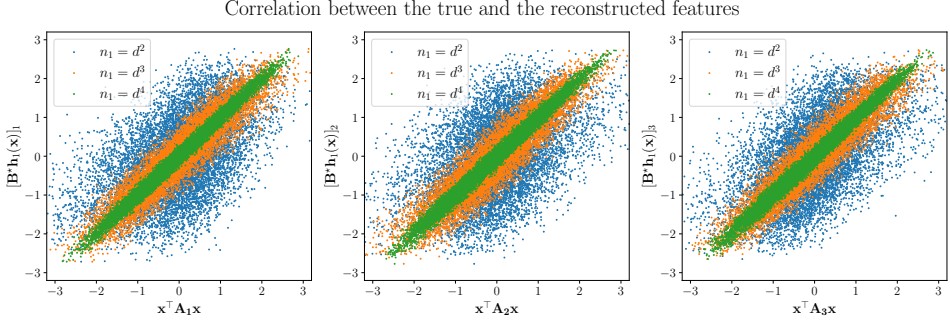

Figure 4: The linear correlation between the three true features and their corresponding reconstructed features for varying first-stage sample sizes $n_1$. The reconstructed features are standardized to match the variance of the true features. For $i = 1, 2, 3$, the $i$-th scatter plot represents $10,000$ test sample points of $([\mathbf{B}^\star \mathbf{h}^{(1)}(\mathbf{x})]_i, \mathbf{x}^\top \mathbf{A}_i \mathbf{x})$ for $n_1 \in \{d^2, d^3, d^4\}$, where $d = 16$.

## B    TECHINICAL BACKGROUND

### B.1    ASYMPTOTIC NOTATION

Throughout the proof we will let $C$ be a fixed but sufficiently large constant.

**Definition 1** (high probability events). Let $\iota = C \log(d n_1 n_2 m_1 m_2)$. We say that an event happens *with high probability* if it happens with probability at least $1 - \text{poly}(d, n_1, n_2, m_1, m_2) e^{-\iota}$.

**Example 1.** If $z \sim N(0, 1)$ then $|z| \leq \sqrt{2\iota}$ with high probability.

Note that high probability events are closed under union bounds over sets of size $\text{poly}(d, n_1, n_2, m_1, m_2)$, such as $\mathcal{D}_1$, $\mathcal{D}_2$ and $\{\mathbf{w}_j\}_{j \in [m_1]}$. We will also assume throughout the paper that $\iota \leq C^{-1} d$.

### B.2    MULTIVARIATE GAUSSIAN APPROXIMATION

In this section, we assume that $\mathbf{X} \sim \mathcal{N}(\mathbf{0}, \mathbf{I}_d)$ and aim to establish an upper bound of Wasserstein distance between the distribution of $\mathbf{p}(\mathbf{X})$ and the standard $r$-dimensional Gaussian distribution, i.e., Lemma 1.

To prove Lemma 1, we introduce Stein's method (Ross, 2011) for multivariate Gaussian approximation. We will use the following additional notations.

- $\mathcal{G}f(\mathbf{x}) := \int_0^\infty \mathbb{E}_{\mathbf{Z} \sim \mathcal{N}(\mathbf{0}, \mathbf{I})} \left[ f\big(e^{-t}\mathbf{x} + \sqrt{1 - e^{-2t}}\mathbf{Z}\big) - f(\mathbf{Z}) \right] \mathrm{d}\,t$ denotes the potential operator of $f$.
- $\mathcal{J}(\mathbf{p}) := [\nabla p_1, \nabla p_2, ..., \nabla p_r]^\top \in \mathbb{R}^{r \times n}$ denotes the Jacobian matrix of $\mathbf{p}$.

Now we state the supporting lemmas to prove Lemma 1.

**Lemma 2** (Corollary 9.12 in van Handel (2016)). *For any probability measure $\mu$ in $\mathbb{R}^r$, we have*

$$W_1(\mu, \mathcal{N}(\mathbf{0}, \mathbf{I}_r)) \leq \sup_{\|\nabla g\| \leq 1, \|\nabla^2 g\| \leq \sqrt{\frac{2}{\pi}}} \mathbb{E}_{\mathbf{Y} \sim \mu} \left[ \Delta g(\mathbf{Y}) - \langle \nabla g(\mathbf{Y}), \mathbf{Y} \rangle \right].$$

**Lemma 3** (Lemma 9.21 in van Handel (2016)). *Suppose $\mathbf{X} = (X_1, X_2, ..., X_d) \sim \mathcal{N}(\mathbf{0}, \mathbf{I}_n)$ is an $d$-dimensional standard Gaussian variable. Then for any functions $g : \mathbb{R}^d \to \mathbb{R}$ and $h : \mathbb{R}^d \to \mathbb{R}$, we have*

$$Cov(g(\mathbf{X}), h(\mathbf{X})) = \mathbb{E}_{\mathbf{X}} \left[ \langle \nabla g(\mathbf{X}), \nabla \mathcal{G}h(\mathbf{X}) \rangle \right]$$

With the lemmas above, we begin our proof of Lemma 1.

*Proof of Lemma 1.* By invoking Lemma 2 with $\mu = \text{Law}(\mathbf{p})$ and $\mathbf{Y} = \mathbf{p}(\mathbf{X})$, for any $g(\mathbf{y}) : \mathbb{R}^r \to \mathbb{R}$ with $\|\nabla g\| \leq 1$ and $\left\| \nabla^2 g \right\| \leq \sqrt{\frac{2}{\pi}}$, we aim to bound

$$\underbrace{\mathbb{E}_{\mathbf{X}} \left[ \Delta g(\mathbf{p}(\mathbf{X})) - \langle \nabla g(\mathbf{p}(\mathbf{X})), \mathbf{p}(\mathbf{X}) \rangle \right]}_{\spadesuit} = \sum_{i=1}^r \mathbb{E}_{\mathbf{X}} \left[ \frac{\partial^2 g}{\partial y_i^2} \bigg|_{\mathbf{y} = \mathbf{p}(\mathbf{X})} - p_i(\mathbf{X}) \frac{\partial g}{\partial y_i} \bigg|_{\mathbf{y} = \mathbf{p}(\mathbf{X})} \right].$$

Since for any $i \in [r]$, $\mathbb{E}\left[ p_i(\mathbf{X}) \right] = 0$, we have

$$\mathbb{E} \left[ p_i(\mathbf{X}) \frac{\partial g}{\partial y_i} \bigg|_{\mathbf{y} = \mathbf{p}(\mathbf{X})} \right] = \text{Cov}\left( p_i(\mathbf{X}), \frac{\partial g}{\partial y_i} \bigg|_{\mathbf{y} = \mathbf{p}(\mathbf{X})} \right)$$

$$= \mathbb{E} \left[ \left\langle \nabla_{\mathbf{x}} \frac{\partial g}{\partial y_i} \bigg|_{\mathbf{y} = \mathbf{p}(\mathbf{X})}, \nabla_{\mathbf{x}} \mathcal{G} p_i(\mathbf{X}) \right\rangle \right]$$

$$= \mathbb{E} \left[ \left\langle \sum_{j=1}^r \frac{\partial^2 g}{\partial y_i \partial y_j} \bigg|_{\mathbf{y} = \mathbf{p}(\mathbf{X})} \nabla_{\mathbf{x}} p_j(\mathbf{X}), \nabla_{\mathbf{x}} \mathcal{G} p_i(\mathbf{X}) \right\rangle \right]$$

$$= \sum_{j=1}^r \mathbb{E} \left[ \frac{\partial^2 g}{\partial y_i \partial y_j} \bigg|_{\mathbf{y} = \mathbf{p}(\mathbf{X})} \langle \nabla_{\mathbf{x}} p_j(\mathbf{X}), \nabla_{\mathbf{x}} \mathcal{G} p_i(\mathbf{X}) \rangle \right],$$

where the second equality follows from Lemma 3 and we obtain the third equality by the chain rule. Thus, we have

$$\spadesuit = \mathbb{E}\left[\langle \nabla^2 g(\mathbf{p}(\mathbf{X})), \mathbf{I}_r - \mathcal{J}(\mathbf{p}(\mathbf{X}))\mathcal{J}(\mathcal{G}\mathbf{p}(\mathbf{X}))^\top \rangle\right]. \tag{7}$$

For a special case, for any $i, j \in [r]$, we take $g(\mathbf{y}) = y_i y_j$ in (7), obtaining that

$$\mathbb{E}\left[\langle \nabla_{\mathbf{x}} p_j(\mathbf{X}), \nabla_{\mathbf{x}} \mathcal{G}p_i(\mathbf{X})\rangle\right] = \begin{cases} \mathbb{E}\left[2p_j(\mathbf{X})p_i(\mathbf{X})\right] = 0, & i \neq j, \\ \mathbb{E}\left[2p_i^2(\mathbf{X})\right] - 1 = 1, & i = j. \end{cases}$$

Thus, $\mathbb{E}\left[\mathbf{I}_r - \mathcal{J}(\mathbf{p}(\mathbf{X}))\mathcal{J}(\mathcal{G}\mathbf{p}(\mathbf{X}))^\top\right] = \mathbf{0}_{r \times r}$. Since $\left\|\nabla^2 g\right\| \leq \sqrt{\frac{2}{\pi}}$, we have $\left|\left[\nabla^2 g\right]_{i,j}\right| \leq \sqrt{\frac{2}{\pi}}$ for any $i, j \in [r]$. We can therefore estimate

$$\begin{aligned}
W_1(\text{Law}(\mathbf{p}(\mathbf{X})), \mathcal{N}(\mathbf{0}, \mathbf{I}_r)) &\leq \sqrt{\frac{2}{\pi}} \sum_{i,j \in [r]} \mathbb{E}\left[|\delta_{i,j} - \langle \nabla_{\mathbf{x}} p_j(\mathbf{X}), \nabla_{\mathbf{x}} \mathcal{G}p_i(\mathbf{X})\rangle|\right] \\
&\leq \sqrt{\frac{2}{\pi}} \sum_{i,j \in [r]} \text{Var}[\langle \nabla_{\mathbf{x}} p_j(\mathbf{X}), \nabla_{\mathbf{x}} \mathcal{G}p_i(\mathbf{X})\rangle]^{1/2} \\
&\leq \sqrt{\frac{2}{\pi}} \sum_{i,j \in [r]} \mathbb{E}\left[\|\nabla_{\mathbf{x}} \langle \nabla_{\mathbf{x}} p_j(\mathbf{X}), \nabla_{\mathbf{x}} \mathcal{G}p_i(\mathbf{X})\rangle\|^2\right]^{1/2},
\end{aligned}$$

where we invoke Poincaré inequality in the last inequality. For any $i, j \in [r]$, we have

$$\begin{aligned}
&\mathbb{E}\left[\|\nabla_{\mathbf{x}} \langle \nabla_{\mathbf{x}} p_j(\mathbf{X}), \nabla_{\mathbf{x}} \mathcal{G}p_i(\mathbf{X})\rangle\|^2\right] \\
&= \mathbb{E}\left[\left\|\nabla_{\mathbf{x}}^2 p_j(\mathbf{X})\nabla\mathcal{G}p_i(\mathbf{X}) + \nabla_{\mathbf{x}} p_j(\mathbf{X})\nabla^2 \mathcal{G}p_i(\mathbf{X})\right\|^2\right] \\
&\leq 2\mathbb{E}\left[\left\|\nabla_{\mathbf{x}}^2 p_j(\mathbf{X})\nabla\mathcal{G}p_i(\mathbf{X})\right\|^2\right] + 2\mathbb{E}\left[\left\|\nabla_{\mathbf{x}} p_j(\mathbf{X})\nabla^2 \mathcal{G}p_i(\mathbf{X})\right\|^2\right] \\
&\leq 2\mathbb{E}\left[\left\|\nabla^2 p_j\right\|^4\right]^{1/2}\mathbb{E}\left[\left\|\nabla\mathcal{G}p_i\right\|^4\right]^{1/2} + 2\mathbb{E}\left[\left\|\nabla p_j\right\|^4\right]^{1/2}\mathbb{E}\left[\left\|\nabla^2 \mathcal{G}p_i\right\|^4\right]^{1/2} \\
&\leq 2\mathbb{E}\left[\left\|\nabla^2 p_j\right\|^4\right]^{1/2}\mathbb{E}\left[\left\|\nabla p_i\right\|^4\right]^{1/2} + 2\mathbb{E}\left[\left\|\nabla p_j\right\|^4\right]^{1/2}\mathbb{E}\left[\left\|\nabla^2 p_i\right\|^4\right]^{1/2}.
\end{aligned}$$

The last inequality follows from the inequality in Page 308 in van Handel (2016). By adding up all the terms along $i$ and $j$, we have

$$\begin{aligned}
&W_1(\text{Law}(\mathbf{p}(\mathbf{X})), \mathcal{N}(\mathbf{0}, \mathbf{I}_r)) \\
&\leq \sqrt{\frac{2}{\pi}} \sum_{i,j \in [r]} \sqrt{2\mathbb{E}\left[\left\|\nabla^2 p_j\right\|^4\right]^{1/2}\mathbb{E}\left[\left\|\nabla p_i\right\|^4\right]^{1/2} + 2\mathbb{E}\left[\left\|\nabla p_j\right\|^4\right]^{1/2}\mathbb{E}\left[\left\|\nabla^2 p_i\right\|^4\right]^{1/2}} \\
&\leq \frac{2}{\sqrt{\pi}} \sum_{i,j \in [r]} \left(\mathbb{E}\left[\left\|\nabla^2 p_j\right\|^4\right]^{1/4}\mathbb{E}\left[\left\|\nabla p_i\right\|^4\right]^{1/4} + \mathbb{E}\left[\left\|\nabla p_j\right\|^4\right]^{1/4}\mathbb{E}\left[\left\|\nabla^2 p_i\right\|^4\right]^{1/4}\right) \\
&= \frac{4}{\sqrt{\pi}} \left(\sum_{i=1}^r \mathbb{E}\left[\left\|\nabla p_i\right\|^4\right]^{1/4}\right)\left(\sum_{j=1}^r \mathbb{E}\left[\left\|\nabla^2 p_j\right\|^4\right]^{1/4}\right).
\end{aligned}$$

We complete our proof. $\qquad\qquad\square$

### B.3 HYPERCONTRACTIVITY OF POLYNOMIALS

The following Lemma is cited from Mei et al. (2021) and is designed for uniform distribution on the sphere in $d$ dimension.

**Lemma 4.** *For any $\ell \in \mathbb{N}$ and $f \in L^2(\mathbb{S}^{d-1})$ to be a degree $\ell$ polynomial, for any $q \geqslant 2$, we have*

$$\left(\mathbb{E}_{\mathbf{z} \sim \text{Unif}(\mathbb{S}^{d-1}(\sqrt{d}))}\left[f(\mathbf{z})^q\right]\right)^{2/q} \leq (q-1)^\ell \mathbb{E}_{\mathbf{z} \sim \text{Unif}(\mathbb{S}^{d-1}(\sqrt{d}))}\left[f(\mathbf{z})^2\right].$$

We remark that the results above are also multiplicative.

**Lemma 5.** *For any $\ell \in \mathbb{N}$ and $f \in L^2((\mathbb{S}^{d-1})^k)$ to be a degree $\ell$ polynomial in the components of each $\mathbf{z}_1, \mathbf{z}_2, \ldots, \mathbf{z}_k$, for any $q \geqslant 2$, we have*

$$\left( \mathbb{E}_{\mathbf{z} \sim \mathrm{Unif}\left( \mathbb{S}^{d-1}(\sqrt{d}) \right)^k} \left[ f(\mathbf{z})^q \right] \right)^{2/q} \leq (q-1)^{k\ell} \, \mathbb{E}_{\mathbf{z} \sim \mathrm{Unif}\left( \mathbb{S}^{d-1}(\sqrt{d}) \right)^k} \left[ f(\mathbf{z})^2 \right].$$

*Here $\mathbf{z} = \mathrm{Vec}\left( [\mathbf{z}_1, \mathbf{z}_2, \ldots, \mathbf{z}_k] \right)$.*

For the case where the input distribution is standard Gaussian in $d$ dimension (denoted as $\gamma$), we have the next Lemma from Theorem 4.3, Prato and Tubaro (2007).

**Lemma 6.** *For any $\ell \in \mathbb{N}$ and $f \in L^2(\gamma)$ to be a degree $\ell$ polynomial, for any $q \geqslant 2$, we have*

$$\mathbb{E}_{\mathbf{z} \sim \gamma} \left[ f(\mathbf{z})^q \right] \leq \mathcal{O}_{q,\ell}(1) \left( \mathbb{E}_{\mathbf{z} \sim \gamma} \left[ f(\mathbf{z})^2 \right] \right)^{q/2}.$$

*where we use $\mathcal{O}_{q,\ell}(1)$ to denote some universal constant that only depends on $q, \ell$.*

Moreover, we introduce lemmas to control the deviation of random variables which polynomially depend on some Gaussian random variables. We will use a slightly modified version of Lemma 30 from Damian et al. (2022).

**Lemma 7.** *Let $g$ be a polynomial of degree $p$ and $\mathbf{x} \sim \mathcal{N}(\mathbf{0}, \mathbf{I}_d)$. Then there exists an absolute positive constant $C_p$ depending only on $p$ such that for any $\delta > 1$,*

$$\mathbb{P}\left[ |g(\mathbf{x}) - \mathbb{E}[g(\mathbf{x})]| \geqslant \delta \sqrt{\mathrm{Var}(g(\mathbf{x}))} \right] \leq 2 \exp\left( -C_p \delta^{2/p} \right).$$

We also have the spherical version of Lemma 7.

**Lemma 8.** *Let $g$ be a polynomial of degree $p$ and $\mathbf{x} \sim \mathbb{S}^{d-1}(\sqrt{d})$. Then there exists an absolute positive constant $C_p$ depending only on $p$ such that for any $\delta > 1$,*

$$\mathbb{P}\left[ |g(\mathbf{x}) - \mathbb{E}[g(\mathbf{x})]| \geqslant \delta \sqrt{\mathrm{Var}(g(\mathbf{x}))} \right] \leq 2 \exp\left( -C_p \delta^{2/p} \right).$$

Thus, for a degree-p polynomial $g$, we have $g(\mathbf{x}) \lesssim \iota^{p/2} \|g\|_{L^2}$ with high probability.

## B.4 Moments and Factorization of Polynomials

In this section, we present formulae for calculating moments of Gaussian or spherical variables, cited from Damian et al. (2022).

**Lemma 9** (Expectations of Gaussian tensors). *For $\mathbf{w} \in \mathcal{N}(\mathbf{0}_d, \mathbf{I}_d)$ and $k \in \mathbb{N}$, we have*

$$\mathbb{E}_{\mathbf{w}} \left[ \mathbf{w}^{\otimes 2k} \right] = (2k-1)!! \mathrm{Sym}(\mathbf{I}_d^{\otimes k})$$

*Here $\mathrm{Sym}(\mathbf{T})$ is the symmetrization of a $k$-tensor $\mathbf{T} \in (\mathbb{R}^d)^{\otimes k}$ across all $k$ axes.*

Leveraging this calculation, we can factorize any polynomial $g$ into inner products between high-order tensors and bound the Frobenius norm of the tensors.

**Lemma 10.** *(Lemma 21 in Damian et al. (2022)) Given Let $g : \mathbb{R}^r \to \mathbb{R}$ be an degree-$p$ polynomial. Then there exists $\mathbf{T}_0, \mathbf{T}_1, \ldots, \mathbf{T}_p$ such that*

$$g(\mathbf{z}) = \sum_{k=0}^{p} \left\langle \mathbf{T}_k, \mathbf{z}^{\otimes k} \right\rangle \quad with \quad \|\mathbf{T}_k\|_{\mathrm{F}} \lesssim \|g\|_{L^2} \, r^{\frac{p-k}{4}}, \quad k = 0, 1, \ldots, p.$$

*Here $\|g\|_{L^2} = \mathbb{E}_{\mathbf{z} \sim \mathcal{N}(\mathbf{0}, \mathbf{I})} \left[ g^2(\mathbf{z}) \right].$*

As a corollary, we then have $\nabla g(\mathbf{z}) = \sum_{k \geqslant 1}^{p} k \mathbf{T}_k(z^{\otimes k-1})$ and

$$\|\nabla g(\mathbf{z})\| \leq \sum_{k=1}^{p} k \|\mathbf{T}_k\|_{\mathrm{F}} \|\mathbf{z}\|^{k-1} \lesssim \|g\|_{L^2} \sum_{k=1}^{p} k r^{\frac{p-k}{4}} \|\mathbf{z}\|^{k-1}. \tag{8}$$

For a spherical variable $\mathbf{x} \sim \mathrm{Unif}(\mathbb{S}^{d-1}(\sqrt{d}))$ we can also compute its moments.

**Lemma 11** (Expectations of Spherical tensors). *For* $\mathbf{x} \sim \mathrm{Unif}(\mathbb{S}^{d-1}(\sqrt{d}))$ *and* $k \in \mathbb{N}$*, we have*

$$\mathbb{E}_{\mathbf{z}}\left[\mathbf{z}^{\otimes 2k}\right] = d^k \cdot \frac{\mathbb{E}_{\mathbf{w} \sim \mathcal{N}(\mathbf{0}_d, \mathbf{I}_d)}\left[\mathbf{w}^{\otimes 2k}\right]}{\mathbb{E}_{v \sim \chi(d)}\left[v^{2k}\right]},$$

*where* $\chi(d)$ *represents the chi-distribution with the degree of freedom being* $d$*, and its moments can be computed as*

$$\mathbb{E}_{v \sim \chi(d)}\left[v^{2k}\right] = \prod_{j=0}^{k-1}(d+2j) = \Theta(d^k).$$

As an example, the moments of spherical quadratic forms $\mathbf{x}^\top \mathbf{A} \mathbf{x}$ can be computed explicitly as

$$\mathbb{E}_{\mathbf{x}}\left[\mathbf{x}^\top \mathbf{A} \mathbf{x}\right] = \mathrm{tr}(\mathbf{A}), \quad \text{and} \quad \mathbb{E}_{\mathbf{x}}\left[(\mathbf{x}^\top \mathbf{A} \mathbf{x})(\mathbf{x}^\top \mathbf{B} \mathbf{x})\right] = \frac{d}{d+2} \cdot (\mathrm{tr}(\mathbf{A})\mathrm{tr}(\mathbf{B}) + 2\langle \mathbf{A}, \mathbf{B}\rangle).$$

Thus, to satisfy Assumption 1, we require $\mathrm{tr}(\mathbf{A}_k) = 0$, $\|\mathbf{A}_k\|_{\mathrm{F}} = \sqrt{(d+2)/(2d)}$ and $\langle \mathbf{A}_k, \mathbf{A}_\ell\rangle = 0$ for any $k, \ell \in [r]$.

## B.5 SPHERICAL HARMONICS AND GEGENBAUER POLYNOMIALS

We introduce some facts of spherical harmonics and Gegenbauer polynomials, with the first four properties from Ghorbani et al. (2021) and the last one from Koornwinder (2018).

1. For $\mathbf{x}, \mathbf{y} \in \mathbb{S}^{d-1}(\sqrt{d})$,

$$|Q_j(\langle \mathbf{x}, \mathbf{y}\rangle)| \leq Q_j(d) = 1. \tag{9}$$

2. For $\mathbf{x}, \mathbf{y} \in \mathbb{S}^{d-1}(\sqrt{d})$,

$$\langle Q_j(\langle \mathbf{x}, \cdot\rangle), Q_k(\langle \mathbf{y}, \cdot\rangle)\rangle_{L^2} = \frac{1}{B(d,k)}\delta_{jk}Q_k(\langle \mathbf{x}, \mathbf{y}\rangle). \tag{10}$$

Here $B(d, k)$ denotes the dimension of subspace of degree $k$ spherical harmonics

$$B(d,k) := \dim(V_{d,k}) = \frac{2k+d-2}{k}\binom{k+d-3}{k-1} = \Theta(d^k).$$

3. For $\mathbf{x}, \mathbf{y} \in \mathbb{S}^{d-1}(\sqrt{d})$,

$$Q_k(\langle \mathbf{x}, \mathbf{y}\rangle) = \frac{1}{B(d,k)}\sum_{i=1}^{B(d,k)} Y_{k,i}(\mathbf{x})Y_{k,i}(\mathbf{y}). \tag{11}$$

4. For any $k \in \mathbb{N}_{\geqslant 1}$,

$$\frac{t}{d}Q_k(t) = \frac{k}{2k+d-2}Q_{k-1}(t) + \frac{k+d-2}{2k+d-2}Q_{k+1}(t). \tag{12}$$

5. For any $i, j \in \mathbb{N}$,

$$Q_i(t)Q_j(t) = \sum_{k=0}^{\min(i,j)} b_{i+j-2k}^{(i,j)}\binom{i}{k}\binom{j}{k}k!Q_{i+j-2k}(t). \tag{13}$$

Here, we have

$$b_{i+j-2k}^{(i,j)} = \frac{2(i+j-2k)+d-2}{d-2} \cdot \frac{((d-2)/2)_k((d-2)/2)_{i-k}((d-2)/2)_{j-k}(d-2)_{i+j-k}}{(d-2)_i(d-2)_j(d/2)_{i+j-k}}.$$

We note that $(z)_k = z(z+1)\cdots(z+k-1) = \Gamma(z+k)/\Gamma(z)$ is the Pochhammer symbol. Given any $i$ and $j$, we have $d^k b_{i+j-2k}^{(i,j)} \to 1$ when $d \to \infty$. We derive a quantitative bound on the scale of $b_{i+j-2k}^{(i,j)}$ in Lemma 12.

**Lemma 12.** *For any $i, j \geqslant k \geqslant 0$, denote*

$$c_{i+j-2k}^{(i,j)} = \frac{((d-2)/2)_k((d-2)/2)_{i-k}((d-2)/2)_{j-k}(d-2)_{i+j-k}}{(d-2)_i(d-2)_j(d/2)_{i+j-k}}.$$

*Then, when $d \geqslant 4$, it holds that $c_{i+j-2k}^{(i,j)} \leq \frac{1}{(d-2)_k}$.*

*Proof of Lemma 12.* Note that when $d \geqslant 4$, i.e., $d - 2 \geqslant d/2$,

$$\frac{c_{(i+1)+j-2k}^{(i+1,j)}}{c_{i+j-2k}^{(i,j)}} = \frac{\left(\frac{d-2}{2} + i - k\right)(d - 2 + i + j - k)}{(d - 2 + i)\left(\frac{d}{2} + i + j - k\right)} \quad \text{(monotone decreasing with } j)$$

$$\leq \frac{\left(\frac{d-2}{2} + i - k\right)(d - 2 + i)}{(d - 2 + i)\left(\frac{d}{2} + i\right)}$$

$$< 1.$$

Thus, we have $c_{(i+1)+j-2k}^{(i+1,j)} \leq c_{i+j-2k}^{(i,j)}$. Similarly, we have $c_{i+(j+1)-2k}^{(i,j+1)} \leq c_{i+j-2k}^{(i,j)}$. Consequently, for any $i, j \geqslant k$, we have

$$c_{i+j-2k}^{(i,j)} \leq c_{k+k-2k}^{(k,k)}$$

$$= \frac{((d-2)/2)_k((d-2)/2)_0((d-2)/2)_0(d-2)_k}{(d-2)_k(d-2)_k(d/2)_k}$$

$$= \frac{((d-2)/2)_k}{(d-2)_k(d/2)_k}$$

$$= \frac{(d-2)/2}{(d-2)_k(d/2 + k - 1)}$$

$$\leq \frac{1}{(d-2)_k}.$$

The proof is complete. $\qquad\square$

## C  APPROXIMATION THEORY OF THE INNER LAYER

Since we focus on the first training stage throughout this section, we denote $n = n_1$ for notation simplicity when the context is clear, and let the training set be $\mathcal{D}_1 = \{\mathbf{x}_1, \mathbf{x}_2, \ldots, \mathbf{x}_n\}$.

### C.1  ASYMPTOTIC ANALYSIS OF THE LEARNED FEATURE

In this subsection, we analyse the learned feature $\mathbf{h}^{(1)}(\mathbf{x}')$ in the asymptotic way, i.e., $m_2, n \to \infty$. Note that we can rewrite the learned feature as

$$\mathbf{h}^{(1)}(\mathbf{x}') = \frac{1}{nm_2} \sum_{i=1}^{n} f^{\star}(\mathbf{x}_i)\langle \mathbf{h}^{(0)}(\mathbf{x}_i), \mathbf{h}^{(0)}(\mathbf{x}')\rangle \mathbf{h}^{(0)}(\mathbf{x}_i)$$

$$= \frac{1}{n} \sum_{i=1}^{n} f^{\star}(\mathbf{x}_i) K_{m_2}^{(0)}(\mathbf{x}, \mathbf{x}') \mathbf{h}^{(0)}(\mathbf{x}_i).$$

where the initial kernel $K_{m_2}^{(0)}(\mathbf{x}, \mathbf{x}')$ is defined as

$$K_{m_2}^{(0)}(\mathbf{x}, \mathbf{x}') = \frac{1}{m_2}\langle \sigma_2(\mathbf{V}\mathbf{x}), \sigma_2(\mathbf{V}\mathbf{x}')\rangle \approx \mathbb{E}_{\mathbf{v}}\left[\sigma_2(\mathbf{v}^{\top}\mathbf{x})\sigma_2(\mathbf{v}^{\top}\mathbf{x}')\right].$$

In this case, we have for any $j \in [m_2]$,

$$[\mathbf{h}^{(1)}(\mathbf{x}')]_j = \frac{1}{n} \sum_{i=1}^{n} K_{m_2}^{(0)}(\mathbf{x}_i, \mathbf{x}')\sigma_2(\mathbf{v}_j^{\top}\mathbf{x}_i) \overset{m_2, n \to \infty}{\to} \mathbb{E}_{\mathbf{x}}\left[f^{\star}(\mathbf{x})K^{(0)}(\mathbf{x}, \mathbf{x}')\sigma_2(\mathbf{v}_j^{\top}\mathbf{x})\right].$$

Here the infinite-inner-width kernel $K^{(0)}$ is defined as

$$K^{(0)}(\mathbf{x}, \mathbf{x}') := \mathbb{E}_{\mathbf{v}}\left[\sigma_2(\mathbf{v}^{\top}\mathbf{x})\sigma_2(\mathbf{v}^{\top}\mathbf{x}')\right] = \sum_{i=2}^{\infty} \frac{c_i^2}{B(d, i)} Q_i(\mathbf{x}^{\top}\mathbf{x}').$$

Recall that $Q_2(t) = \frac{t^2-d}{d(d-1)}$, so $Q_2(\mathbf{v}_j^\top \mathbf{x}) = \langle \mathbf{x}\mathbf{x}^\top - \mathbf{I}, \mathbf{v}_j \mathbf{v}_j^\top - \mathbf{I}\rangle/(d(d-1))$. Let's focus on the contribution of the quadratic term $Q_2$ in $K^{(0)}(\mathbf{x}, \mathbf{x}')$ and $\sigma_2(\mathbf{v}_j^\top \mathbf{x})$, which is

$$\frac{c_2^3}{B(d,2)} \cdot \mathbb{E}_\mathbf{x} \left[ f^\star(\mathbf{x}) Q_2(\mathbf{x}^\top \mathbf{x}') Q_2(\mathbf{v}_j^\top \mathbf{x}) \right]$$

$$= \frac{c_2^3}{B(d,2)d^2(d-1)^2} \cdot \mathbb{E}_\mathbf{x} \left[ f^\star(\mathbf{x}) \langle \mathbf{x}\mathbf{x}^\top - \mathbf{I}, \mathbf{v}_j \mathbf{v}_j^\top - \mathbf{I}\rangle \langle \mathbf{x}\mathbf{x}^\top - \mathbf{I}, \mathbf{x}'\mathbf{x}'^\top - \mathbf{I}\rangle \right]$$

$$\approx \frac{1}{d^6} \left\langle \mathbb{E}_\mathbf{x} \left[ f^\star(\mathbf{x})(\mathbf{x}\mathbf{x}^\top - \mathbf{I})^{\otimes 2} \right], (\mathbf{v}_j \mathbf{v}_j^\top - \mathbf{I}) \otimes (\mathbf{x}'\mathbf{x}'^\top - \mathbf{I}) \right\rangle.$$

The following proposition provides an approximation of the tensor $\mathbb{E}_\mathbf{x} \left[ f^\star(\mathbf{x})(\mathbf{x}\mathbf{x}^\top - \mathbf{I})^{\otimes 2} \right]$, which lays the foundation of our feature reconstruction theory.

**Proposition 3.** *Consider two linear operators $T$ and $T^\star$ that map $\mathbb{R}^{d\times d}$ to $\mathbb{R}^{d\times d}$ and satisfy*

$$T(\mathbf{W}) = \mathbb{E}_\mathbf{x} \left[ f^\star(\mathbf{x}) \langle \mathbf{W}, \mathbf{x}\mathbf{x}^\top - \mathbf{I}\rangle (\mathbf{x}\mathbf{x}^\top - \mathbf{I}) \right], \quad and \tag{14}$$

$$T^\star(\mathbf{W}) = \sum_{k=1}^r \frac{1}{\|\mathbf{A}_k\|_F^2} \langle \mathbf{W}, \mathbf{A}_k\rangle \sum_{j=1}^r \mathbf{H}_{k,j} \mathbf{A}_j \tag{15}$$

*for any $\mathbf{W} \in \mathbb{R}^{d\times d}$, where $\mathbf{H}$ is the expected Hessian matrix $\mathbf{H} = \mathbb{E}_{\mathbf{z}\sim\mathcal{N}(\mathbf{0}_r,\mathbf{I}_r)} \left[ \nabla^2 g^\star(\mathbf{z}) \right]$. Then for any $\mathbf{W} \in \mathbb{R}^{d\times d}$, we have*

$$\|T(\mathbf{W}) - T^\star(\mathbf{W})\|_F \lesssim d^{-1/6} L r^2 \kappa_1 \log^2 d \cdot \|\mathbf{W}\|_F.$$

*Here $L = \widetilde{\mathcal{O}}(r^{\frac{p-1}{2}})$ is the Lipschitz constant of $g^\star$ that holds with high probability.*

The proof is provided in Appendix C.1.1. This proposition shows that, when $\mathbf{H}$ is well-conditioned and $d \gg r$, $T$ can fully recover the space spanned by $\mathbf{A}_1, \mathbf{A}_2, \ldots, \mathbf{A}_r$, which enables us to reconstruct the features efficiently. Specifically, when taking $\mathbf{W}_k = \sum_{j=1}^r [\mathbf{H}^{-1}]_{k,j} \mathbf{A}_j$ for any $k \in [r]$, we have $T(\mathbf{W}_k) \approx T^\star(\mathbf{W}_k) = \mathbf{A}_k$.

Now we consider the construction of $\mathbf{B}^\star$. If we set $\mathbf{B}^\star = \frac{1}{m_2}[\mathbf{p}_0(\mathbf{v}_1), \ldots, \mathbf{p}_0(\mathbf{v}_{m_2})]$ for some vector-valued function $\mathbf{p}_0 : \mathbb{R}^d \to \mathbb{R}^r$ and denote $\mathbf{p}_0(\mathbf{v}) = [p_{0,1}(\mathbf{v}), \ldots, p_{0,r}(\mathbf{v})]^\top$, we directly have for any $k \in [r]$,

$$[\mathbf{B}^\star \mathbf{h}^{(1)}(\mathbf{x}')]_k \approx \frac{1}{m_2} \sum_{j=1}^{m_2} \mathbb{E}_\mathbf{x} \left[ f^\star(\mathbf{x}) K^{(0)}(\mathbf{x}, \mathbf{x}') \sigma_2(\mathbf{v}_j^\top \mathbf{x}) p_{0,k}(\mathbf{v}_j) \right]$$

$$\approx \frac{1}{d^6} \mathbb{E}_\mathbf{v} \left[ \langle \mathbb{E}_\mathbf{x} \left[ f^\star(\mathbf{x})(\mathbf{x}\mathbf{x}^\top - \mathbf{I})^{\otimes 2} \right], p_{0,k}(\mathbf{v})(\mathbf{v}\mathbf{v}^\top - \mathbf{I}) \otimes (\mathbf{x}'\mathbf{x}'^\top - \mathbf{I}) \rangle \right].$$

$$\approx \frac{1}{d^6} \left\langle T^\star \left( \mathbb{E}_\mathbf{v} \left[ p_{0,k}(\mathbf{v})(\mathbf{v}\mathbf{v}^\top - \mathbf{I}) \right] \right), \mathbf{x}'\mathbf{x}'^\top - \mathbf{I} \right\rangle.$$

Thus, it suffices to solve

$$T^\star \left( \mathbb{E}_\mathbf{v} \left[ p_{0,k}(\mathbf{v})(\mathbf{v}\mathbf{v}^\top - \mathbf{I}) \right] \right) \propto \mathbf{A}_k, \quad k = 1, 2, \ldots, r,$$

which is equivalent to solving

$$\mathbb{E}_\mathbf{v} \left[ p_{0,k}(\mathbf{v})(\mathbf{v}\mathbf{v}^\top - \mathbf{I}) \right] \propto \mathbf{W}_k = \sum_{j=1}^r [\mathbf{H}^{-1}]_{k,j} \mathbf{A}_j.$$

Since we have $\mathbb{E}_\mathbf{v} \left[ (\mathbf{v}^\top \mathbf{A}_k \mathbf{v})(\mathbf{v}\mathbf{v}^\top - \mathbf{I}) \right] \propto \mathbf{A}_k$, we can explicitly construct $p_{0,k}(\mathbf{v})$ as

$$p_{0,k}(\mathbf{v}) \propto \sum_{j=1}^r [\mathbf{H}^{-1}]_{k,j} \mathbf{v}^\top \mathbf{A}_j \mathbf{v}, \quad \text{i.e.,} \quad \mathbf{p}_0(\mathbf{v}) \propto \mathbf{H}^{-1} \mathbf{p}(\mathbf{v}).$$

Thus, with a well conditioned $\mathbf{H}$, we can fully reconstruct the features.

### C.1.1 PROOF OF PROPOSITION 3

To prove Proposition 3, it suffices to prove that the approximation error

$$R(\mathbf{W}, \mathbf{V}) = \langle T(\mathbf{W}) - T^\star(\mathbf{W}), \mathbf{V} \rangle \lesssim d^{-1/6} L r^2 \kappa_1 \log^2 d$$

holds for any test matrix $\mathbf{V}$ with $\|\mathbf{V}\|_F = 1$. We rely on the following three lemmas.

**Lemma 13** (Bound $R(\mathbf{A}_i, \mathbf{A}_j)$). *For any $i, j \in [r]$, we have*

$$\left| \mathbb{E}_{\mathbf{x}} \left[ g^\star(\mathbf{x}^\top \mathbf{A}_1 \mathbf{x}, \ldots, \mathbf{x}^\top \mathbf{A}_r \mathbf{x}) \langle \mathbf{A}_i, \mathbf{x}\mathbf{x}^\top - \mathbf{I} \rangle \langle \mathbf{A}_j, \mathbf{x}\mathbf{x}^\top - \mathbf{I} \rangle \right] - \mathbb{E}_{\mathbf{z} \sim \mathcal{N}(\mathbf{0}_r, \mathbf{I}_r)} \left[ \nabla^2 g(\mathbf{z}) \right]_{i,j} \right| \leq \frac{L r^2 \kappa_1 \log^2 d}{\sqrt{d}}.$$

*Here $L = C_g R r^{\frac{p-1}{2}}$ is the Lipschitz constant of $g^\star$ that holds with high probability.*

Following the proof above, we have the following more general lemma.

**Lemma 14** (Bound $R(\mathbf{A}_i, \mathbf{B})$). *For any matrix $\mathbf{B} \in \mathbb{R}^{d \times d}$ satisfying $\mathbb{E}\left[\mathbf{x}^\top \mathbf{B}\mathbf{x}\right] = 0$, $\mathbb{E}\left[\left(\mathbf{x}^\top \mathbf{B}\mathbf{x}\right)^2\right] = 1$ and $\langle \mathbf{B}, \mathbf{A}_i \rangle = 0$ for any $i = 1, 2, \ldots, r$, we have*

$$\left| \mathbb{E}\left[ g^\star(\mathbf{x}^\top \mathbf{A}_1 \mathbf{x}, \ldots, \mathbf{x}^\top \mathbf{A}_r \mathbf{x}) \langle \mathbf{A}_i, \mathbf{x}\mathbf{x}^\top - \mathbf{I} \rangle \langle \mathbf{B}, \mathbf{x}\mathbf{x}^\top - \mathbf{I} \rangle \right] \right| \lesssim d^{-1/4} L r^2 \kappa_1 \log^2 d.$$

**Lemma 15** (Bound $R(\mathbf{B}_1, \mathbf{B}_2)$). *For any two matrices $\mathbf{B}_1, \mathbf{B}_2 \in \mathbb{R}^{d \times d}$ satisfying $\mathbb{E}\left[\mathbf{x}^\top \mathbf{B}_j \mathbf{x}\right] = 0$, $\mathbb{E}\left[\left(\mathbf{x}^\top \mathbf{B}_j \mathbf{x}\right)^2\right] = 1$ and $\langle \mathbf{B}_j, \mathbf{A}_i \rangle = 0$ for any $j = 1, 2$ and $i = 1, 2, \ldots, r$, we have*

$$\left| \mathbb{E}\left[ g^\star(\mathbf{x}^\top \mathbf{A}_1 \mathbf{x}, \ldots, \mathbf{x}^\top \mathbf{A}_r \mathbf{x}) \langle \mathbf{B}_1, \mathbf{x}\mathbf{x}^\top - \mathbf{I} \rangle \langle \mathbf{B}_2, \mathbf{x}\mathbf{x}^\top - \mathbf{I} \rangle \right] \right| \lesssim d^{-1/6} L r^2 \kappa_1 \log^2 d.$$

*Here $L \lesssim \iota r^{\frac{p-1}{2}}$ is a Lipschitz constant satisfying $\|\nabla g^\star(\mathbf{p}(\mathbf{x}))\|_2 \leq L$ with high probability.*

The proof of the three lemmas is provided in Appendix C.1.2. With the lemmas above, we begin our proof of Proposition 3.

*Proof of Proposition 3.* Given any $\mathbf{W} \in \mathbb{R}^{d \times d}$, we assume $\|\mathbf{W}\|_F = 1$ without loss of generality. Let's decompose $\mathbf{W}$ as

$$\mathbf{W} = \sum_{k=1}^r \lambda_k \mathbf{A}_k + \lambda_{r+1} \frac{\mathbf{I}_d}{\sqrt{d}} + \lambda_{r+2} \mathbf{B}, \quad \text{where } \langle \mathbf{B}, \mathbf{A}_k \rangle = \langle \mathbf{B}, \mathbf{I}_d \rangle = 0, \ k \in [r].$$

Here the coefficients $\{\lambda_k\}_{k=1}^{r+2}$ satisfy $\sum_{k=1}^{r+2} \lambda_k^2 \lesssim 1$, so $\sum_{k=1}^{r+2} |\lambda_k| \lesssim \sqrt{r}$. Since $T^\star(\mathbf{B}) = T(\mathbf{I}) = T^\star(\mathbf{I}) = \mathbf{0}_{d \times d}$, we have

$$\begin{aligned}
\|T(\mathbf{W}) - T^\star(\mathbf{W})\|_F &\leq \sum_{k=1}^r |\lambda_k| \|T(\mathbf{A}_k) - T^\star(\mathbf{A}_k)\|_F \\
&\quad + |\lambda_{r+1}| \left\| T\left(\frac{\mathbf{I}_d}{\sqrt{d}}\right) - T^\star\left(\frac{\mathbf{I}_d}{\sqrt{d}}\right) \right\|_F + |\lambda_{r+2}| \|T(\mathbf{B}) - T^\star(\mathbf{B})\|_F \\
&= \sum_{k=1}^r |\lambda_k| \|T(\mathbf{A}_k) - T^\star(\mathbf{A}_k)\|_F + |\lambda_{r+2}| \|T(\mathbf{B})\|_F.
\end{aligned}$$

Since both $T(\mathbf{A}_k)$ and $T^\star(\mathbf{A}_k)$ are traceless, by Lemma 13 and 14, we have for any $k = 1, 2, \cdots, r$,

$$\begin{aligned}
\|T(\mathbf{A}_k) - T^\star(\mathbf{A}_k)\|_F &= \max_{\|\mathbf{V}\|=1, \text{tr}(\mathbf{V})=0} \langle T(\mathbf{A}_k) - T^\star(\mathbf{A}_k), \mathbf{V} \rangle \\
&\lesssim \sqrt{r} \cdot d^{-1/2} L r^2 \kappa_1 \log^2 d + d^{-1/4} L r^2 \kappa_1 \log^2 d \\
&\lesssim d^{-1/4} L r^2 \kappa_1 \log^2 d.
\end{aligned}$$

This is because we can decompose $\mathbf{V} = \sum_{k=1}^r c_k \mathbf{A}_k + c_{r+1} \mathbf{B}'$ with $\langle \mathbf{B}', \mathbf{A}_k \rangle = 0$ and apply the two lemmas to obtain the results above. Similarly, by Lemma 14 and 15, we have

$$\begin{aligned}
\|T(\mathbf{B})\|_F &= \max_{\|\mathbf{V}\|=1, \text{tr}(\mathbf{V})=0} \langle T(\mathbf{B}), \mathbf{V} \rangle \\
&\lesssim \sqrt{r} \cdot d^{-1/4} L r^2 \kappa_1 \log^2 d + d^{-1/6} L r^2 \kappa_1 \log^2 d \\
&\lesssim d^{-1/6} L r^2 \kappa_1 \log^2 d.
\end{aligned}$$

Thus, we have

$$\|T(\mathbf{W}) - T^\star(\mathbf{W})\|_F \lesssim \sum_{k=1}^{r+2} |\lambda_k|\, d^{-1/4} L r^2 \log^2 d + |\lambda_{r+2}|\, d^{-1/6} L r^2 \kappa_1 \log^2 d$$
$$\lesssim d^{-1/6} L r^2 \kappa_1 \log^2 d.$$

Here we invoke $\sum_{k=1}^{r+2} |\lambda_k| \lesssim \sqrt{r} = o_d(1)$ in the last inequality. The proof is complete. $\square$

### C.1.2 OMITTED PROOFS IN APPENDIX C.1.1

The following lemmas lay the foundation for our approximation process.

**Lemma 16.** *Suppose Assumption 1 holds. Then the Wasserstein-1 distance between the distribution of $(\mathbf{x}^\top \mathbf{A}_1 \mathbf{x}, \dots, \mathbf{x}^\top \mathbf{A}_r \mathbf{x})$ and standard Gaussian $\mathcal{N}(\mathbf{0}_r, \mathbf{I}_r)$ can be bounded by*

$$W_1\big((\mathbf{x}^\top \mathbf{A}_1 \mathbf{x}, \dots, \mathbf{x}^\top \mathbf{A}_r \mathbf{x}), \mathcal{N}(\mathbf{0}_r, \mathbf{I}_r)\big) \lesssim \frac{r^2 \kappa_1}{\sqrt{d}}. \tag{16}$$

*Moreover, for any orthogonal unit vectors $\mathbf{u}_1, \mathbf{u}_2, \dots \mathbf{u}_s \in \mathbb{R}^d$, we have a similar bound of*

$$W_1\big((\mathbf{x}^\top \mathbf{A}_1 \mathbf{x}, \cdots, \mathbf{x}^\top \mathbf{A}_r \mathbf{x}, \mathbf{u}_1^\top \mathbf{x}, \mathbf{u}_2^\top \mathbf{x}, \cdots, \mathbf{u}_s^\top \mathbf{x}), \mathcal{N}(\mathbf{0}_{r+s}, \mathbf{I}_{r+s})\big) \lesssim \frac{(r+s)^2 \kappa_1}{\sqrt{d}}. \tag{17}$$

*Proof of Lemma 16.* For a fixed matrix $\mathbf{A}_i$, define the function $f_i(\mathbf{z}) = d \frac{\mathbf{z}^\top \mathbf{A}_i \mathbf{z}}{\|\mathbf{z}\|^2}$ and let $\mathbf{x} = \frac{\mathbf{z}\sqrt{d}}{\|\mathbf{z}\|}$. Observe that when $\mathbf{z} \sim \mathcal{N}(0, \mathbf{I})$, we have $\mathbf{x} \sim \mathrm{Unif}(\mathcal{S}^{d-1}(\sqrt{d}))$. Therefore $[f_i(\mathbf{z})]_{i \in [r]}$ is equal in distribution to $[\mathbf{x}^\top \mathbf{A}\mathbf{x}]_{i \in [r]}$. We have for any $i \in [r]$,

$$\nabla f_i(\mathbf{z}) = 2d \left( \frac{\mathbf{A}_i \mathbf{z}}{\|\mathbf{z}\|^2} - \frac{\mathbf{z}^\top \mathbf{A}_i \mathbf{z} \cdot \mathbf{z}}{\|\mathbf{z}\|^4} \right)$$

and

$$\nabla^2 f_i(\mathbf{z}) = 2d \left( \frac{\mathbf{A}_i}{\|\mathbf{z}\|^2} - \frac{2\mathbf{A}_i \mathbf{z}\mathbf{z}^\top}{\|\mathbf{z}\|^4} - \frac{2\mathbf{z}\mathbf{z}^\top \mathbf{A}_i}{\|\mathbf{z}\|^4} - 2\frac{\mathbf{z}^\top \mathbf{A}_i \mathbf{z}}{\|\mathbf{z}\|^4}\mathbf{I} + 4\frac{\mathbf{z}^\top \mathbf{A}_i \mathbf{z}\mathbf{z}^\top}{\|\mathbf{z}\|^6} \right).$$

Thus, we have

$$\|\nabla f_i(\mathbf{z})\| \le 2d \left( \frac{\|\mathbf{A}_i \mathbf{z}\|}{\|\mathbf{z}\|^2} + \frac{|\mathbf{z}^\top \mathbf{A}_i \mathbf{z}|}{\|\mathbf{z}\|^3} \right) \le \frac{\sqrt{d}}{\|\mathbf{z}\|} \cdot \|\mathbf{A}_i \mathbf{x}\| + \frac{|\mathbf{x}^\top \mathbf{A}_i \mathbf{x}|}{\|\mathbf{z}\|}.$$

and

$$\left\| \nabla^2 f_i(\mathbf{z}) \right\|_{op} \lesssim \frac{d}{\|\mathbf{z}\|^2} \|\mathbf{A}_i\|_{op}.$$

Since $\|\mathbf{z}\|^2$ is distributed as a chi-squared random variable with $d$ degrees of freedom, and thus

$$\mathbb{E}\left[ \|\mathbf{z}\|^{-2k} \right] = \frac{1}{\prod_{j=1}^k (d - 2j)}.$$

Therefore, we have

$$\mathbb{E}\left[ \left\| \nabla^2 f_i(\mathbf{z}) \right\|_{op}^4 \right]^{1/4} \lesssim d \|\mathbf{A}_i\|_{op} \mathbb{E}\left[ \|\mathbf{z}\|^{-8} \right]^{1/4} \lesssim \|\mathbf{A}_i\|_{op}.$$

Then, using the fact that $\mathbf{x}$ and $\|\mathbf{z}\|$ are independent,

$$\mathbb{E}\left[ \|\nabla f_i(\mathbf{z})\|^4 \right]^{1/4} \lesssim \sqrt{d}\mathbb{E}\left[ \|\mathbf{z}\|^{-4} \right]^{1/4} \mathbb{E}\left[ \|\mathbf{A}_i \mathbf{x}\|^4 \right]^{1/4} + \mathbb{E}\left[ \|\mathbf{z}\|^{-4} \right]^{1/4} \mathbb{E}\left[ (\mathbf{x}^T \mathbf{A}_i \mathbf{x})^4 \right]^{1/4} \lesssim 1.$$

Thus by Lemma 1 we have

$$W_1\big((\mathbf{x}^\top \mathbf{A}_1 \mathbf{x}, \ldots, \mathbf{x}^\top \mathbf{A}_r \mathbf{x}), \mathcal{N}(\mathbf{0}_r, \mathbf{I}_r)\big)$$
$$= W_1(f_1(\mathbf{z}), \ldots, f_r(\mathbf{z}), \mathcal{N}(\mathbf{0}_r, \mathbf{I}_r))$$
$$\lesssim \frac{4}{\sqrt{\pi}} \left( \sum_{i=1}^r \mathbb{E}\left[ \|\nabla f_i(\mathbf{z})\|^4 \right]^{1/4} \right) \left( \sum_{j=1}^r \mathbb{E}\left[ \|\nabla^2 f_j(\mathbf{z})\|^4 \right]^{1/4} \right)$$
$$\lesssim \frac{r^2 \kappa_1}{\sqrt{d}}.$$

Now let's focus on the function $g_j(\mathbf{z}) = \sqrt{d} \frac{\mathbf{u}_j^\top \mathbf{z}}{\|\mathbf{z}\|}$. It holds that

$$\nabla g_j(\mathbf{z}) = \sqrt{d} \left( \frac{\mathbf{u}_j}{\|\mathbf{z}\|} - \frac{\mathbf{u}_j^\top \mathbf{z} \cdot \mathbf{z}}{\|\mathbf{z}\|^3} \right)$$

and

$$\nabla^2 g_j(\mathbf{z}) = \sqrt{d} \left( -\frac{\mathbf{u}_j \mathbf{z}^\top + \mathbf{z} \mathbf{u}_j^\top}{\|\mathbf{z}\|^3} + 3 \frac{\mathbf{u}_j^\top \mathbf{z} \cdot \mathbf{z} \mathbf{z}^\top}{\|\mathbf{z}\|^5} \right).$$

Thus, we have

$$\|\nabla g_j(\mathbf{z})\| \le \frac{2\sqrt{d}}{\|\mathbf{z}\|} \quad \text{and} \quad \left\| \nabla^2 g_j(\mathbf{z}) \right\|_{op} \le \frac{5\sqrt{d}}{\|\mathbf{z}\|^2},$$

which directly gives rise to

$$\mathbb{E}\left[ \|\nabla g_j(\mathbf{z})\|^4 \right]^{1/4} \lesssim 1 \quad \text{and} \quad \mathbb{E}\left[ \|\nabla^2 g_j(\mathbf{z})\|_{op}^4 \right]^{1/4} \lesssim \frac{1}{\sqrt{d}}.$$

Again by Lemma 1, we have

$$W_1\big((\mathbf{x}^\top \mathbf{A}_1 \mathbf{x}, \cdots, \mathbf{x}^\top \mathbf{A}_r \mathbf{x}, \mathbf{u}_1^\top \mathbf{x}, \mathbf{u}_2^\top \mathbf{x}, \cdots, \mathbf{u}_s^\top \mathbf{x}), \mathcal{N}(\mathbf{0}_{r+s}, \mathbf{I}_{r+s})\big)$$
$$= W_1(f_1(\mathbf{z}), \ldots, f_r(\mathbf{z}), g_1(\mathbf{z}), \cdots, g_s(\mathbf{z}), \mathcal{N}(\mathbf{0}_{r+s}, \mathbf{I}_{r+s}))$$
$$\lesssim \frac{4}{\sqrt{\pi}} \left( \sum_{i=1}^r \mathbb{E}\left[ \|\nabla f_i(\mathbf{z})\|^4 \right]^{1/4} \sum_{i=1}^s \mathbb{E}\left[ \|\nabla g_i(\mathbf{z})\|^4 \right]^{1/4} \right)$$
$$\cdot \left( \sum_{j=1}^r \mathbb{E}\left[ \|\nabla^2 f_j(\mathbf{z})\|^4 \right]^{1/4} + \sum_{j=1}^s \mathbb{E}\left[ \|\nabla^2 g_j(\mathbf{z})\|^4 \right]^{1/4} \right)$$
$$\lesssim \frac{(r+s)^2 \kappa_1}{\sqrt{d}}.$$

The proof is complete. $\qquad\qquad\qquad\qquad\qquad\qquad\qquad\qquad\qquad\qquad\qquad$ $\square$

With the lemma above, we begin our proof of Lemma 13.

*Proof of Lemma 13.* For $\mathbf{z} \in \mathbb{R}^d$, define $H(\mathbf{z}) = g(\mathbf{z}) z_i z_j$. Then by Stein's Lemma, we have
$$\mathbb{E}_{\mathbf{z} \sim \mathcal{N}(\mathbf{0}_r, \mathbf{I}_r)}[H(\mathbf{z})] = \mathbb{E}_{\mathbf{z} \sim \mathcal{N}(\mathbf{0}_r, \mathbf{I}_r)} \left[ \nabla^2 g(\mathbf{z}) \right]_{i,j} + \delta_{i,j} \mathbb{E}_{\mathbf{z} \sim \mathcal{N}(\mathbf{0}_r, \mathbf{I}_r)}[g(\mathbf{z})].$$

Moreover, let $R > 0$ be a truncation radius and we define $\overline{H}(\mathbf{z}) = H(\mathrm{clip}(\mathbf{z}, R))$. Here the clipping function is defined as
$$\mathrm{clip}(\mathbf{z}, R)_i = \max(\min(z_i, R), -R), \quad i = 1, 2, \ldots, r.$$

By (8), we know $g(\mathrm{clip}(\mathbf{z}, R))$ is $\mathcal{O}(R r^{\frac{p-1}{2}})$-Lipschitz continuous, so $\overline{H}$ has a Lipschitz constant of $\mathcal{O}(R^3 r^{\frac{p-1}{2}})$. Thus, by Lemma 16, we have

$$\left| \mathbb{E}_{\mathbf{x}} \left[ g(\mathrm{clip}(\mathbf{x}^\top \mathbf{A}_1 \mathbf{x}, \ldots, \mathbf{x}^\top \mathbf{A}_r \mathbf{x}), R) \right] - \mathbb{E}_{\mathbf{z} \sim \mathcal{N}(\mathbf{0}_r, \mathbf{I}_r)}[g(\mathrm{clip}(\mathbf{z}, R))] \right| \lesssim \frac{R r^{\frac{p-1}{2}} \cdot r^2 \kappa_1}{\sqrt{d}},$$

$$\left| \mathbb{E} \left[ \overline{H}(\mathbf{x}^\top \mathbf{A}_1 \mathbf{x}, \ldots, \mathbf{x}^\top \mathbf{A}_r \mathbf{x}) \right] - \mathbb{E}_{\mathbf{z} \sim \mathcal{N}(\mathbf{0}_r, \mathbf{I}_r)} \left[ \overline{H}(\mathbf{z}) \right] \right| \lesssim \frac{R^3 r^{\frac{p-1}{2}} \cdot r^2 \kappa_1}{\sqrt{d}}.$$

Since $\mathbb{E}_{\mathbf{x}}[(\mathbf{x}^\top \mathbf{A}_k \mathbf{x})^2] = 1$ for any $k \in [r]$, by Lemma 8, choosing $R = C \log d$ for an appropriate constant $C$ can ensure that

$$\left| \mathbb{E}\left[ \overline{H}(\mathbf{x}^\top \mathbf{A}_1 \mathbf{x}, \dots, \mathbf{x}^\top \mathbf{A}_r \mathbf{x}) - H(\mathbf{x}^\top \mathbf{A}_1 \mathbf{x}, \dots, \mathbf{x}^\top \mathbf{A}_r \mathbf{x}) \right] \right| \leq \frac{1}{d},$$

$$\left| \mathbb{E}_{\mathbf{x}}\left[ g(\mathrm{clip}(\mathbf{x}^\top \mathbf{A}_1 \mathbf{x}, \dots, \mathbf{x}^\top \mathbf{A}_r \mathbf{x}), R) - g(\mathbf{x}^\top \mathbf{A}_1 \mathbf{x}, \dots, \mathbf{x}^\top \mathbf{A}_r \mathbf{x}) \right] \right| \leq \frac{1}{d},$$

$$\left| \mathbb{E}_{\mathbf{z} \sim \mathcal{N}(\mathbf{0}_r, \mathbf{I}_r)}\left[ H(\mathbf{z}) - \overline{H}(\mathbf{z}) \right] \right| \leq \frac{1}{d},$$

$$\left| \mathbb{E}_{\mathbf{z} \sim \mathcal{N}(\mathbf{0}_r, \mathbf{I}_r)}\left[ g(\mathrm{clip}(\mathbf{z}, R)) - g(\mathbf{z}) \right] \right| \leq \frac{1}{d}.$$

Altogether, we have

$$\left| \mathbb{E}\left[ g^\star(\mathbf{x}^\top \mathbf{A}_1 \mathbf{x}, \dots, \mathbf{x}^\top \mathbf{A}_r \mathbf{x}) \langle \mathbf{A}_i, \mathbf{x}\mathbf{x}^\top - \mathbf{I} \rangle \langle \mathbf{A}_j, \mathbf{x}\mathbf{x}^\top - \mathbf{I} \rangle \right] - \mathbb{E}_{\mathbf{z} \sim \mathcal{N}(\mathbf{0}_r, \mathbf{I}_r)}\left[ \nabla^2 g(\mathbf{z}) \right]_{i,j} \right| \leq \frac{L r^2 \kappa_1 \log^2 d}{\sqrt{d}}.$$

Here $L = C_g R r^{\frac{p-1}{2}}$ for some constant $C_g > 0$ is the Lipschitz constant of $g^\star$ that holds with high probability. The proof is complete. $\qquad\square$

Following the above proof and replacing $\mathbf{A}_i$ and $\mathbf{A}_j$ by any other traceless matrices $\mathbf{B}_1$ and $\mathbf{B}_2$ that are orthogonal to all $\mathbf{A}_k$, we directly have the following corollary:

**Corollary 1.** *For any two matrices $\mathbf{B}_1, \mathbf{B}_2 \in \mathbb{R}^{d \times d}$ satisfying $\mathbb{E}\left[ \mathbf{x}^\top \mathbf{B}_j \mathbf{x} \right] = 0$ and $\langle \mathbf{B}_j, \mathbf{A}_i \rangle = 0$ for any $i = 1, 2, \dots, r$ and $j = 1, 2$, we have*

$$\left| \mathbb{E}\left[ g^\star(\mathbf{x}^\top \mathbf{A}_1 \mathbf{x}, \dots, \mathbf{x}^\top \mathbf{A}_r \mathbf{x}) \langle \mathbf{A}_i, \mathbf{x}\mathbf{x}^\top - \mathbf{I} \rangle \langle \mathbf{B}_j, \mathbf{x}\mathbf{x}^\top - \mathbf{I} \rangle \right] \right|$$

$$\lesssim \left( \frac{r\kappa_1}{\sqrt{d}} + \frac{\|\mathbf{B}_j\|_{op}}{\|\mathbf{B}_j\|_F} \right) \|\mathbf{B}_j\|_F \, L r \log^2 d,$$

*for any $j = 1, 2$, and*

$$\left| \mathbb{E}\left[ g^\star(\mathbf{x}^\top \mathbf{A}_1 \mathbf{x}, \dots, \mathbf{x}^\top \mathbf{A}_r \mathbf{x}) \langle \mathbf{B}_1, \mathbf{x}\mathbf{x}^\top - \mathbf{I} \rangle \langle \mathbf{B}_2, \mathbf{x}\mathbf{x}^\top - \mathbf{I} \rangle \right] \right|$$

$$\lesssim \left( \frac{r\kappa_1}{\sqrt{d}} + \frac{\|\mathbf{B}_1\|_{op}}{\|\mathbf{B}_1\|_F} + \frac{\|\mathbf{B}_2\|_{op}}{\|\mathbf{B}_2\|_F} \right) \|\mathbf{B}_1\|_F \|\mathbf{B}_2\|_F \, L r \log^2 d.$$

Also, by (17) we know for any unit vector $u \in \mathbb{R}^d$, $(\mathbf{x}^\top \mathbf{A}_1 \mathbf{x}, \mathbf{x}^\top \mathbf{A}_2 \mathbf{x}, \dots, \mathbf{x}^\top \mathbf{A}_r \mathbf{x}, \mathbf{u}^\top \mathbf{x})$ is approximately Gaussian when $d$ is sufficiently large, which gives rise to the following lemma by the same deduction.

**Corollary 2.** *For any $i \in [r]$ and unit vector $\mathbf{u}_1, \mathbf{u}_2 \in \mathbb{R}^d$ and matrix $\mathbf{B}$ satisfying the same requirements in Lemma 14, we have*

$$\left| \mathbb{E}\left[ g^\star(\mathbf{x}^\top \mathbf{A}_1 \mathbf{x}, \dots, \mathbf{x}^\top \mathbf{A}_r \mathbf{x}) \langle \mathbf{A}_i, \mathbf{x}\mathbf{x}^\top - \mathbf{I} \rangle ((\mathbf{u}_j^\top \mathbf{x})^2 - 1) \right] \right| \lesssim \frac{L r^2 \kappa_1 \log^2 d}{\sqrt{d}},$$

$$\left| \mathbb{E}\left[ g^\star(\mathbf{x}^\top \mathbf{A}_1 \mathbf{x}, \dots, \mathbf{x}^\top \mathbf{A}_r \mathbf{x}) \langle \mathbf{B}, \mathbf{x}\mathbf{x}^\top - \mathbf{I} \rangle ((\mathbf{u}_j^\top \mathbf{x})^2 - 1) \right] \right| \lesssim \left( \frac{r\kappa_1}{\sqrt{d}} + \frac{\|\mathbf{B}_j\|_{op}}{\|\mathbf{B}_j\|_F} \right) \|\mathbf{B}_j\|_F \, L r \log^2 d$$

*for any $j = 1, 2$, and we further have*

$$\left| \mathbb{E}\left[ g^\star(\mathbf{x}^\top \mathbf{A}_1 \mathbf{x}, \dots, \mathbf{x}^\top \mathbf{A}_r \mathbf{x}) ((\mathbf{u}_1^\top \mathbf{x})^2 - 1)((\mathbf{u}_2^\top \mathbf{x})^2 - 1) \right] \right| \lesssim \frac{L r^2 \kappa_1 \log^2 d}{\sqrt{d}}.$$

With the lemmas above, we can derive a stronger version of Corollary 1, i.e., Lemma 14, in which the error gets rid of the dependence on $\|\mathbf{B}\|_{op}$.

*Proof of Lemma 14.* Let $\tau > 1/\sqrt{d}$ be a threshold to be determined later. Decompose $\mathbf{B}$ as follows:

$$\mathbf{B} = \sum_{j=1}^{d} \lambda_j \mathbf{u}_j \mathbf{u}_j^\top = \sum_{|\lambda_j| > \tau} \lambda_j \left( \mathbf{u}_j \mathbf{u}_j^\top - \frac{1}{d}\mathbf{I} \right) - \sum_{k=1}^{r} \frac{1}{\|\mathbf{A}_k\|_F^2} \sum_{|\lambda_j| > \tau} \lambda_j \mathbf{u}_j^\top \mathbf{A}_k \mathbf{u}_j \cdot \mathbf{A}_k + \widetilde{\mathbf{B}},$$

where $\{u_j\}_{i=1}^d$ are orthogonal unit vectors and

$$\widetilde{\mathbf{B}} = \sum_{|\lambda_j|\leq\tau} \lambda_j \mathbf{u}_j \mathbf{u}_j^\top + \mathbf{I} \cdot \frac{1}{d} \sum_{|\lambda_j|>\tau} \lambda_j + \sum_{k=1}^r \frac{1}{\|\mathbf{A}_k\|_F^2} \sum_{|\lambda_j|>\tau} \lambda_j \mathbf{u}_j^\top \mathbf{A}_k \mathbf{u}_j \cdot \mathbf{A}_k.$$

By construction, we have

$$\mathrm{tr}(\widetilde{\mathbf{B}}) = \sum_{|\lambda_j|\leq\tau} \lambda_j + \sum_{|\lambda_j|>\tau} \lambda_j = \sum_{j\in[d]} \lambda_j = 0.$$

Moreover, for any $k \in [r]$, we have

$$\langle\widetilde{\mathbf{B}}, \mathbf{A}_k\rangle = \sum_{|\lambda_j|\leq\tau} \lambda_j \mathbf{u}_j^\top \mathbf{A}_k \mathbf{u}_j + \sum_{|\lambda_j|>\tau} \lambda_j \mathbf{u}_j^\top \mathbf{A}_k \mathbf{u}_j = \langle\mathbf{A}_k, \mathbf{B}\rangle = 0.$$

Therefore by Lemma 1, we have

$$\left| \mathbb{E}\left[ g^\star(\mathbf{x}^\top\mathbf{A}_1\mathbf{x}, \ldots, \mathbf{x}^\top\mathbf{A}_r\mathbf{x})\langle\mathbf{A}_i, \mathbf{x}\mathbf{x}^\top - \mathbf{I}\rangle\langle\widetilde{\mathbf{B}}, \mathbf{x}\mathbf{x}^\top - \mathbf{I}\rangle \right] \right|$$

$$\lesssim \left( \frac{r\kappa_1}{\sqrt{d}} + \frac{\left\|\widetilde{\mathbf{B}}\right\|_{op}}{\left\|\widetilde{\mathbf{B}}\right\|_F} \right) \left\|\widetilde{\mathbf{B}}\right\|_F Lr\log^2 d. \tag{18}$$

Since $\sum_{j=1}^d \lambda_j^2 = \|\mathbf{B}\|_F^2 = (d+2)/(2d) = \mathcal{O}(1)$, there are at most $O(\tau^{-2})$ indices $j$ satisfying $|\lambda_j| > \tau$, which gives rise to

$$\sum_{|\lambda_j|>\tau} |\lambda_j| \lesssim \sqrt{\tau^{-2} \cdot \sum_{|\lambda_j|>\tau} |\lambda_j|^2} \leq \tau^{-1}.$$

Thus, we can bound the Frobenius norm of $\widetilde{\mathbf{B}}$ by

$$\left\|\widetilde{\mathbf{B}}\right\|_F^2 \lesssim \sum_{|\lambda_j|\leq\tau} \lambda_j^2 + \frac{1}{d}\left(\sum_{\lambda_j>\tau}\lambda_j\right)^2 + \left\|\sum_{k=1}^r \frac{1}{\|\mathbf{A}_k\|_F^2}\sum_{|\lambda_j|>\tau}\lambda_j\mathbf{u}_j^\top\mathbf{A}_k\mathbf{u}_j\cdot\mathbf{A}_k\right\|_F^2$$

$$= \left\|\widetilde{\mathbf{B}}\right\|_F^2 \sum_{|\lambda_j|\leq\tau}\lambda_j^2 + \frac{1}{d}\left(\sum_{\lambda_j>\tau}\lambda_j\right)^2 + \sum_{k=1}^r\left(\sum_{|\lambda_j|>\tau}\lambda_j\mathbf{u}_j^\top\mathbf{A}_k\mathbf{u}_j\right)^2$$

$$\lesssim \sum_{|\lambda_j|\leq\tau}\lambda_j^2 + \left(\frac{1}{d}+\sum_{k=1}^r\|\mathbf{A}_k\|_{op}^2\right)\left(\sum_{|\lambda_j|>\tau}\lambda_j\right)^2$$

$$\lesssim 1 + \frac{r\kappa_1^2}{d\tau^2}.$$

Thus, we have $\left\|\widetilde{\mathbf{B}}\right\|_F \lesssim 1 + \frac{\sqrt{r}\kappa_1}{\sqrt{d}\tau}$ and

$$\left\|\widetilde{\mathbf{B}}\right\|_{op} \leq \tau + \left(\frac{1}{d}+\sum_{k=1}^r\|\mathbf{A}_k\|_{op}\right)\left|\sum_{|\lambda_j|>\tau}\lambda_j u_j^\top A u_j\right|$$

$$\lesssim \tau + \frac{r\kappa_1}{d\tau}.$$

Thus, plugging the norm bounds into (18), we obtain that

$$\left|\mathbb{E}\left[g^\star(\mathbf{x}^\top\mathbf{A}_1\mathbf{x},\ldots,\mathbf{x}^\top\mathbf{A}_r\mathbf{x})\langle\mathbf{A}_i, \mathbf{x}\mathbf{x}^\top-\mathbf{I}\rangle\langle\widetilde{\mathbf{B}}, \mathbf{x}\mathbf{x}^\top-\mathbf{I}\rangle\right]\right| \lesssim \left(\frac{r\kappa_1}{\sqrt{d}}+\frac{r^{3/2}\kappa_1^2}{d\tau}+\tau+\frac{r\kappa_1}{d\tau}\right)Lr\log^2 d.$$

Next, applying Corollary 2 with $\mathbf{u} = \mathbf{u}_1, \mathbf{u}_2, \ldots, \mathbf{u}_d$, we have

$$\left|\mathbb{E}\left[g^\star(\mathbf{x}^\top\mathbf{A}_1\mathbf{x},\ldots,\mathbf{x}^\top\mathbf{A}_r\mathbf{x})\langle\mathbf{A}_i, \mathbf{x}\mathbf{x}^\top-\mathbf{I}\rangle\langle\mathbf{u}_j\mathbf{u}_j^\top-\mathbf{I}/d, \mathbf{x}\mathbf{x}^\top-\mathbf{I}\rangle\right]\right| \lesssim \frac{Lr^2\kappa_1\log^2 d}{\sqrt{d}}, \quad \forall j\in[d].$$

Thus, we have

$$
\left| \mathbb{E} \left[ g^\star(\mathbf{x}^\top \mathbf{A}_1 \mathbf{x}, \ldots, \mathbf{x}^\top \mathbf{A}_r \mathbf{x}) \langle \mathbf{A}_i, \mathbf{x}\mathbf{x}^\top - \mathbf{I} \rangle \left\langle \sum_{|\lambda_j|>\tau} \lambda_j \left( \mathbf{u}_j \mathbf{u}_j^\top - \frac{1}{d}\mathbf{I} \right), \mathbf{x}\mathbf{x}^\top - \mathbf{I} \right\rangle \right] \right|
$$

$$
\leq \sum_{|\lambda_j|>\tau} |\lambda_j| \frac{L r^2 \kappa_1 \log^2 d}{\sqrt{d}}
$$

$$
\lesssim \frac{L r^2 \kappa_1 \log^2 d}{\sqrt{d}\tau}.
$$

Besides, by Lemma 13, we have

$$
\left| \mathbb{E} \left[ g^\star(\mathbf{x}^\top \mathbf{A}_1 \mathbf{x}, \ldots, \mathbf{x}^\top \mathbf{A}_r \mathbf{x}) \langle \mathbf{A}_i, \mathbf{x}\mathbf{x}^\top - \mathbf{I} \rangle \left\langle \sum_{k=1}^r \frac{1}{\|\mathbf{A}_k\|_F^2} \sum_{|\lambda_j|>\tau} \lambda_j \mathbf{u}_j^\top \mathbf{A}_k \mathbf{u}_j \cdot \mathbf{A}_k, \mathbf{x}\mathbf{x}^\top - \mathbf{I} \right\rangle \right] \right|
$$

$$
\lesssim \sum_{k=1}^r \left| \sum_{|\lambda_j|>\tau} \lambda_j \mathbf{u}_j^\top \mathbf{A}_k \mathbf{u}_j \right| \left( \mathbb{E}_{\mathbf{z} \sim \mathcal{N}(\mathbf{0}_r, \mathbf{I}_r)} \left[ \nabla^2 g(\mathbf{z}) \right]_{k,i} + \frac{L r^2 \kappa_1 \log^2 d}{\sqrt{d}} \right)
$$

$$
\lesssim \sum_{k=1}^r L \left| \sum_{|\lambda_j|>\tau} \lambda_j \mathbf{u}_j^\top \mathbf{A}_k \mathbf{u}_j \right|
$$

$$
\lesssim \frac{L r \kappa_1}{\sqrt{d}\tau}.
$$

Altogether, we have

$$
\left| \mathbb{E} \left[ g^\star(\mathbf{x}^\top \mathbf{A}_1 \mathbf{x}, \ldots, \mathbf{x}^\top \mathbf{A}_r \mathbf{x}) \langle \mathbf{A}_i, \mathbf{x}\mathbf{x}^\top - \mathbf{I} \rangle \langle \mathbf{B}, \mathbf{x}\mathbf{x}^\top - \mathbf{I} \rangle \right] \right|
$$

$$
\lesssim \left( \frac{r \kappa_1}{\sqrt{d}} + \frac{r^{3/2} \kappa_1^2}{d\tau} + \tau + \frac{r \kappa_1}{d\tau} \right) L r \log^2 d + \frac{L r^2 \kappa_1 \log^2 d}{\sqrt{d}\tau} + \frac{L r \kappa_1}{\sqrt{d}\tau}
$$

$$
\lesssim d^{-1/4} L r^2 \kappa_1 \log^2 d.
$$

where we set $\tau = \kappa_1 d^{-1/4}$. The proof is complete. $\qquad\square$

Following the proof above, we can complete the proof of Lemma 15.

*Proof of Lemma 15.* Similar to the proof of Lemma 14, we decompose $\mathbf{B}_1$ and $\mathbf{B}_2$ as follows:

$$
\mathbf{B}_i = \sum_{j=1}^d \lambda_{i,j} \mathbf{u}_{i,j} \mathbf{u}_{i,j}^\top
$$

$$
= \sum_{|\lambda_{i,j}|>\tau} \lambda_{i,j} \left( \mathbf{u}_{i,j} \mathbf{u}_{i,j}^\top - \frac{1}{d}\mathbf{I} \right) - \sum_{k=1}^r \frac{1}{\|\mathbf{A}_k\|_F^2} \sum_{|\lambda_{i,j}|>\tau} \lambda_{i,j} \mathbf{u}_{i,j}^\top \mathbf{A}_k \mathbf{u}_{i,j} \cdot \mathbf{A}_k + \widetilde{\mathbf{B}}_i,
$$

where $\{u_{i,j}\}_{j=1}^d$ are orthogonal unit vectors for $i = 1, 2$, respectively, and

$$
\widetilde{\mathbf{B}}_i = \sum_{|\lambda_{i,j}|\leq\tau} \lambda_{i,j} \mathbf{u}_{i,j} \mathbf{u}_{i,j}^\top + \mathbf{I} \cdot \frac{1}{d} \sum_{|\lambda_{i,j}|>\tau} \lambda_{i,j} + \sum_{k=1}^r \frac{1}{\|\mathbf{A}_k\|_F^2} \sum_{|\lambda_{i,j}|>\tau} \lambda_{i,j} \mathbf{u}_{i,j}^\top \mathbf{A}_k \mathbf{u}_{i,j} \cdot \mathbf{A}_k.
$$

Then following the proof of Lemma 14, we know for any $i = 1, 2$ and $k = 1, 2, \ldots, r$,

$$
\mathrm{tr}\left(\widetilde{\mathbf{B}}_i\right) = \langle \widetilde{\mathbf{B}}_i, \mathbf{A}_k \rangle = 0, \quad \left\|\widetilde{\mathbf{B}}_i\right\|_F \lesssim 1 + \frac{\sqrt{r}\kappa_1}{\sqrt{d}\tau}, \quad \left\|\widetilde{\mathbf{B}}_i\right\|_{\mathrm{op}} \lesssim \tau + \frac{r\kappa_1}{d\tau}, \quad \sum_{|\lambda_{i,j}|>\tau} |\lambda_{i,j}| \lesssim \tau^{-1}.
$$

Let's denote the bi-linear operator $\Gamma(\cdot, \cdot) : \mathbb{R}^{d\times d} \times \mathbb{R}^{d\times d} \to \mathbb{R}$ being

$$
\Gamma(\mathbf{A}, \mathbf{B}) = \mathbb{E}\left[ f^\star(\mathbf{x}) \langle \mathbf{A}, \mathbf{x}\mathbf{x}^\top - \mathbf{I} \rangle \langle \mathbf{B}, \mathbf{x}\mathbf{x}^\top - \mathbf{I} \rangle \right]. \tag{19}
$$

By Corollary 1 and the proof of Lemma 14, we have

$$\left| \Gamma(\widetilde{\mathbf{B}}_1, \widetilde{\mathbf{B}}_2) \right| \lesssim \left( \frac{r\kappa_1}{\sqrt{d}} \left( 1 + \frac{r\kappa_1^2}{d\tau^2} \right) + \left( \tau + \frac{r\kappa_1}{d\tau} \right) \left( 1 + \frac{\sqrt{r}}{\sqrt{d}\tau} \right) \right) Lr \log^2 d. \tag{20}$$

$$\left| \Gamma\left( \widetilde{\mathbf{B}}_i, \sum_{|\lambda_{-i,j}|>\tau} \lambda_{-i,j} \left( \mathbf{u}_{-i,j}\mathbf{u}_{-i,j}^\top - \mathbf{I}/d \right) \right) \right|$$

$$\lesssim \tau^{-1} \left( \frac{r\kappa_1}{\sqrt{d}} + \frac{r^{3/2}\kappa_1^2}{d\tau} + \tau + \frac{r\kappa_1}{d\tau} \right) Lr \log^2 d \tag{21}$$

$$\left| \Gamma\left( \widetilde{\mathbf{B}}_i, \sum_{k=1}^r \frac{1}{\|\mathbf{A}_k\|_F^2} \sum_{|\lambda_{-i,j}|>\tau} \lambda_{-i,j}\mathbf{u}_{-i,j}^\top \mathbf{A}_k \mathbf{u}_{-i,j} \cdot \mathbf{A}_k \right) \right|$$

$$\lesssim \tau^{-1} \left( \frac{r\kappa_1}{\sqrt{d}} + \frac{r^{3/2}\kappa_1^2}{d\tau} + \tau + \frac{r\kappa_1}{d\tau} \right) Lr \log^2 d. \tag{22}$$

Here $-i$ means 2 when $i = 1$ and 1 when $i = 2$. Moreover, by Lemma 13, we have

$$\left| \Gamma\left( \sum_{k=1}^r \frac{1}{\|\mathbf{A}_k\|_F^2} \sum_{|\lambda_{1,j}|>\tau} \lambda_{1,j}\mathbf{u}_{1,j}^\top \mathbf{A}_k \mathbf{u}_{1,j} \cdot \mathbf{A}_k, \sum_{k=1}^r \frac{1}{\|\mathbf{A}_k\|_F^2} \sum_{|\lambda_{2,j}|>\tau} \lambda_{2,j}\mathbf{u}_{2,j}^\top \mathbf{A}_k \mathbf{u}_{2,j} \cdot \mathbf{A}_k \right) \right|$$

$$\lesssim \left( \sum_{k=1}^r \left| \sum_{|\lambda_{1,j}|>\tau} \lambda_{1,j}\mathbf{u}_{1,j}^\top \mathbf{A}_k \mathbf{u}_{1,j} \right| \right) \left( \sum_{k=1}^r \left| \sum_{|\lambda_{2,j}|>\tau} \lambda_{2,j}\mathbf{u}_{2,j}^\top \mathbf{A}_k \mathbf{u}_{2,j} \right| \right) \left( 1 + \frac{Lr^2\kappa_1 \log^2 d}{\sqrt{d}} \right)$$

$$\lesssim \frac{r^2\kappa_1^2}{d\tau^2}, \tag{23}$$

and

$$\left| \Gamma\left( \sum_{k=1}^r \frac{1}{\|\mathbf{A}_k\|_F^2} \sum_{|\lambda_{i,j}|>\tau} \lambda_{i,j}\mathbf{u}_{i,j}^\top \mathbf{A}_k \mathbf{u}_{i,j} \cdot \mathbf{A}_k, \sum_{|\lambda_{-i,j}|>\tau} \lambda_{-i,j} \left( \mathbf{u}_{-i,j}\mathbf{u}_{-i,j}^\top - \mathbf{I}/d \right) \right) \right|$$

$$\lesssim \left( \sum_{k=1}^r \left| \sum_{|\lambda_{i,j}|>\tau} \lambda_{i,j}\mathbf{u}_{i,j}^\top \mathbf{A}_k \mathbf{u}_{i,j} \right| \right) \left( \sum_{|\lambda_{-i,j}|>\tau} |\lambda_{-i,j}| \right) \frac{Lr^2\kappa_1 \log^2 d}{\sqrt{d}}$$

$$\lesssim \frac{r\kappa_1}{\sqrt{d}\tau^2} \cdot \frac{Lr^2\kappa_1 \log^2 d}{\sqrt{d}}. \tag{24}$$

Finally, we have

$$\left| \Gamma\left( \sum_{|\lambda_{1,j}|>\tau} \lambda_{1,j} \left( \mathbf{u}_{1,j}\mathbf{u}_{1,j}^\top - \mathbf{I}/d \right), \sum_{|\lambda_{2,j}|>\tau} \lambda_{2,j} \left( \mathbf{u}_{2,j}\mathbf{u}_{2,j}^\top - \mathbf{I}/d \right) \right) \right|$$

$$\lesssim \left( \sum_{|\lambda_{1,j}|>\tau} |\lambda_{1,j}| \right) \left( \sum_{|\lambda_{2,j}|>\tau} |\lambda_{2,j}| \right) \frac{Lr^2\kappa_1 \log^2 d}{\sqrt{d}}.$$

$$\lesssim \frac{1}{\tau^2} \cdot \frac{Lr^2\kappa_1 \log^2 d}{\sqrt{d}}. \tag{25}$$

Summing (20) to (25) altogether, we have

$$
\begin{aligned}
|\Gamma(\mathbf{B}_1, \mathbf{B}_2)| \lesssim & \left( \frac{r\kappa_1}{\sqrt{d}} \left(1 + \frac{r\kappa_1^2}{d\tau^2}\right) + \left(\tau + \frac{r\kappa_1}{d\tau}\right) \left(1 + \frac{\sqrt{r}}{\sqrt{d}\tau}\right) \right) Lr \log^2 d \\
& + \tau^{-1} \left( \frac{r\kappa_1}{\sqrt{d}} + \frac{r^{3/2}\kappa_1^2}{d\tau} + \tau + \frac{r\kappa_1}{d\tau} \right) Lr \log^2 d \\
& + \frac{r}{\sqrt{d}\tau} \cdot \frac{Lr^2\kappa_1 \log^2 d}{\sqrt{d}} + \frac{r^2\kappa_1^2}{d\tau^2} + \frac{r\kappa_1}{\sqrt{d}\tau^2} \cdot \frac{Lr^2\kappa_1 \log^2 d}{\sqrt{d}} + \frac{1}{\tau^2} \cdot \frac{Lr^2\kappa_1 \log^2 d}{\sqrt{d}} \\
\lesssim & \; d^{-1/6} Lr^2\kappa_1 \log^2 d,
\end{aligned}
$$

where we take $\tau = \kappa_1 d^{-1/6}$. The proof is complete. $\qquad\square$

## C.2 BOUNDEDNESS OF THE LEARNED FEATURE

In this section, we aim to upper bound the magnitude of the learned feature $\langle \mathbf{w}, \mathbf{h}^{(1)}(\mathbf{x}') \rangle$. Since we focus on the first training stage throughout this section, we denote $n = n_1$ for notation simplicity when the context is clear, and let the training set be $\mathcal{D}_1 = \{\mathbf{x}_1, \mathbf{x}_2, \ldots, \mathbf{x}_n\}$. We have the following proposition:

**Proposition 4.** *Suppose $m_2 \geqslant d^4 C_\sigma^4$ and $n \geqslant C\iota^2 d^2$ for some sufficiently large $C$. With high probability jointly on $\mathbf{V}$ and the training dataset $\mathcal{D}_1$, and with probability at least $1 - 4n\exp(-\iota^2/2)$ on $\mathbf{w}$, for any $\mathbf{x}' \in \mathcal{D}_2$, we have*

$$
\left| \langle \mathbf{w}, \mathbf{h}^{(1)}(\mathbf{x}') \rangle \right| \lesssim \frac{\iota^{p+2}}{d^3} + \frac{\iota^{p+2}\sqrt{m_2}}{d^4\sqrt{n}} + \frac{\sqrt{m_2}\iota^3 \log^2(m_2 n_2)}{d^6} \cdot \left( \|\mathcal{P}_{>2}(f^\star)\|_{L^2} + \sqrt{d}\|\mathcal{P}_2(f^\star)\|_{L^2} \right).
$$

As a corollary, when $m_2 \gtrsim d^6\iota^{2p+4}$, $n \gtrsim d^4\iota^{2p+4}$ and $\|\mathcal{P}_2(f^\star)\|_{L^2} \lesssim \frac{\kappa_2}{\sqrt{d}}$, we have for any $\mathbf{x}' \in \mathcal{D}_2$,

$$
\frac{1}{\sqrt{m_2}} \left| \langle \mathbf{w}, \mathbf{h}^{(1)}(\mathbf{x}') \rangle \right| \lesssim \frac{\kappa_2 \iota^5}{d^6}. \tag{26}
$$

Thus, by taking the learning rate $\eta = C m_2^{-1/2} \kappa_2^{-1} \iota^{-5} d^6$ for an appropriate constant $C > 0$, we can ensure that $\left| \eta \langle \mathbf{w}, \mathbf{h}^{(1)}(\mathbf{x}') \rangle \right| \leq 1$ with high probability.

*Proof of Proposition 4.* Note that

$$
\begin{aligned}
\frac{1}{m_2} \langle \mathbf{w}, \mathbf{h}^{(1)}(\mathbf{x}') \rangle &= \frac{1}{m_2 n} \sum_{i=1}^{n} \sum_{j=1}^{m_2} w_j f^\star(\mathbf{x}_i) K_{m_2}^{(0)}(\mathbf{x}_i, \mathbf{x}') \sigma_2(\mathbf{v}_j^\top \mathbf{x}_i) \\
&= \frac{1}{m_2} \sum_{j=1}^{m_2} \frac{1}{n} \sum_{i=1}^{n} f^\star(\mathbf{x}_i) K_{m_2}^{(0)}(\mathbf{x}_i, \mathbf{x}') w_j \sigma_2(\mathbf{v}_j^\top \mathbf{x}_i).
\end{aligned}
$$

We do a decomposition as follows

$$
\begin{aligned}
&\frac{1}{m_2} \sum_{j=1}^{m_2} \frac{1}{n} \sum_{i=1}^{n} f^\star(\mathbf{x}_i) K_{m_2}^{(0)}(\mathbf{x}_i, \mathbf{x}') w_j \sigma_2(\mathbf{v}_j^\top \mathbf{x}_i) \\
&= \underbrace{\frac{1}{m_2} \sum_{j=1}^{m_2} \frac{1}{n} \sum_{i=1}^{n} f^\star(\mathbf{x}_i) \left( K_{m_2}^{(0)}(\mathbf{x}_i, \mathbf{x}') - K^{(0)}(\mathbf{x}_i, \mathbf{x}') \right) w_j \sigma_2(\mathbf{v}_j^\top \mathbf{x}_i)}_{A_1} \\
&\quad + \underbrace{\frac{1}{m_2} \sum_{j=1}^{m_2} \frac{1}{n} \left( \sum_{i=1}^{n} f^\star(\mathbf{x}_i) K^{(0)}(\mathbf{x}_i, \mathbf{x}') w_j \sigma_2(\mathbf{v}_j^\top \mathbf{x}_i) - \mathbb{E}_{\mathbf{x}} \left[ f^\star(\mathbf{x}) K^{(0)}(\mathbf{x}, \mathbf{x}') w_j \sigma_2(\mathbf{v}_j^\top \mathbf{x}) \right] \right)}_{A_2} \\
&\quad + \underbrace{\frac{1}{m_2} \sum_{j=1}^{m_2} w_j \mathbb{E}_{\mathbf{x}} \left[ f^\star(\mathbf{x}) K^{(0)}(\mathbf{x}, \mathbf{x}') \sigma_2(\mathbf{v}_j^\top \mathbf{x}) \right]}_{A_3}.
\end{aligned}
$$

We consider derive an upper bound on $A_1$, $A_2$ and $A_3$, respectively.

**Lemma 17** (Bound $A_1$). *Suppose $m_2 \geqslant d^4 C_\sigma^4$. With high probability jointly on $\mathbf{V}$ and the training dataset $\mathcal{D}_1$, and with probability at least $1 - 2n \exp(-\iota^2/2)$ on $\mathbf{w}$, for any $\mathbf{x}' \in \mathcal{D}_2$, we have*

$$|A_1| \lesssim \frac{\iota^{p+2}}{m_2 d^3}.$$

**Lemma 18** (Bound $A_2$). *Suppose $m_2 \geqslant d^4 C_\sigma^4$ and $n \geqslant C\iota^2 d^2$ for some sufficiently large $C$. With high probability on the training dataset $\mathcal{D}_1$, for any $\mathbf{x}' \in \mathcal{D}_2$, we have*

$$|A_2| \lesssim \frac{\iota^{p+2}}{d^4 \sqrt{m_2 n}}.$$

**Lemma 19** (Bound $A_3$). *Suppose $m_2 \geqslant d^4 C_\sigma^4$ for some sufficiently large $C$. With high probability jointly on $\mathbf{V}$ and the training dataset $\mathcal{D}_1$, and with probability at least $1 - 2n_2 \exp(-\iota^2/2)$ on $\mathbf{w}$, we have*

$$|A_3| \lesssim \frac{\iota^3 \log^2(m_2 n_2)}{\sqrt{m_2} d^6} \cdot \left( \|\mathcal{P}_{>2}(f)\|_{L^2} + \sqrt{d} \|\mathcal{P}_2(f)\|_{L^2} \right).$$

*Similarly, for a single point $\mathbf{x}'$, with high probability on $\mathbf{V}$ and $\mathcal{D}_1$, with probability $1 - 2\exp(-\iota^2/2)$ on $\mathbf{w}$, we have*

$$|A_3| \lesssim \frac{\iota^3 \log^2(m_2 n_2)}{\sqrt{m_2} d^6} \cdot \left( \|\mathcal{P}_{>2}(f)\|_{L^2} + \sqrt{d} \|\mathcal{P}_2(f)\|_{L^2} \right).$$

The proof of the three lemmas are provided in Appendix C.2.1. Combining the results in the three lemmas above directly concludes our proof. $\qquad\square$

### C.2.1 OMITTED PROOFS FOR PROPOSITION 4

*Proof of Lemma 17.* We can rewrite $A_1$ as

$$|A_1| = \frac{1}{n} \sum_{i=1}^{n} f^\star(\mathbf{x}_i) \left( K_{m_2}^{(0)}(\mathbf{x}_i, \mathbf{x}') - K^{(0)}(\mathbf{x}_i, \mathbf{x}') \right) \frac{1}{m_2} \sum_{j=1}^{m_2} w_j \sigma_2 (\mathbf{v}_j^\top \mathbf{x}_i)$$

Since $\frac{1}{\sqrt{m_2}} \sum_{j=1}^{m_2} w_j \sigma_2 (\mathbf{v}_j^\top \mathbf{x}) \sim \mathcal{N}\left( 0, \frac{1}{m_2} \sum_{j=1}^{m_2} \sigma_2 (\mathbf{v}_j^\top \mathbf{x})^2 \right)$, we know given any $\mathbf{x}$ and $\mathbf{V}$, with probability at least $1 - 2\exp(-\iota^2/2)$ on $\mathbf{w}$, we have

$$\left| \frac{1}{m_2} \sum_{j=1}^{m_2} w_j \sigma_2 (\mathbf{v}_j^\top \mathbf{x}) \right| \leq \frac{\iota}{\sqrt{m_2}} \cdot \sqrt{\frac{1}{m_2} \sum_{j=1}^{m_2} \sigma_2 (\mathbf{v}_j^\top \mathbf{x})^2}$$

Moreover, by (34) in the proof of Lemma 24, we know for any $\mathbf{x}$ and $t > 0$, we have

$$\Pr\left[ \frac{1}{m_2} \sum_{j=1}^{m_2} \left( \sigma_2 (\mathbf{v}_j^\top \mathbf{x})^2 - \mathbb{E}_{\mathbf{v}_j} \left[ \sigma_2 (\mathbf{v}_j^\top \mathbf{x})^2 \right] \right) \geqslant \sqrt{\frac{t}{m_2}} \right]$$

$$= \Pr\left[ \left| K_{m_2}^{(0)}(\mathbf{x}, \mathbf{x}) - K^{(0)}(\mathbf{x}, \mathbf{x}) \right| \geqslant \sqrt{\frac{t}{m_2}} \right]$$

$$\leq 2 \exp\left( \frac{-t/2}{\frac{C_4}{d^4} + \frac{C_\sigma^2}{3} \sqrt{\frac{t}{m_2}}} \right).$$

Altogether, when $m_2 \geqslant d^4 C_\sigma^4$, by taking $t = C^2 \iota^2/d^4$ for sufficiently large $C$ and union bounding over the dataset $\mathcal{D}_1$, we can ensure that with probability at least $1 - n\exp(-\iota)$ on $\mathbf{V}$ and at least $1 - 2n\exp(-\iota^2/2)$ on $\mathbf{w}$, i.e., high probability on $\mathbf{w}, \mathbf{V}$, we have

$$\left| \frac{1}{m_2} \sum_{j=1}^{m_2} w_j \sigma_2 (\mathbf{v}_j^\top \mathbf{x}) \right| \leq \frac{C\iota}{\sqrt{m_2}} \sqrt{\frac{C_2}{d^2} + \frac{\iota}{\sqrt{m_2} d^2}} \leq \frac{\iota C_3}{\sqrt{m_2} d}, \quad \forall \mathbf{x} \in \mathcal{D}_1. \tag{27}$$

Here $C_3$ is a constant. We denote this joint event by $E_1$. On the other hand, by Lemma 24, with high probability on $\mathbf{V}$, we have for any $\mathbf{x}_i \in \mathcal{D}_1$ and $\mathbf{x}' \in \mathcal{D}_2$,

$$\left| K_{m_2}^{(0)}(\mathbf{x}_i, \mathbf{x}') - K^{(0)}(\mathbf{x}_i, \mathbf{x}') \right| \leq \frac{\iota}{\sqrt{m_2}d^2}. \tag{28}$$

We denote this event by $E_2$. Last, we truncate the range of the target function $f$. Denoting the truncation radius as $R = (C\eta)^p$ for a sufficient large constant $C$ and $\eta = \log(dm_1m_2n_1n_2)$ $\Pr\left[|f(\mathbf{x})| \geqslant R\right] \leq 2e^{-2\eta}$ (this could be guaranteed by Lemma 4). Given $n$ i.i.d. samples $\mathbf{x}_1, \mathbf{x}_2 \ldots, \mathbf{x}_n \sim \mathbb{S}^{d-1}(\sqrt{d})$, we have

$$\Pr\left[|f(\mathbf{x}_i)| \leq R, \forall i \in [n]\right] \geqslant 1 - 2ne^{-2\eta}. \tag{29}$$

Thus, with high probability on the dataset $\mathcal{D}$, we have $|f(\mathbf{x})| \leq \iota^p$ for any $\mathbf{x} \in \mathcal{D}_1$. We denote this event by $E_3$. Thus, combining (27) (32) and the truncation radius of $f$, we directly have

$$|A_1| \leq \iota^p \cdot \frac{\iota}{\sqrt{m_2}d^2} \cdot \frac{\iota C_3}{\sqrt{m_2}d} = \frac{C_3 \iota^{p+2}}{m_2 d^3}.$$

with high probability (under events $E_1$, $E_2$ and $E_3$). The proof is complete. $\square$

*Proof of Lemma 18.* We rewrite $A_2$ as

$$A_2 = \frac{1}{n}\sum_{i=1}^n \mathbb{E}_\mathbf{x}\left[ f^\star(\mathbf{x}_i)K^{(0)}(\mathbf{x}_i, \mathbf{x}')\frac{1}{m_2}\sum_{j=1}^{m_2} w_j\sigma_2\big(\mathbf{v}_j^\top \mathbf{x}_i\big) - f^\star(\mathbf{x})K^{(0)}(\mathbf{x}, \mathbf{x}')\frac{1}{m_2}\sum_{j=1}^{m_2} w_j\sigma_2\big(\mathbf{v}_j^\top \mathbf{x}\big) \right].$$

Denote $Y(\mathbf{x}) = f^\star(\mathbf{x})K^{(0)}(\mathbf{x}, \mathbf{x}')\frac{1}{m_2}\sum_{j=1}^{m_2} w_j\sigma_2\big(\mathbf{v}_j^\top \mathbf{x}\big)$. By the proof of bounding $A_1$, we could choose the truncation radius as $R = (C\eta)^p$ such that $|f(\mathbf{x})| \leq R$ for all $\mathbf{x} \in \mathcal{D}$ with high probability $(1 - 2ne^{-2\eta})$ on the dataset $\mathcal{D}$. Now we denote a truncated version of $Y$ by

$$\widetilde{Y}(\mathbf{x}) = f^\star(\mathbf{x})\mathbf{1}\{f^\star(\mathbf{x}) \leq R\}K^{(0)}(\mathbf{x}, \mathbf{x}')\frac{1}{m_2}\sum_{j=1}^{m_2} w_j\sigma_2\big(\mathbf{v}_j^\top \mathbf{x}\big)\mathbf{1}\left\{ \frac{1}{m_2}\sum_{j=1}^{m_2} w_j\sigma_2\big(\mathbf{v}_j^\top \mathbf{x}\big) \leq \frac{\iota C_3}{\sqrt{m_2}d} \right\}.$$

Here $C_3$ is a constant defined in (27). Now, we decompose the concentration error as

$$\frac{1}{n}\sum_{i=1}^n Y(\mathbf{x}_i) - \mathbb{E}_\mathbf{x}\left[Y(\mathbf{x})\right] = \underbrace{\frac{1}{n}\sum_{i=1}^n \left( Y(x_i) - \widetilde{Y}(x_i) \right)}_{\mathcal{L}_0} + \underbrace{\frac{1}{n}\sum_{i=1}^n \left( \widetilde{Y}(x_i) - \mathbb{E}_{x_i}\left[\widetilde{Y}(x_i)\right] \right)}_{\mathcal{L}_1}$$
$$+ \underbrace{\frac{1}{n}\sum_{i=1}^n \left( \mathbb{E}_{x_i}\left[\widetilde{Y}(x_i)\right] - \mathbb{E}_{x_i}\left[Y(x_i)\right] \right)}_{\mathcal{L}_2}.$$

We know with probability at least $1 - 2ne^{-2\eta}$ on $\mathcal{D}$, $\mathcal{L}_0 = 0$.

**Bounding $\mathcal{L}_1$.** We attempt to use Bernstein's type bound. First we derive a uniform upper bound of $\widetilde{Y}(\mathbf{x})$. By the definition, we have

$$\left| \widetilde{Y}(\mathbf{x}) \right| \leq R\left| K^{(0)}(\mathbf{x}, \mathbf{x}') \right|\left| \frac{1}{m_2}\sum_{j=1}^{m_2} w_j\sigma_2\big(\mathbf{v}_j^\top \mathbf{x}\big) \vee \frac{\iota C_3}{\sqrt{m_2}d} \right|$$
$$\leq R \cdot \frac{C_2}{d^2}\frac{\iota C_3}{\sqrt{m_2}d}$$
$$= \frac{RC_2C_3\iota}{d^3\sqrt{m_2}}.$$

Then, we bound the second moments of $\widetilde{Y}(\mathbf{x}) - \mathbb{E}_{\mathbf{x}}\left[\widetilde{Y}(\mathbf{x})\right]$, which is

$$
\begin{aligned}
\mathrm{Var}\left[\widetilde{Y}(\mathbf{x})\right] &\leq \mathbb{E}_{\mathbf{x}}\left[\widetilde{Y}_k^2(\mathbf{x})\right] \\
&\leq r^2\kappa_1 \mathbb{E}_{\mathbf{x}}\left[\left(K^{(0)}(\mathbf{x},\mathbf{x}')\right)^2\left(\frac{1}{m_2}\sum_{j=1}^{m_2}w_j\sigma_2(\mathbf{v}_j^\top\mathbf{x})\vee\frac{\iota C_3}{\sqrt{m_2d}}\right)^2\right] \\
&\leq r^2\kappa_1 \mathbb{E}_{\mathbf{x}}\left[\left(K^{(0)}(\mathbf{x},\mathbf{x}')\right)^2\right]\left(\frac{\iota C_3}{\sqrt{m_2d}}\right)^2 \\
&\leq r^2\kappa_1 \sum_{k=2}^\infty\frac{c_k^4}{B(d,k)^3}\cdot\left(\frac{\iota C_3}{\sqrt{m_2d}}\right)^2 \\
&\leq \frac{C_4 r^2\kappa_1\iota^2}{m_2 d^8}.
\end{aligned}
$$

Here $C_4$ is a constant. Thus, by Bernstein's inequality, we have

$$
\Pr\left[|\mathcal{L}_1|\geqslant\frac{R\iota}{d^4\sqrt{m_2}}\sqrt{\frac{t}{n}}\right]\leq 2\exp\left(\frac{-\frac{t}{2}}{C_4+\frac{C_2 C_3}{3}\sqrt{\frac{d^2 t}{n}}}\right).
$$

Thus, when $n\geqslant C_\epsilon\iota^2 d^2$, by taking $t=\iota^2$ and $R=(C\eta)^p\leq\iota^p$, with high probability on $\mathcal{D}$, $\mathbf{V}$ and $\mathbf{w}$, we have

$$
|\mathcal{L}_1|\leq\frac{\iota^{p+2}}{d^4\sqrt{m_2 n}}.
$$

**Bounding $\mathcal{L}_2$.** It suffices to bound

$$
\left|\mathbb{E}_{\mathbf{x}}\left[\widetilde{Y}_k(\mathbf{x})\right]-\mathbb{E}_{\mathbf{x}}\left[Y_k(\mathbf{x})\right]\right|
$$

$$
\leq\mathbb{E}_{\mathbf{x}}\left[|f^\star(\mathbf{x})|\left|K^{(0)}(x,x')\frac{1}{m_2}\sum_{j=1}^{m_2}w_j\sigma_2(\mathbf{v}_j^\top\mathbf{x})\right|\mathbf{1}\left\{f^\star(\mathbf{x})>R\ \text{ or }\ \frac{1}{m_2}\sum_{j=1}^{m_2}w_j\sigma_2(\mathbf{v}_j^\top\mathbf{x})>\frac{\iota C_3}{\sqrt{m_2 d}}\right\}\right]
$$

$$
\leq\mathbb{E}_{\mathbf{x}}\left[(f^\star(\mathbf{x}))^2\right]^{\frac{1}{2}}\Pr\left[f^\star(\mathbf{x})>R\ \text{ or }\ \frac{1}{m_2}\sum_{j=1}^{m_2}w_j\sigma_2(\mathbf{v}_j^\top\mathbf{x})>\frac{\iota C_3}{\sqrt{m_2 d}}\right]^{\frac{1}{4}}\mathbb{E}_{\mathbf{x}}\left[K^{(0)}(\mathbf{x},\mathbf{x}')^4\right]^{\frac{1}{4}}\frac{\tau C_3}{\sqrt{m_2 d}}
$$

$$
\lesssim\frac{(\exp(-\eta)+\exp(-\iota))\iota C_3}{d^3\sqrt{m_2}}.
$$

Taking $\eta\geqslant 2\log n+2\log d+\log(C_3)$ and $\iota\geqslant C\eta$, we ensure that $\mathcal{L}_2\leq\iota/(d^4\sqrt{m_2 n})$. Altogether, with high probability (event $E_3$) on $\mathcal{D}$, we have

$$
|A_2|=\left|\frac{1}{n}\sum_{i=1}^n Y(\mathbf{x}_i)-\mathbb{E}_{\mathbf{x}}\left[Y(\mathbf{x})\right]\right|\leq\frac{2\iota^{p+2}}{d^4\sqrt{m_2 n}}.
$$

The proof is complete. $\qquad\square$

*Proof of Lemma 19.* We remember that

$$
A_3=\frac{1}{m_2}\sum_{j=1}^{m_2}w_j\mathbb{E}_{\mathbf{x}}\left[f^\star(\mathbf{x})K^{(0)}(\mathbf{x},\mathbf{x}')\sigma_2(\mathbf{v}_j^\top\mathbf{x})\right]=\frac{1}{m_2}\sum_{j=1}^{m_2}w_j h(\mathbf{v}_j,\mathbf{x}'),
$$

where $h(\mathbf{v},\mathbf{x}')=\mathbb{E}_{\mathbf{x}}\left[f^\star(\mathbf{x})K^{(0)}(\mathbf{x},\mathbf{x}')\sigma_2(\mathbf{v}^\top\mathbf{x})\right]$. To bound $h(\mathbf{v},\mathbf{x}')$ uniformly, we have the following lemma:

**Lemma 20.** *With high probability on $\mathbf{V}$ and the datasets $\mathcal{D}_1$ and $\mathcal{D}_2$, we have for any $j\in[m_2]$ and $\mathbf{x}'\in\mathcal{D}_2$,*

$$
|h(\mathbf{v}_j,\mathbf{x}')|\lesssim\frac{\iota^2\log^2(m_2 n_2)}{d^6}\cdot\left(\|\mathcal{P}_{>2}(f)\|_{L^2}+\sqrt{d}\,\|\mathcal{P}_2(f)\|_{L^2}\right)
$$

The proof of Lemma 20 is deferred to the end of this section. Thus, condition on the event above, by invoking the upper bound of Gaussian tail and uniformly bounding over $\mathbf{x}' \in \mathcal{D}_2$, we have with probability $1 - 2n \exp(-\iota^2/2)$ on $\mathbf{w}$, for any $\mathbf{x}' \in \mathcal{D}_2$, we have

$$|A_3| \leq \frac{\iota}{\sqrt{m_2}} \sqrt{\frac{1}{m_2} \sum_{j=1}^{m_2} h^2(\mathbf{v}_j, \mathbf{x}')} \lesssim \frac{\iota^3 \log^2(m_2 n_2)}{\sqrt{m_2} d^6} \cdot \left( \|\mathcal{P}_{>2}(f)\|_{L^2} + \sqrt{d} \|\mathcal{P}_2(f)\|_{L^2} \right).$$

Also, for a single point $\mathbf{x}'$, with probability $1 - 2 \exp(-\iota^2/2)$ on $\mathbf{w}$, we have

$$|A_3| \lesssim \frac{\iota^3 \log^2(m_2 n_2)}{\sqrt{m_2} d^6} \cdot \left( \|\mathcal{P}_{>2}(f)\|_{L^2} + \sqrt{d} \|\mathcal{P}_2(f)\|_{L^2} \right).$$

The proof is complete. $\qquad\square$

*Proof of Lemma 20.* Recall that the activation function $\sigma_2$ admits a Gegenbauer expansion

$$\sigma_2(t) = \sum_{i=2}^{\infty} c_i Q_i(t).$$

Let's fix $\mathbf{x}'$ and $\mathbf{v}$. Note that we can decompose $h(\mathbf{v}, \mathbf{x}')$ as

$$
\begin{aligned}
h(\mathbf{v}, \mathbf{x}') &= \mathbb{E}_{\mathbf{x}} \left[ f^\star(\mathbf{x}) K^{(0)}(\mathbf{x}, \mathbf{x}') \sigma_2(\mathbf{v}^\top \mathbf{x}) \right] \\
&= \mathbb{E}_{\mathbf{x}} \left[ f^\star(\mathbf{x}) \sum_{i=2}^{\infty} \frac{c_i^2 Q_i(\mathbf{x}^\top \mathbf{x}')}{B(d, i)} \sum_{j=2}^{\infty} c_j Q_j(\mathbf{x}^\top \mathbf{v}) \right] \\
&= \sum_{i=2}^{\infty} \sum_{j=2}^{\infty} \mathbb{E}_{\mathbf{x}} \left[ f^\star(\mathbf{x}) \frac{c_i^2}{B(d, i)^2} \langle \mathbf{Y}_i(\mathbf{x}), \mathbf{Y}_i(\mathbf{x}') \rangle \cdot \frac{c_j}{B(d, j)} \langle \mathbf{Y}_j(\mathbf{x}), \mathbf{Y}_j(\mathbf{v}) \rangle \right] \\
&= \sum_{i=2}^{\infty} \sum_{j=2}^{\infty} \frac{c_i^2 c_j}{B(d, i)^2 B(d, j)} \langle \mathbf{Y}_i(\mathbf{x}') \otimes \mathbf{Y}_j(\mathbf{v}), \mathbb{E}_{\mathbf{x}} [f^\star(\mathbf{x}) \mathbf{Y}_i(\mathbf{x}) \otimes \mathbf{Y}_j(\mathbf{x})] \rangle. \\
&=: \sum_{i=2}^{\infty} \sum_{j=2}^{\infty} \frac{c_i^2 c_j}{B(d, i)^2 B(d, j)} h_{i,j}(\mathbf{v}, \mathbf{x}').
\end{aligned}
$$

By the definition of $h_{i,j}(\mathbf{v}, \mathbf{x}')$, we have

$$
\begin{aligned}
& \mathbb{E}_{\mathbf{v}, \mathbf{x}'} \left[ h_{i,j}^2(\mathbf{v}, \mathbf{x}') \right] \qquad\qquad\qquad\qquad\qquad\qquad\qquad\qquad\qquad\qquad\qquad (30) \\
&= \mathbb{E}_{\mathbf{v}, \mathbf{x}'} \left[ \langle \mathbf{Y}_i(\mathbf{x}') \otimes \mathbf{Y}_j(\mathbf{v}), \mathbb{E}_{\mathbf{x}} [f^\star(\mathbf{x}) \mathbf{Y}_i(\mathbf{x}) \otimes \mathbf{Y}_j(\mathbf{x})] \rangle^2 \right] \\
&= \langle \mathbb{E}_{\mathbf{x}'} [f^\star(\mathbf{x}') \mathbf{Y}_i(\mathbf{x}') \otimes \mathbf{Y}_j(\mathbf{x}')], \mathbb{E}_{\mathbf{x}} [f^\star(\mathbf{x}) \mathbf{Y}_i(\mathbf{x}) \otimes \mathbf{Y}_j(\mathbf{x})] \rangle \\
&= B(d, i) B(d, j) \, \mathbb{E}_{\mathbf{x}, \mathbf{x}'} \left[ f(\mathbf{x}) f(\mathbf{x}') Q_i(\mathbf{x}^\top \mathbf{x}') Q_j(\mathbf{x}^\top \mathbf{x}') \right] \\
&= B(d, i) B(d, j) \, \mathbb{E}_{\mathbf{x}, \mathbf{x}'} \left[ f(\mathbf{x}) f(\mathbf{x}') \sum_{k=0}^{\min(i,j)} b_{i+j-2k}^{(i,j)} Q_{i+j-2k}(\mathbf{x}^\top \mathbf{x}') \right] \\
&= B(d, i) B(d, j) \, \mathbb{E}_{\mathbf{x}, \mathbf{x}'} \left[ f(\mathbf{x}) f(\mathbf{x}') \sum_{k=0}^{\min(i,j)} \frac{b_{i+j-2k}^{(i,j)}}{B(d, i+j-2k)} \langle \mathbf{Y}_{i+j-2k}(\mathbf{x}), \mathbf{Y}_{i+j-2k}(\mathbf{x}') \rangle \right] \\
&= B(d, i) B(d, j) \sum_{k=0}^{\min(i,j)} \frac{b_{i+j-2k}^{(i,j)}}{B(d, i+j-2k)} \|\mathbb{E}_{\mathbf{x}} [f(\mathbf{x}) \mathbf{Y}_{i+j-2k}(\mathbf{x})]\|_F^2 \\
&= B(d, i) B(d, j) \sum_{k=0}^{\min(i,j)} \frac{b_{i+j-2k}^{(i,j)}}{B(d, i+j-2k)} \|[\mathcal{P}_{i+j-2k}(f)]\|_{L^2}^2. \qquad\qquad\qquad\qquad\qquad\qquad (31)
\end{aligned}
$$

Since $h_{i,j}(\mathbf{v}, \mathbf{x}')$ is a degree $i$ polynomial of $\mathbf{x}'$ and a degree $j$ polynomial of $\mathbf{v}$, by Lemma 5, we have for any $q \geqslant 2$,

$$\mathbb{E}_{\mathbf{v},\mathbf{x}'}\left[|h_{i,j}(\mathbf{v},\mathbf{x}')|^q\right]^{2/q} \leq (q-1)^{i+j}\mathbb{E}_{\mathbf{v},\mathbf{x}'}\left[h_{i,j}(\mathbf{v},\mathbf{x}')^2\right]$$

Let $\delta = (2e\iota\log{(m_2 n_2)})^{(i+j)/2}$ for some $\iota > 1$, taking $q = 1 + e^{-1}\delta^{2/(i+j)}$ and Markov inequality, we have

$$\begin{aligned}
\Pr\left[|h_{i,j}(\mathbf{v},\mathbf{x}')| \geqslant \delta\sqrt{\mathbb{E}_{\mathbf{v},\mathbf{x}'}\left[h_{i,j}(\mathbf{v},\mathbf{x}')^2\right]}\right] &\leq \frac{\mathbb{E}_{\mathbf{v},\mathbf{x}'}\left[|h_{i,j}(\mathbf{v},\mathbf{x}')|^q\right]}{\left(\delta\sqrt{\mathbb{E}_{\mathbf{v},\mathbf{x}'}\left[h_{i,j}(\mathbf{v},\mathbf{x}')^2\right]}\right)^\infty} \\
&\leq (q-1)^{(i+j)q/2}\delta^{-q} \\
&= \exp\left(-\frac{i+j}{2}\left(1 + \frac{2e\iota\log{(m_2 n_2)}}{e}\right)\right) \\
&= (m_2 n_2)^{-\iota(i+j)}\exp(-(i+j)/2).
\end{aligned}$$

Thus, with probability at least $-(m_2 n_2)^{1-\iota(i+j)}\exp(-(i+j)/2)$,

$$\begin{aligned}
h_{i,j}(\mathbf{v}_i, \mathbf{x}') &\leq (2e\iota\log{(m_2 n_2)})^{(i+j)/2}\sqrt{\mathbb{E}_{\mathbf{v},\mathbf{x}'}\left[h_{i,j}(\mathbf{v},\mathbf{x}')^2\right]} \\
&\leq (2e\iota\log{(m_2 n_2)})^{(i+j)/2}\sqrt{\sum_{k=0}^{[(i+j)/2]}\frac{B(d,i)B(d,j)b_{i+j-2k}^{(i,j)}}{B(d,i+j-2k)}\|[\mathcal{P}_{i+j-2k}(f)]\|_{L^2}^2}.
\end{aligned}$$

In the second inequality we invoke (31). Summing over $i$ and $j$ gives rise to

$$\begin{aligned}
&|h(\mathbf{v}, \mathbf{x}')| \\
&= \left|\sum_{i=2}^\infty\sum_{j=2}^\infty \frac{c_i^2 c_j}{B(d,i)^2 B(d,j)}h_{i,j}(\mathbf{v},\mathbf{x}')\right| \\
&\leq \sum_{i=2}^\infty\sum_{j=2}^\infty \frac{c_i^2\,|c_j|\,(2e\iota\log{(m_2 n_2)})^{(i+j)/2}}{B(d,i)^2 B(d,j)}\sqrt{\sum_{k=0}^{[(i+j)/2]}\frac{B(d,i)B(d,j)b_{i+j-2k}^{(i,j)}}{B(d,i+j-2k)}\|[\mathcal{P}_{i+j-2k}(f)]\|_{L^2}^2} \\
&\leq \sqrt{\sum_{i=2}^\infty\sum_{j=2}^\infty \frac{c_i^2\,|c_j|\,(2e\iota\log{(m_2 n_2)})^{(i+j)/2}}{B(d,i)^2 B(d,j)^{1/2}}} \\
&\quad\cdot\sqrt{\sum_{i=2}^\infty\sum_{j=2}^\infty \frac{c_i^2\,|c_j|\,(2e\iota\log{(m_2 n_2)})^{(i+j)/2}}{B(d,i)^2 B(d,j)^{3/2}}\sum_{k=0}^{[(i+j)/2]}\frac{B(d,i)B(d,j)b_{i+j-2k}^{(i,j)}}{B(d,i+j-2k)}\|[\mathcal{P}_{i+j-2k}(f)]\|_{L^2}^2} \\
&\lesssim \frac{\iota\log(m_2 n_2)}{d^{5/2}}\cdot\sqrt{\sum_{\ell=0}^\infty\frac{1}{B(d,\ell)}\sum_{\substack{i+j-\ell\text{ even}}}^{2\leq i,j}\frac{c_i^2\,|c_j|\,b_\ell^{(i,j)}(2e\iota\log{(m_2 n_2)})^{(i+j)/2}}{B(d,i)B(d,j)^{1/2}}\|[\mathcal{P}_\ell(f)]\|_{L^2}^2}.
\end{aligned}$$

In the second inequality we invoke Cauchy inequality. Then by plugging the bound on $b_\ell^{(i,j)}$ in Lemma 12, we have

$$\begin{aligned}
&|h(\mathbf{v}, \mathbf{x}')| \\
&\lesssim \frac{\iota\log(m_2 n_2)}{d^{5/2}}\cdot\sqrt{\sum_{\ell=0}^\infty\frac{4(2\ell+d-2)}{B(d,\ell)(d-2)}\sum_{\substack{i+j-\ell=2k}}^{i,j\geqslant\max(k,2)}\frac{c_i^2\,|c_j|\,(2e\iota\log{(m_2 n_2)})^{(i+j)/2}}{B(d,i)B(d,j)^{1/2}(d-2)_k}\binom{i}{k}\binom{j}{k}k!\,\|[\mathcal{P}_\ell(f)]\|_{L^2}^2} \\
&\lesssim \frac{\iota\log(m_2 n_2)}{d^{5/2}}\cdot\left(\frac{\iota\log(m_2 n_2)}{d^{7/2}}\cdot\|\mathcal{P}_{>2}(f)\|_{L^2} + \frac{\iota\log(m_2 n_2)}{d^3}\cdot\|\mathcal{P}_2(f)\|_{L^2}\right) \\
&= \frac{\iota^2\log^2(m_2 n_2)}{d^6}\cdot\|\mathcal{P}_{>2}(f)\|_{L^2} + \frac{\iota^2\log^2(m_2 n_2)}{d^{11/2}}\cdot\|\mathcal{P}_2(f)\|_{L^2}.
\end{aligned}$$

The probability of this event is at least

$$1 - \sum_{i=2}^{\infty} \sum_{j=2}^{\infty} (m_2 n_2)^{-\iota(i+j)} \exp((i+j)/2) = 1 - \frac{m_2 n_2 \left(-(m_2 n_2)^{-\iota} e^{-1/2}\right)^2}{(m_2 n_2)^{4\iota} e^2},$$

which is a high probability event when uniformly bounding over $\mathbf{x}' \in \mathcal{D}_2$ and $\mathbf{v} = \mathbf{v}_1, \mathbf{v}_2 \ldots, \mathbf{v}_{m_2}$. The proof is complete. □

### C.3 PROOF OF PROPOSITION 1

#### C.3.1 THE FORMAL STATEMENT OF PROPOSITION 1 AND THE COROLLARY

Let's consider a formal version of Proposition 1. We remind the readers that throughout this section we denote $n = n_1$ for notation simplicity, since we only focus on the first training stage.

**Proposition 5** (Reconstruct the feature). *Suppose $m_2, n \geqslant Cd^4$ for some sufficiently large $C$. With high probability jointly on $\mathbf{V}$ and the training datasets $\mathcal{D}_1$ and $\mathcal{D}_2$, there exists a matrix $\mathbf{B}^\star \in \mathbb{R}^{r \times m_2}$ satisfying $\|\mathbf{B}^\star\|_{\mathrm{op}} \lesssim \frac{d^6}{\lambda_{\min}(\mathbf{H})} \sqrt{\frac{1}{m_2}}$ such that for any $\mathbf{x}' \in \mathcal{D}_2$, we have*

$$\left\| \mathbf{B}^\star \mathbf{h}^{(1)}(\mathbf{x}') - \mathbf{p}(\mathbf{x}') \right\|_2 \lesssim \frac{\sqrt{r}}{\lambda_{\min}(\mathbf{H})} \cdot \left( \frac{\iota^{p+2} d^5}{m_2} + \frac{\iota d^3}{\sqrt{m_2}} + \frac{\iota^{p+3/2} d}{\sqrt{n}} + \frac{\iota L r^2 \kappa_1 \log^2 d}{d^{1/6}} \right).$$

*Here $L \lesssim \iota r^{\frac{p-1}{2}}$ is a Lipschitz constant satisfying $\|\nabla g^\star(\mathbf{p}(\mathbf{x}))\|_2 \leq L$ with high probability.*

With the proposition above, we directly have the following result.

**Corollary 3.** *Under the same assumption in Proposition 5, with high probability, we have*

$$\sup_{\mathbf{x} \in \mathcal{D}_2} \left| g(\mathbf{B}^\star \mathbf{h}^{(1)}(\mathbf{x})) - g(\mathbf{p}(\mathbf{x})) \right| \lesssim \|g\|_{L^2} \cdot \frac{r^{p/2}}{\lambda_{\min}(\mathbf{H})} \cdot \left( \frac{\iota^{p+2} d^5}{m_2} + \frac{\iota d^3}{\sqrt{m_2}} + \frac{\iota^{p+3/2} d}{\sqrt{n}} + \frac{\iota L r^2 \kappa_1 \log^2 d}{d^{1/6}} \right).$$

We provide the main proof of Proposition 5 in Appendix C.3.2, and defer the proof of Corollary 3 and other supporting lemmas to Appendix C.3.3.

#### C.3.2 PROOF OF PROPOSITION 5

*Proof.* Denote the target features by $\mathbf{p}(\mathbf{v}) = [\mathbf{v}^\top \mathbf{A}_1 \mathbf{v}, \cdots, \mathbf{v}^\top \mathbf{A}_r \mathbf{v}]^\top \in \mathbb{R}^r$ for any $\mathbf{v} \in \mathbb{R}^d$, and we further let $\mathbf{P} = [\mathbf{p}(\mathbf{v}_1), \mathbf{p}(\mathbf{v}_2), \cdots, \mathbf{p}(\mathbf{v}_{m_2})]^\top \in \mathbb{R}^{m_2 \times r}$. Then for any $\mathbf{x}' \in \mathcal{D}_2$, we have the following decomposition

$$\frac{1}{m_2} \mathbf{P}^\top \mathbf{h}^{(1)}(\mathbf{x}') = \frac{1}{m_2 n} \sum_{i=1}^{n} \sum_{j=1}^{m_2} f^\star(\mathbf{x}_i) K_{m_2}^{(0)}(\mathbf{x}_i, \mathbf{x}') \sigma_2\left(\mathbf{v}_j^\top \mathbf{x}_i\right) \mathbf{p}(\mathbf{v}_j)$$

$$= \underbrace{\frac{1}{n} \sum_{i=1}^{n} \frac{1}{m_2} \sum_{j=1}^{m_2} f^\star(\mathbf{x}_i) \left( K_{m_2}^{(0)}(\mathbf{x}_i, \mathbf{x}') - K^{(0)}(\mathbf{x}_i, \mathbf{x}') \right) \sigma_2\left(\mathbf{v}_j^\top \mathbf{x}_i\right) \mathbf{p}(\mathbf{v}_j)}_{\mathbf{D}_{1,1}}$$

$$+ \underbrace{\frac{1}{n} \sum_{i=1}^{n} f^\star(\mathbf{x}_i) K^{(0)}(\mathbf{x}_i, \mathbf{x}') \left( \frac{1}{m_2} \sum_{j=1}^{m_2} \sigma_2\left(\mathbf{v}_j^\top \mathbf{x}_i\right) \mathbf{p}(\mathbf{v}_j) - \frac{c_2}{B(d,2)} \mathbf{p}(\mathbf{x}_i) \right)}_{\mathbf{D}_{1,2}}$$

$$+ \underbrace{\frac{c_2}{n B(d,2)} \left( \sum_{i=1}^{n} f^\star(\mathbf{x}_i) K^{(0)}(\mathbf{x}_i, \mathbf{x}') \mathbf{p}(\mathbf{x}_i) - \mathbb{E}_\mathbf{x}\left[ f^\star(\mathbf{x}) K^{(0)}(\mathbf{x}, \mathbf{x}') \mathbf{p}(\mathbf{x}_i) \right] \right)}_{\mathbf{D}_2}$$

$$+ \underbrace{\frac{c_2}{B(d,2)} \mathbb{E}_\mathbf{x}\left[ f^\star(\mathbf{x}) K^{(0)}(\mathbf{x}, \mathbf{x}') \mathbf{p}(\mathbf{x}) \right]}_{\mathbf{D}_3}.$$

We will derive an upper bound on the concentration error terms $\mathbf{D}_{1,1}$, $\mathbf{D}_{1,2}$ and $\mathbf{D}_2$, respectively. Moreover, leveraging the asymptotic analysis in Appendix C.1, we show that $\mathbf{D}_3 \approx d^{-6}\mathbf{Hp}(\mathbf{x}')$ with high probability,

**Lemma 21** (Bound $\mathbf{D}_{1,1}$ and $\mathbf{D}_{1,2}$). *Under the same assumptions in Proposition 5, with high probability on $\mathbf{V}$, $\mathcal{D}_1$ and $\mathcal{D}_2$, we have*

$$\|\mathbf{D}_{1,1}\|_\infty \leq \frac{9C_4^{1/4}\iota^{p+2}}{m_2 d} \quad and \quad \|\mathbf{D}_{1,2}\|_\infty \leq \frac{9\iota C_4^{1/4}C_2}{\sqrt{m_2}d^3}.$$

**Lemma 22** (Bound $\mathbf{D}_2$). *Under the same assumptions in Proposition 5, with high probability on $\mathcal{D}_1$ and $\mathcal{D}_2$, we have*

$$\|\mathbf{D}_2\|_\infty \lesssim \frac{\iota^{p+3/2}}{\sqrt{n}d^5}.$$

**Lemma 23** (Compute $\mathbf{D}_3$). *Under the same assumptions in Proposition 5, with high probability on $\mathcal{D}_2$, for any $\mathbf{x}' \in \mathcal{D}_2$, we have*

$$\left\|\mathbf{D}_3 - \frac{c_2^2}{B(d,2)^2 d(d-1)} \cdot \mathbf{Hp}(\mathbf{x}')\right\|_\infty \lesssim \frac{\iota L r^2 \kappa_1 \log^2 d}{d^{6+1/6}}.$$

We defer the detailed proof of the three lemmas to Appendix C.3.3. Combining all the results above and choosing

$$\mathbf{B}^\star = \frac{B(d,2)^2 d(d-1)}{c_2^2} \cdot \frac{1}{m_2}\mathbf{H}^{-1}\mathbf{P}^\top,$$

we have with high probability on $\mathbf{V}$, $\mathcal{D}_1$ and $\mathcal{D}_2$,

$$
\begin{aligned}
&\left\|\mathbf{B}^\star \mathbf{h}^{(1)}(\mathbf{x}') - \mathbf{p}(\mathbf{x}')\right\|_2 \\
&\leq \frac{B(d,2)^2 d(d-1)}{c_2^2} \cdot \left\|\mathbf{H}^{-1}\left(\mathbf{D}_{1,1} + \mathbf{D}_{1,2} + \mathbf{D}_2 + \mathbf{D}_3 - \frac{c_2^2}{B(d,2)d(d-1)} \cdot \mathbf{Hp}(\mathbf{x}')\right)\right\|_2 \\
&\lesssim \frac{d^6\sqrt{r}}{\lambda_{\min}(\mathbf{H})} \cdot \Bigg(\|\mathbf{D}_{1,1}\|_\infty + \|\mathbf{D}_{1,2}\|_\infty + \|\mathbf{D}_2\|_\infty \\
&\qquad\qquad + \left\|\mathbf{D}_3 - \frac{c_2^2}{B(d,2)d(d-1)} \cdot \mathbf{Hp}(\mathbf{x}')\right\|_\infty\Bigg) \\
&\lesssim \frac{\sqrt{r}}{\lambda_{\min}(\mathbf{H})} \cdot \left(\frac{\iota^{p+2}d^5}{m_2} + \frac{\iota d^3}{\sqrt{m_2}} + \frac{\iota^{p+3/2}d}{\sqrt{n}} + \frac{\iota L r^2 \kappa_1 \log^2 d}{d^{1/6}}\right),
\end{aligned}
$$

To bound $\|\mathbf{B}^\star\|_{\mathrm{op}}$, note that

$$
\begin{aligned}
\|\mathbf{B}^\star\|_{\mathrm{op}}^2 &= \left\|\mathbf{B}^\star \mathbf{B}^{\star\top}\right\|_{\mathrm{op}} \\
&\lesssim \frac{d^{12}}{m_2^2 \lambda_{\min}^2(\mathbf{H})}\left\|\mathbf{PP}^\top\right\|_{\mathrm{op}} \\
&= \frac{d^{12}}{m_2 \lambda_{\min}^2(\mathbf{H})}\left\|\frac{1}{m_2}\sum_{j=1}^{m_2}\mathbf{p}(\mathbf{v}_j)\mathbf{p}(\mathbf{v}_j)^\top\right\|_{\mathrm{op}}.
\end{aligned}
$$

Moreover, for any $j \in [m_2]$, we have

$$\left\|\mathbf{p}(\mathbf{v}_j)\mathbf{p}(\mathbf{v}_j)^\top\right\|_{\mathrm{op}} = \|\mathbf{p}(\mathbf{v}_j)\|_2^2 = \sum_{k=1}^{r}(\mathbf{v}_j^\top \mathbf{A}_k \mathbf{v}_j)^2 \lesssim rd^2,$$

and we have

$$
\begin{aligned}
\left\| \mathbb{E}_{\mathbf{v}} \left[ \left( \mathbf{p}(\mathbf{v}) \mathbf{p}(\mathbf{v})^\top \right)^2 \right] \right\|_{\mathrm{op}} &= \left\| \mathbb{E}_{\mathbf{v}} \left[ \sum_{k=1}^r (\mathbf{v}^\top \mathbf{A}_k \mathbf{v})^2 \mathbf{p}(\mathbf{v}) \mathbf{p}(\mathbf{v})^\top \right] \right\|_{\mathrm{op}} \\
&\leq \sum_{k=1}^r \left\| \mathbb{E}_{\mathbf{v}} \left[ (\mathbf{v}^\top \mathbf{A}_k \mathbf{v})^2 \mathbf{p}(\mathbf{v}) \mathbf{p}(\mathbf{v})^\top \right] \right\|_{\mathrm{op}} \\
&\leq \sum_{k=1}^r d^2 \left\| \mathbb{E}_{\mathbf{v}} \left[ \mathbf{p}(\mathbf{v}) \mathbf{p}(\mathbf{v})^\top \right] \right\|_{\mathrm{op}} \\
&= r d^2.
\end{aligned}
$$

The second inequality holds because $\mathbf{p}(\mathbf{v}) \mathbf{p}(\mathbf{v})^\top$ is positive semi-definite. By Matrix Bernstein Inequality, we have

$$
\Pr \left[ \left\| \frac{1}{m_2} \sum_{j=1}^{m_2} \mathbf{p}(\mathbf{v}_j) \mathbf{p}(\mathbf{v}_j)^\top - \mathbf{I} \right\|_{\mathrm{op}} \geqslant 1 + \frac{\sqrt{r} d \iota}{\sqrt{m_2}} \right] \leq \exp \left( -\frac{\frac{r d^2 \iota^2}{2 m_2}}{\frac{r d^2}{m_2} + \frac{r d^2}{3 m_2} \cdot \frac{\sqrt{r} d \iota}{\sqrt{m_2}}} \right)
$$

$$
= \exp \left( -\frac{\frac{\iota^2}{2}}{1 + \frac{\sqrt{r} d \iota}{3 \sqrt{m_2}}} \right).
$$

Thus, when $m_2 \geqslant d^4$, we know with high probability on $\mathbf{V}$,

$$
\left\| \frac{1}{m_2} \sum_{j=1}^{m_2} \mathbf{p}(\mathbf{v}_j) \mathbf{p}(\mathbf{v}_j)^\top \right\|_{\mathrm{op}} \leq 1 + \frac{\sqrt{r} d \iota}{\sqrt{m_2}} \lesssim 1.
$$

Thus, we have $\|\mathbf{B}^\star\|_{\mathrm{op}} \lesssim \frac{d^6}{\lambda_{\min}(\mathbf{H})} \sqrt{\frac{1}{m_2}}$. The proof is complete. $\qquad \square$

### C.3.3 Omitted Proofs in Appendices C.3.1 and C.3.2

*Proof of Lemma 21.* Let's first bound $\mathbf{D}_{1,1}$. We can rewrite $\mathbf{D}_{1,1}$ as

$$
\mathbf{D}_{1,1} = \frac{1}{n} \sum_{i=1}^n f^\star(\mathbf{x}_i) \left( K_{m_2}^{(0)}(\mathbf{x}_i, \mathbf{x}') - K^{(0)}(\mathbf{x}_i, \mathbf{x}') \right) \frac{1}{m_2} \sum_{j=1}^{m_2} \sigma_2 \left( \mathbf{v}_j^\top \mathbf{x}_i \right) \mathbf{p}(\mathbf{v}_j)
$$

By Lemma 25, for any $k \in [r]$, we have with high probability on $\mathbf{V}$

$$
\left| \frac{1}{m_2} \sum_{j=1}^{m_2} (\mathbf{v}_j^\top \mathbf{A}_k \mathbf{v}_j) \sigma_2(\mathbf{v}_j^\top \mathbf{x}_i) - \frac{c_2}{B(d, 2)} \mathbf{x}_i^\top \mathbf{A}_k \mathbf{x}_i \right| \leq \frac{9 \iota d^{-1} C_4^{1/4}}{\sqrt{m_2}}.
$$

Thus, by enumerating $\mathbf{A}_k$ over $\{\mathbf{A}_1, \mathbf{A}_2, \dots, \mathbf{A}_r\}$, we have with high probability on $\mathbf{V}$,

$$
\left\| \frac{1}{m_2} \sum_{j=1}^{m_2} \mathbf{p}(\mathbf{v}_j) \sigma_2(\mathbf{v}_j^\top \mathbf{x}_i) - \frac{c_2}{B(d, 2)} \mathbf{p}(\mathbf{x}_i) \right\|_{\infty} \leq \frac{9 \iota d^{-1} C_4^{1/4}}{\sqrt{m_2}}.
$$

On the other hand, by Lemma 24, with high probability on $\mathbf{V}$, we have for any $\mathbf{x}_i \in \mathcal{D}_1, \mathbf{x}' \in \mathcal{D}_2$,

$$
\left| K_{m_2}^{(0)}(\mathbf{x}_i, \mathbf{x}') - K^{(0)}(\mathbf{x}_i, \mathbf{x}') \right| \leq \frac{\iota}{\sqrt{m_2} d^2}. \tag{32}
$$

Moreover, under the event $E_3$ (defined in (29)), with high probability on the dataset $\mathcal{D}_1$, we have $|f(\mathbf{x})| \leq \iota^p$ for any $\mathbf{x} \in \mathcal{D}_1$. Thus, altogther we have

$$
\|\mathbf{D}_{1,1}\|_{\infty} \leq \iota^p \cdot \frac{\iota}{\sqrt{m_2} d^2} \cdot \frac{9 \iota d^{-1} C_4^{1/4}}{\sqrt{m_2}} = \frac{9 C_4^{1/4} \iota^{p+2}}{m_2 d}.
$$

with high probability. To bound $\mathbf{D}_{1,2}$, from the proof above, we know with high probability,

$$\left\| \frac{1}{m_2} \sum_{j=1}^{m_2} \mathbf{p}(\mathbf{v}_j)\sigma_2(\mathbf{v}_j^\top \mathbf{x}_i) - \frac{c_2}{B(d,2)}\mathbf{p}(\mathbf{x}_i) \right\|_\infty \leq \frac{9\iota d^{-1} C_4^{1/4}}{\sqrt{m_2}}.$$

Moreover, for any $\mathbf{x} \in \mathcal{D}_1$ and $\mathbf{x}' \in \mathcal{D}_2$,

$$K^{(0)}(\mathbf{x},\mathbf{x}') = \mathbb{E}_\mathbf{v}\left[\sigma_2(\mathbf{v}^\top \mathbf{x})\sigma_2(\mathbf{v}^\top \mathbf{x}')\right] \leq \sqrt{\mathbb{E}_\mathbf{v}\left[\sigma_2(\mathbf{v}^\top \mathbf{x})^2\right]\mathbb{E}_\mathbf{v}\left[\sigma_2(\mathbf{v}^\top \mathbf{x}')^2\right]} \leq \frac{C_2}{d^2}. \tag{33}$$

Thus, we can bound $\mathbf{D}_{1,2}$ with high probability by

$$\|\mathbf{D}_{1,2}\|_\infty \leq \frac{C_2}{d^2} \cdot \frac{9\iota d^{-1} C_4^{1/4}}{\sqrt{m_2}} = \frac{9\iota C_4^{1/4} C_2}{\sqrt{m_2}d^3}.$$

The proof is complete. $\qquad\qquad\qquad\qquad\qquad\qquad\qquad\qquad\qquad\qquad\qquad\qquad\square$

*Proof of Lemma 22.* Thus, let's focus on the concentration of a single element

$$Y_k(\mathbf{x}) = f^\star(\mathbf{x})K^{(0)}(\mathbf{x},\mathbf{x}')\mathbf{x}^\top \mathbf{A}_k \mathbf{x}, \quad k = 1,2,\ldots,r.$$

Similar to the proof of Lemma 17, we denote a truncated version of $Y_k$ by

$$\widetilde{Y}_k(\mathbf{x}) = f^\star(\mathbf{x})\mathbf{1}\{f^\star(\mathbf{x}) \leq R\}K^{(0)}(\mathbf{x},\mathbf{x}')\mathbf{x}^\top \mathbf{A}_k \mathbf{x}, \quad k = 1,2,\ldots,r.$$

Here, $R = (C\eta)^p$ for some large constant $C$. Now, we decompose the concentration error as

$$\frac{1}{n}\sum_{i=1}^n Y_k(\mathbf{x}_i) - \mathbb{E}_\mathbf{x}\left[Y_k(\mathbf{x})\right] = \underbrace{\frac{1}{n}\sum_{i=1}^n \left(Y_k(\mathbf{x}_i) - \widetilde{Y}_k(\mathbf{x}_i)\right)}_{\mathcal{L}_0} + \underbrace{\frac{1}{n}\sum_{i=1}^n \left(\widetilde{Y}_k(\mathbf{x}_i) - \mathbb{E}_{x_i}\left[\widetilde{Y}_k(\mathbf{x}_i)\right]\right)}_{\mathcal{L}_1}$$

$$+ \underbrace{\frac{1}{n}\sum_{i=1}^n \left(\mathbb{E}_{x_i}\left[\widetilde{Y}_k(\mathbf{x}_i)\right] - \mathbb{E}_{x_i}\left[Y_k(\mathbf{x}_i)\right]\right)}_{\mathcal{L}_2}.$$

By (29), we know with probability at least $1 - 2ne^{-2\eta}$, $\mathcal{L}_0 = 0$.

**Bounding $\mathcal{L}_1$.** First we derive a uniform upper bound of $\widetilde{Y}_k(\mathbf{x})$, which is

$$|Y_k(\mathbf{x})| \leq R\left|K^{(0)}(\mathbf{x},\mathbf{x}')\mathbf{x}^\top \mathbf{A}_k \mathbf{x}\right|$$

$$\leq R\left|K^{(0)}(\mathbf{x},\mathbf{x}')\right|\left|\mathbf{x}^\top \mathbf{A}_k \mathbf{x}\right|$$

$$\leq R \cdot \frac{C_2}{d^2}d\|\mathbf{A}_k\|_{op}$$

$$= \frac{RC_2\|\mathbf{A}_k\|_{\text{op}}}{d}.$$

Then, we bound the second moments of $\widetilde{Y}_k(\mathbf{x}) - \mathbb{E}_\mathbf{x}\left[\widetilde{Y}_k(\mathbf{x})\right]$. Again by Lemma 8, we know that there exists a sufficient large constant $C > 0$ s.t. $\Pr\left[\left|\mathbf{x}^\top \mathbf{A}_k \mathbf{x}\right| \geqslant C\iota\right] \leq 2\exp(-\iota)$. By taking $\iota \geqslant 2\log d$, we have

$$\text{Var}\left[\widetilde{Y}_k(\mathbf{x})\right] \leq \mathbb{E}_\mathbf{x}\left[\widetilde{Y}_k^2(\mathbf{x})\right]$$

$$= \mathbb{E}_\mathbf{x}\left[\widetilde{Y}_k^2(\mathbf{x})\mathbf{1}\left\{\left|\mathbf{x}^\top \mathbf{A}_k \mathbf{x}\right| \leq C\iota\right\}\right] + \mathbb{E}_\mathbf{x}\left[\widetilde{Y}_k^2(\mathbf{x})\mathbf{1}\left\{\left|\mathbf{x}^\top \mathbf{A}_k \mathbf{x}\right| > C\iota\right\}\right]$$

$$\leq C^2 r^2 \kappa_1 \iota^2 \mathbb{E}_\mathbf{x}\left[\left(K^{(0)}(\mathbf{x},\mathbf{x}')\right)^2\right] + \frac{r^2 \kappa_1 C_2^2}{d^4} \cdot \mathbb{E}_\mathbf{x}\left[\mathbf{1}\left\{\left|\mathbf{x}^\top \mathbf{A}_k \mathbf{x}\right| > C\iota\right\}\right]$$

$$\leq C^2 r^2 \kappa_1 \iota^2 \cdot \sum_{i=2}^\infty \frac{c_i^4}{B(d,i)^3} + \frac{2r^2 \kappa_1 C_2^2 \exp(-\iota)}{d^4}$$

$$\lesssim \frac{C' r^2 \kappa_1 \iota^2}{d^6}.$$

Here $C'$ is a sufficiently large constant independent of $d$. We invoke (33) in the second inequality. Thus, by Bernstein's inequality, we have

$$\Pr\left[|\mathcal{L}_1| \geqslant \frac{R\iota}{d^3}\sqrt{\frac{C'\iota}{n}}\right] \leq 2\exp\left(\frac{-\frac{\iota^3 C' r^2 \kappa_1}{2nd^6}}{\frac{C' r^2 \kappa_1 \iota^2}{nd^6} + \frac{RC_2\|\mathbf{A}_k\|_{\mathrm{op}}}{3nd}\sqrt{\frac{r^2 \kappa_1 \iota}{nd^6}}}\right)$$

$$= 2\exp\left(\frac{-\frac{\iota}{2}}{1 + \frac{C_2\|\mathbf{A}_k\|_{\mathrm{op}}}{3C'}\sqrt{\frac{d^4}{n\iota^3}}}\right).$$

Thus, when $n \geqslant C_2^2 \|\mathbf{A}_k\|_{\mathrm{op}}^2 d^4$, we have with high probability on the training dataset $\mathcal{D}_1$,

$$|\mathcal{L}_1| \leq \frac{R\iota}{d^3}\sqrt{\frac{C'\iota}{n}} \lesssim \frac{R\iota^{3/2}}{\sqrt{n}d^3}.$$

**Bounding $\mathcal{L}_2$.** It suffices to bound

$$\left|\mathbb{E}_{\mathbf{x}}\left[\widetilde{Y}_k(\mathbf{x})\right] - \mathbb{E}_{\mathbf{x}}\left[Y_k(\mathbf{x})\right]\right| \leq \mathbb{E}_{\mathbf{x}}\left[|f^\star(\mathbf{x})|\,\mathbf{1}\{f^\star(\mathbf{x}) > R\}\left|K^{(0)}(x,x')\mathbf{x}^\top\mathbf{A}_k\mathbf{x}\right|\right]$$

$$\leq \mathbb{E}_{\mathbf{x}}\left[(f^\star(\mathbf{x}))^2\right]^{\frac{1}{2}}\Pr\left[f^\star(\mathbf{x}) > R\right]^{\frac{1}{4}}\mathbb{E}_{\mathbf{x}}\left[K^{(0)}(\mathbf{x},\mathbf{x}')^8\right]^{\frac{1}{8}}\mathbb{E}_{\mathbf{x}}\left[(\mathbf{x}^\top\mathbf{A}_k\mathbf{x})^8\right]^{\frac{1}{8}}$$

$$\leq 1 \cdot \exp(-\eta/2) \cdot \frac{C_2}{d^2} \cdot (8-1)$$

$$= \frac{7C_2\exp(-\eta/2)}{d^2}.$$

Here we invoke (33) and Lemma 8 in the last inequality. By taking $\eta = \iota \geqslant 2\log n + 8\log d$, we can ensure that with high probability, we have

$$\left|\frac{1}{n}\sum_{i=1}^n Y_k(\mathbf{x}_i) - \mathbb{E}_{\mathbf{x}}\left[Y_k(\mathbf{x})\right]\right| \leq |\mathcal{L}_1| + |\mathcal{L}_2| \lesssim \frac{R\iota^{3/2}}{\sqrt{n}d^3}.$$

Thus, by taking $k$ over $[r]$, we have with high probability over the training set $\mathcal{D}_1$, we have

$$\|\mathbf{D}_2\|_\infty \leq \frac{|c_2|}{B(d,2)} \cdot \frac{R\iota^{3/2}}{\sqrt{n}d^3} \lesssim \frac{R\iota^{3/2}}{\sqrt{n}d^5} \lesssim \frac{\iota^{p+3/2}}{\sqrt{n}d^5}.$$

The proof is complete.

$\square$

*Proof of Lemma 23.* Note that for any $k \in [r]$, we have

$$\mathbb{E}_{\mathbf{x}}\left[f^\star(\mathbf{x})K^{(0)}(\mathbf{x},\mathbf{x}')\mathbf{x}^\top\mathbf{A}_k\mathbf{x}\right]$$

$$= \mathbb{E}_{\mathbf{x}}\left[f^\star(\mathbf{x})\sum_{i=2}^\infty \frac{c_i^2 Q_i(\mathbf{x}^\top\mathbf{x}')}{B(d,i)}\cdot\mathbf{x}^\top\mathbf{A}_k\mathbf{x}\right]$$

$$= \sum_{i=2}^\infty \frac{c_i^2}{B(d,i)}\cdot\mathbb{E}_{\mathbf{x}}\left[f^\star(\mathbf{x})Q_i(\mathbf{x}^\top\mathbf{x}')\mathbf{x}^\top\mathbf{A}_k\mathbf{x}\right]$$

$$= \frac{c_2^2}{B(d,2)d(d-1)}\cdot\left\langle\mathbb{E}_{\mathbf{x}}\left[f^\star(\mathbf{x})(\mathbf{x}^\top\mathbf{A}_k\mathbf{x})(\mathbf{x}\mathbf{x}^\top - \mathbf{I})\right], \mathbf{x}'\mathbf{x}'^\top - \mathbf{I}\right\rangle$$

$$+ \sum_{i=3}^\infty \frac{c_2^2}{B(d,i)}\cdot\mathbb{E}_{\mathbf{x}}\left[f^\star(\mathbf{x})Q_i(\mathbf{x}^\top\mathbf{x}')\mathbf{x}^\top\mathbf{A}_k\mathbf{x}\right]$$

$$= \frac{c_i^2}{B(d,2)d(d-1)}\cdot\left\langle T(\mathbf{A}_k), \mathbf{x}'\mathbf{x}'^\top - \mathbf{I}\right\rangle + \sum_{i=3}^\infty \frac{c_i^2}{B(d,i)}\cdot\mathbb{E}_{\mathbf{x}}\left[f^\star(\mathbf{x})Q_i(\mathbf{x}^\top\mathbf{x}')\mathbf{x}^\top\mathbf{A}_k\mathbf{x}\right].$$

Here $T$ is the linear operator defined in (14). Recall by Proposition 3, we have

$$\left\| T(\mathbf{A}_k) - \sum_{j=1}^{r} \mathbf{H}_{k,j}\mathbf{A}_j \right\|_{\mathrm{F}} \lesssim d^{-1/6}Lr^2\kappa_1 \log^2 d.$$

Let's denote $\mathbf{R}_k = T(\mathbf{A}_k) - \sum_{j=1}^{r} \mathbf{H}_{k,j}\mathbf{A}_j$ so that $\|\mathbf{R}_k\|_{\mathrm{F}} \lesssim d^{-1/6}Lr^2\kappa_1 \log^2 d$. Since $\langle \mathbf{R}_k, \mathbf{x}'\mathbf{x}'^\top - \mathbf{I}\rangle$ is a quadratic function of $\mathbf{x}'$, and $\mathbb{E}_{\mathbf{x}'}\left[\langle \mathbf{R}_k, \mathbf{x}'\mathbf{x}'^\top - \mathbf{I}\rangle^2\right] = \frac{2d}{d+2}\|\mathbf{R}_k\|_{\mathrm{F}}^2$. By Lemma 8, there exists a constant $C > 0$ such that

$$\Pr\left[\left|\langle \mathbf{R}_k, \mathbf{x}'\mathbf{x}'^\top - \mathbf{I}\rangle\right| \geqslant C\iota\sqrt{\mathbb{E}_{\mathbf{x}'}\left[\langle \mathbf{R}_k, \mathbf{x}'\mathbf{x}'^\top - \mathbf{I}\rangle^2\right]}\right] \leq 2\exp(-\iota).$$

Thus, by enumerating $k \in [r]$ and $\mathbf{x}' \in \mathcal{D}_2$, we obtain that with high probability $(1 - nr\exp(-\iota))$ on $\mathcal{D}_2$, for any $k \in [r]$, we have

$$\left|\langle T(\mathbf{A}_k), \mathbf{x}'\mathbf{x}'^\top - \mathbf{I}\rangle - \sum_{j=1}^{r} H_{k,j}\mathbf{x}'^\top \mathbf{A}_j \mathbf{x}'\right| \lesssim \frac{\iota Lr^2\kappa_1 \log^2 d}{d^{1/6}}.$$

Moreover, we have for any $\mathbf{x}'$

$$\left|\sum_{i=3}^{\infty} \frac{c_i^2}{B(d,i)} \cdot \mathbb{E}_{\mathbf{x}}\left[f^\star(\mathbf{x})Q_i(\mathbf{x}^\top\mathbf{x}')\mathbf{x}^\top\mathbf{A}_k\mathbf{x}\right]\right|$$

$$\leq \sum_{i=3}^{\infty} \frac{c_i^2}{B(d,i)} \cdot \left|\mathbb{E}_{\mathbf{x}}\left[f^\star(\mathbf{x})Q_i(\mathbf{x}^\top\mathbf{x}')\mathbf{x}^\top\mathbf{A}_k\mathbf{x}\right]\right|$$

$$\leq \sum_{i=3}^{\infty} \frac{c_i^2}{B(d,i)} \cdot \sqrt{\mathbb{E}_{\mathbf{x}}\left[Q_i(\mathbf{x}^\top\mathbf{x}')^2\right]\mathbb{E}_{\mathbf{x}}\left[f^\star(\mathbf{x})^2(\mathbf{x}^\top\mathbf{A}_k\mathbf{x})^2\right]}$$

$$\leq \sum_{i=3}^{\infty} \frac{c_i^2}{B(d,i)^{3/2}} \cdot \sqrt{\mathbb{E}_{\mathbf{x}}\left[f^\star(\mathbf{x})^2(\mathbf{x}^\top\mathbf{A}_k\mathbf{x})^2\right]}.$$

Again by Lemma 8, we know that there exists a sufficient large constant $C > 0$ s.t. $\Pr\left[\left|\mathbf{x}^\top\mathbf{A}_k\mathbf{x}\right| \geqslant C\iota\right] \leq 2\exp(-\iota)$. By taking $\iota \geqslant (2p+2)\log d$, we have

$$\mathbb{E}_{\mathbf{x}}\left[f^\star(\mathbf{x})^2(\mathbf{x}^\top\mathbf{A}_k\mathbf{x})^2\right] = \mathbb{E}_{\mathbf{x}}\left[f^\star(\mathbf{x})^2(\mathbf{x}^\top\mathbf{A}_k\mathbf{x})^2\mathbf{1}\{\left|\mathbf{x}^\top\mathbf{A}_k\mathbf{x}\right| \leq C\iota\}\right]$$

$$+ \mathbb{E}_{\mathbf{x}}\left[f^\star(\mathbf{x})^2(\mathbf{x}^\top\mathbf{A}_k\mathbf{x})^2\mathbf{1}\{\left|\mathbf{x}^\top\mathbf{A}_k\mathbf{x}\right| > C\iota\}\right]$$

$$\lesssim C^2\iota^2 + 2d^{2p+2}\exp(-\iota)$$

$$\lesssim C^2\iota^2.$$

Altogether, with high probability on $\mathcal{D}_2$, for any $k \in [r]$, we have

$$\left|\mathbb{E}_{\mathbf{x}}\left[f^\star(\mathbf{x})K^{(0)}(\mathbf{x},\mathbf{x}')\mathbf{x}^\top\mathbf{A}_k\mathbf{x}\right] - \frac{c_2^2}{B(d,2)d(d-1)} \cdot \sum_{j=1}^{r} H_{k,j}\mathbf{x}'^\top \mathbf{A}_j\mathbf{x}'\right|$$

$$\leq \frac{\iota Lr^2\kappa_1 \log^2 d}{B(d,2)d^{7/6}(d-1)} + \sum_{i=3}^{\infty} \frac{C\iota c_i^2}{B(d,i)^{3/2}}.$$

Thus, by paralleling the $r$ entries together, we have with high probability on $\mathcal{D}_2$

$$\left\|\mathbf{D}_3 - \frac{c_2^2}{B(d,2)^2 d(d-1)} \cdot \mathbf{H}\mathbf{p}(\mathbf{x}')\right\|_\infty \leq \frac{\iota Lr^2\kappa_1 \log^2 d}{B(d,2)^2 d^{7/6}(d-1)} + \sum_{i=3}^{\infty} \frac{C\iota c_i^2}{B(d,2)B(d,i)^{3/2}}$$

$$\lesssim \frac{\iota Lr^2\kappa_1 \log^2 d}{d^{6+1/6}}.$$

The proof is complete. $\qquad\square$

*Proof of Lemma 3.* By the mean value theorem, we have

$$\left| g(\mathbf{B}^\star \mathbf{h}^{(1)}(\mathbf{x})) - g(\mathbf{p}(\mathbf{x})) \right| \lesssim \sup_{\lambda \in [0,1]} \left\| \nabla g(\lambda \mathbf{B}^\star \mathbf{h}^{(1)}(\mathbf{x}) + (1-\lambda)\mathbf{p}(\mathbf{x})) \right\|_2 \left\| \mathbf{B}^\star \mathbf{h}^{(1)}(\mathbf{x}) - \mathbf{p}(\mathbf{x}) \right\|_2.$$

Recall by (8), we have $\|\nabla g(\mathbf{z})\|_2 \lesssim \|g\|_{L^2} \sum_{k=1}^p r^{\frac{p-k}{4}} \|\mathbf{z}\|_2^{k-1}$. Note that with high probability, $\sup_{\mathbf{x} \in \mathcal{D}_2} \|\mathbf{p}(\mathbf{x})\| \leq \widetilde{O}(\sqrt{r})$. Therefore

$$\sup_{\mathbf{x} \in \mathcal{D}_2} \sup_{\lambda \in [0,1]} \left\| \nabla g(\lambda \mathbf{B}^\star \mathbf{h}^{(1)}(\mathbf{x}) + (1-\lambda)\mathbf{p}(\mathbf{x})) \right\| \lesssim \|g\|_{L^2} \, r^{\frac{p-1}{2}}.$$

Altogether, by Proposition 5,

$$\sup_{\mathbf{x} \in \mathcal{D}_2} \left| g(\mathbf{B}^\star \mathbf{h}^{(1)}(\mathbf{x})) - g(\mathbf{p}(\mathbf{x})) \right|$$

$$\lesssim \|g\|_{L^2} \, r^{\frac{p-1}{2}} \left\| \mathbf{B}^\star \mathbf{h}^{(1)}(\mathbf{x}) - \mathbf{p}(\mathbf{x}) \right\|_2$$

$$\leq \|g\|_{L^2} \cdot \frac{r^{p/2}}{\lambda_{\min}(\mathbf{H})} \cdot \left( \frac{\iota^{p+2} d^5}{m_2} + \frac{\iota d^3}{\sqrt{m_2}} + \frac{\iota^{p+3/2} d}{\sqrt{n}} + \frac{\iota L r^2 \kappa_1 \log^2 d}{d^{1/6}} \right)$$

The proof is complete. $\qquad\qquad\qquad\qquad\qquad\qquad\qquad\qquad\qquad\qquad\qquad\qquad\qquad\qquad\square$

## C.4 PROOF OF OTHER SUPPORTING LEMMAS

We first present the concentration of the initial kernel $K_{m_2}^{(0)}(\mathbf{x}, \mathbf{x}')$.

**Lemma 24.** *Let $K_{m_2}^{(0)}(\mathbf{x}, \mathbf{x}') = \frac{1}{m_2} \langle \sigma_2(\mathbf{V}\mathbf{x}), \sigma_2(\mathbf{V}\mathbf{x}') \rangle$ be the initial kernel with inner width being $m_2$, and $K^{(0)}(\mathbf{x}, \mathbf{x}') = \mathbb{E}_{\mathbf{v} \sim Unif\text{-}\mathbb{S}^{d-1}(\sqrt{d})} \left[ \sigma_2(\mathbf{v}^\top \mathbf{x}) \sigma_2(\mathbf{v}^\top \mathbf{x}') \right]$ be the infinite-width kernel. Then there exists a constant $C$ s.t. when $m_2 \geq C d^4$, with high probability probability on $\mathbf{w}$, $\mathbf{V}$ and the training dataset $\mathcal{D}$, for any $\mathbf{x} \in \mathcal{D}_1$ and $\mathbf{x}' \in \mathcal{D}_2$, we have*

$$\left| K_{m_2}^{(0)}(\mathbf{x}, \mathbf{x}') - K^{(0)}(\mathbf{x}, \mathbf{x}') \right| \leq \frac{\iota}{\sqrt{m_2} d^2}.$$

*Proof of Lemma 24.* By Assumption 4, for any $\mathbf{x}, \mathbf{x}' \in \mathcal{D}$ and $\mathbf{v} \in \mathbb{S}^{d-1}(\sqrt{d})$, we have

$$\left| \sigma_2(\mathbf{v}^\top \mathbf{x}) \sigma_2(\mathbf{v}^\top \mathbf{x}') \right| \leq C_\sigma^2$$

and

$$\mathbb{E}_{\mathbf{v}} \left[ \sigma_2(\mathbf{v}^\top \mathbf{x})^2 \sigma_2(\mathbf{v}^\top \mathbf{x}')^2 \right] \leq \sqrt{\mathbb{E}_{\mathbf{v}} \left[ \sigma_2(\mathbf{v}^\top \mathbf{x})^4 \right] \mathbb{E}_{\mathbf{v}} \left[ \sigma_2(\mathbf{v}^\top \mathbf{x}')^4 \right]} \leq \frac{C_4}{d^4}.$$

Thus, by Bernstein inequality, we have

$$\Pr\left[ \left| K_{m_2}^{(0)}(\mathbf{x}, \mathbf{x}') - K^{(0)}(\mathbf{x}, \mathbf{x}') \right| \geq \sqrt{\frac{t}{m_2}} \right] \leq 2\exp\left( \frac{-\frac{t}{2m_2}}{\frac{C_4}{m_2 d^4} + \frac{C_\sigma^2}{3m_2}\sqrt{\frac{t}{m_2}}} \right)$$

$$= \exp\left( \frac{-t/2}{\frac{C_4}{d^4} + \frac{C_\sigma^2}{3}\sqrt{\frac{t}{m_2}}} \right). \qquad (34)$$

By enumerating $\mathbf{x}, \mathbf{x}'$ over $\mathcal{D}$, we have

$$\Pr\left[ \max_{\mathbf{x}, \mathbf{x}' \in \mathcal{D}} \left| K_{m_2}^{(0)}(\mathbf{x}, \mathbf{x}') - K^{(0)}(\mathbf{x}, \mathbf{x}') \right| \geq \sqrt{\frac{t}{m_2}} \right] \leq n^2 \exp\left( \frac{-t/2}{\frac{C_4}{d^4} + \frac{C_\sigma^2}{3}\sqrt{\frac{t}{m_2}}} \right).$$

Thus, when $m_2 \geq d^4$, we can take $t = \iota^2/d^4$ to bound the probability by $poly(d, n, m_2)e^{-\iota}$, which concludes our proof. $\qquad\qquad\qquad\qquad\qquad\qquad\qquad\qquad\qquad\qquad\qquad\qquad\qquad\square$

Then we present the concentration of the reconstructed features.

**Lemma 25.** *Suppose $m_2 \geqslant C_\sigma^2 C_4^{-1/2} d^4 \|\mathbf{A}\|_{\mathrm{op}}^2$. Given any $\mathbf{A}$ such that $\mathbf{v}^\top \mathbf{A} \mathbf{v}$ is a quadratic spherical harmonic, with high probability on $\mathbf{V}$, for any $\mathbf{x} \in \mathcal{D}$, we have*

$$\left| \frac{1}{m_2} \sum_{i=1}^{m_2} (\mathbf{v}_i^\top \mathbf{A} \mathbf{v}_i) \sigma_2(\mathbf{v}_i^\top \mathbf{x}) - \frac{c_2}{B(d,2)} \mathbf{x}^\top \mathbf{A} \mathbf{x} \right| \leq \frac{9 \iota d^{-1} C_4^{1/4}}{\sqrt{m_2}}.$$

*Proof of Lemma 25.* Given any fixed $\mathbf{x} \in \mathcal{D}$ and $\mathbf{A}$ such that $\mathbf{v}^\top \mathbf{A} \mathbf{v}$ is a quadratic spherical harmonic, we have

$$
\begin{aligned}
\mathbb{E}_\mathbf{v} \left[ (\mathbf{v}^\top \mathbf{A} \mathbf{v})^2 \sigma_2^2 (\mathbf{v}^\top \mathbf{x}) \right] &\leq \sqrt{\mathbb{E}_\mathbf{v} \left[ (\mathbf{v}^\top \mathbf{A} \mathbf{v})^4 \right] \mathbb{E}_\mathbf{v} \left[ \sigma_2^4 (\mathbf{v}^\top \mathbf{x}) \right]} \\
&\leq (4-1)^{2*2} \mathbb{E} \left[ (\mathbf{v}_i^\top \mathbf{A} \mathbf{v}_i)^2 \right] d^{-2} C_4^{1/2} \\
&= 81 d^{-2} C_4^{1/2}
\end{aligned}
$$

and

$$\left| (\mathbf{v}^\top \mathbf{A} \mathbf{v}) \sigma_2 (\mathbf{v}^\top \mathbf{x}) \right| \leq d \|\mathbf{A}\|_{\mathrm{op}} \cdot C_\sigma = d C_\sigma \|\mathbf{A}\|_{\mathrm{op}}.$$

Since $\mathbb{E}_\mathbf{v} \left[ (\mathbf{v}^\top \mathbf{A} \mathbf{v}) \sigma_2 (\mathbf{v}^\top \mathbf{x}) \right] = \frac{c_2}{B(d,2)} \mathbf{x}^\top \mathbf{A} \mathbf{x}$, by Bernstein Inequality, we have

$$
\Pr \left[ \left| \frac{1}{m_2} \sum_{i=1}^{m_2} (\mathbf{v}_i^\top \mathbf{A} \mathbf{v}_i) \sigma_2 (\mathbf{v}_i^\top \mathbf{x}) - \frac{c_2}{B(d,2)} \mathbf{x}^\top \mathbf{A} \mathbf{x} \right| \geqslant \frac{9 \iota d^{-1} C_4^{1/4}}{\sqrt{m_2}} \right]
$$

$$
\leq 2 \exp \left( - \frac{\frac{81 d^{-2} C_4^{1/2} \iota^2}{2 m_2}}{\frac{81 d^{-2} C_4^{1/2}}{m_2} + \frac{1}{3 m_2} \cdot d (C_\sigma \|\mathbf{A}\|_{\mathrm{op}} \cdot \frac{9 \iota d^{-1} C_4^{1/4}}{\sqrt{m_2}})} \right)
$$

$$
= 2 \exp \left( - \frac{\iota^2/2}{1 + \frac{C_\sigma d^2 \|\mathbf{A}\|_{\mathrm{op}}}{27 C_4^{1/4} \sqrt{m_2}} \cdot \iota} \right)
$$

Thus, when $m_2 \geqslant C_\sigma^2 C_4^{-1/2} d^4 \|\mathbf{A}\|_{\mathrm{op}}^2$, by enumerating $\mathbf{x} \in \mathcal{D}$, we obtain that with high probability on $\mathbf{V}$, for any $\mathbf{x} \in \mathcal{D}$, we have

$$\left| \frac{1}{m_2} \sum_{i=1}^{m_2} (\mathbf{v}_i^\top \mathbf{A} \mathbf{v}_i) \sigma_2(\mathbf{v}_i^\top \mathbf{x}) - \frac{c_2}{B(d,2)} \mathbf{x}^\top \mathbf{A} \mathbf{x} \right| \leq \frac{9 \iota d^{-1} C_4^{1/4}}{\sqrt{m_2}}.$$

The proof is complete. □

## D  APPROXIMATION THEORY OF THE OUTER LAYER

### D.1  PROOF OF PROPOSITION 2

Since we mainly focus on the first training stage throughout this section, we may sometimes denote $n = n_1$ for notation simplicity, and let the training set be $\mathcal{D}_1 = \{\mathbf{x}_1, \mathbf{x}_2, \dots, \mathbf{x}_n\}$. Let's consider a formal version of Proposition 2.

**Proposition 6.** *Suppose $g$ is a degree $p$ polynomial. By setting $\eta = C \iota^{-5} \kappa_2^{-1} m_2^{-1/2} d^6$ for some constant $C > 0$, with high probability over $\mathcal{D}_1$, $\mathcal{D}_2$, $\{\mathbf{w}_i\}_{i=1}^{m_1}$ and $\mathbf{V}$, there exists $\mathbf{a}^\star \in \mathbb{R}^{m_1}$ such that the parameter $\theta^\star = (\mathbf{a}^\star, \mathbf{W}^{(1)}, \mathbf{b}^{(1)}, \mathbf{V})$ gives rise to*

$$
\begin{aligned}
\mathcal{L}_2(\theta^\star) := & \frac{1}{n_2} \sum_{\mathbf{x} \in \mathcal{D}_2} \left( f(\mathbf{x}; \theta^\star) - g(\mathbf{p}(\mathbf{x})) \right)^2 \\
\lesssim & \|g\|_{L^2}^2 \cdot \frac{r^p}{\lambda_{\min}(\mathbf{H})} \cdot \left( \frac{\iota^{p+2} d^5}{m_2} + \frac{\iota d^3}{\sqrt{m_2}} + \frac{\iota^{p+3/2} d}{\sqrt{n}} + \frac{\iota L r^2 \log^2 d}{d^{1/6}} \right)^2 \\
& + \frac{\iota^{p+1} \|g\|_{L^2}^2}{m_1} \cdot \left( \sum_{k=0}^{p} \eta^{-k} \|\mathbf{B}^\star\|_{\mathrm{op}}^k r^{\frac{p-k}{4}} \right)^2.
\end{aligned}
$$

*Here $\mathbf{a}^\star$ satisfies*

$$\frac{\|\mathbf{a}^\star\|_2^2}{m_1} \lesssim \iota^p \|g\|_{L^2}^2 \cdot \left(\sum_{k=0}^p \eta^{-k} \|\mathbf{B}^\star\|_{\mathrm{op}}^k r^{\frac{p-k}{4}}\right)^2 = \widetilde{\Omega}(\kappa_2^{2p} r^p).$$

To prove the proposition, let's introduce the infinite-outer-width model as a transition term between the finite-outer-width model and the target function. We define the infinite-outer-width model as

$$f_{\infty,m_2}(\mathbf{x};v) = \mathbb{E}_{a,b,\mathbf{w}}\left[v(a,b,\mathbf{w})\sigma_1\Big(a\eta\langle\mathbf{w},\mathbf{h}^{(1)}(\mathbf{x})\rangle + b\Big)\right],$$

where $\mathbf{h}^{(1)}(\mathbf{x}') = \frac{1}{n}\sum_{i=1}^n f^\star(\mathbf{x}_i) \cdot K_{m_2}^{(0)}(\mathbf{x}_i,\mathbf{x}') \cdot \sigma_2\big(\mathbf{V}^\top \mathbf{x}_i\big)$.

We can decompose the $L^2$ loss of the truth model $f(\mathbf{x};\theta)$ as

$$\begin{aligned}
\hat{\mathcal{L}}(\theta^\star) &= \frac{1}{n}\sum_{\mathbf{x}\in\mathcal{D}_2}(f(\mathbf{x};\theta) - f^\star(\mathbf{x}))^2\\
&= \frac{1}{n}\sum_{\mathbf{x}\in\mathcal{D}_2}\Big(f(\mathbf{x};\theta) - f_{\infty,m_2}(\mathbf{x}') + f_{\infty,m_2}(\mathbf{x}') - g(\mathbf{B}^\star\mathbf{h}^{(1)}(\mathbf{x})) + g(\mathbf{B}^\star\mathbf{h}^{(1)}(\mathbf{x})) - g(\mathbf{p}(\mathbf{x}))\Big)^2\\
&\lesssim \underbrace{\frac{1}{n}\sum_{\mathbf{x}\in\mathcal{D}_2}(f(\mathbf{x};\theta) - f_{\infty,m_2}(\mathbf{x}'))^2}_{L_1}\\
&\quad + \underbrace{\frac{1}{n}\sum_{\mathbf{x}\in\mathcal{D}_2}\Big(f_{\infty,m_2}(\mathbf{x}') - g(\mathbf{B}^\star\mathbf{h}^{(1)}(\mathbf{x}))\Big)^2}_{L_2}\\
&\quad + \underbrace{\frac{1}{n}\sum_{\mathbf{x}\in\mathcal{D}_2}\Big(g(\mathbf{B}^\star\mathbf{h}^{(1)}(\mathbf{x})) - g(\mathbf{p}(\mathbf{x}))\Big)^2}_{L_3}.
\end{aligned}$$

We have bounded $L_3$ in Corollary 3. We state Lemmas 26 and 27 as follows to bound $L_1$ and $L_2$, respectively.

**Lemma 26** (Bound $L_2$). *Given $\mathbf{B}^\star \in \mathbb{R}^{r\times m_2}$ and setting the learning rate $\eta = C\iota^{-5}\kappa_2^{-1}m_2^{-1/2}d^6$ for a constant $C > 0$, there exists $v : \{\pm 1\} \times \mathbb{R} \times \mathbb{R}^{m_2} \to \mathbb{R}$ such that*

$$\|v\|_{L^2} \lesssim \|g\|_{L^2}\sum_{k=0}^p \eta^{-k}\|\mathbf{B}^\star\|_{\mathrm{op}}^k r^{\frac{p-k}{4}},$$

*and, with high probability over $\mathcal{D}_1, \mathcal{D}_2$ and $\mathbf{V}$, the infinite-width network satisfies*

$$\frac{1}{n}\sum_{\mathbf{x}\in\mathcal{D}_2}(f_{\infty,m_2}(\mathbf{x};v) - g(\mathbf{B}^\star\mathbf{h}^{(1)}(\mathbf{x})))^2 \lesssim o\left(\frac{1}{d^2 n_1^2 n_2^2 m_1^2 m_2^2}\right).$$

**Lemma 27** (Bound $L_1$). *Given the function $v : \{\pm 1\} \times \mathbb{R} \times \mathbb{R}^{m_2} \to \mathbb{R}$ in Lemma 26. With high probability over $\mathcal{D}_1, \mathcal{D}_2, \{\mathbf{w}_i\}_{i=1}^{m_1}$ and $\mathbf{V}$, it holds that for any $\mathbf{x} \in \mathcal{D}_2$,*

$$\left|\frac{1}{m_1}\sum_{i=1}^{m_1} v(a_i,b_i,\mathbf{w}_i)\sigma_1(\eta a_i\langle\mathbf{w}_i,\mathbf{h}^{(1)}(\mathbf{x})\rangle + b_i) - f_{\infty,m_2}(\mathbf{x};v)\right| \lesssim \sqrt{\frac{\iota^{p+1}\|v\|_{L^2}^2}{m_1}}, \quad with$$

$$\frac{1}{m_1}\sum_{i=1}^{m_1} v(a_i,b_i,\mathbf{w}_i)^2 \lesssim \iota^p\|v\|_{L^2}^2.$$

The proof of Lemmas 26 and 27 is provided in Appendix D.2. Now we begin our proof of Proposition 6.

*Proof of Proposition 6.* By Corollary 3, Lemma 27 and Lemma 26, by defining the vector $\mathbf{a}^\star \in \mathbb{R}^{m_1}$ by $a_i^\star = v(a_i^{(0)}, b_i^{(1)}, \mathbf{w}_i^{(1)})$ and letting $\theta^* = (\mathbf{a}^*, \mathbf{W}^{(1)}, \mathbf{b}^{(1)}, \mathbf{V})$, we have with high probability that

$$\hat{\mathcal{L}}_2(\theta^*) \lesssim L_1 + L_2 + L_3$$

$$\lesssim \|g\|_{L^2}^2 \cdot \frac{r^p}{\lambda_{\min}^2(\mathbf{H})} \cdot \left( \frac{\iota^{p+2} d^5}{m_2} + \frac{\iota d^3}{\sqrt{m_2}} + \frac{\iota^{p+3/2} d}{\sqrt{n}} + \frac{\iota L r^2 \log^2 d}{d^{1/6}} \right)^2$$

$$+ \frac{\iota^{p+1} \|g\|_{L^2}^2}{m_1} \cdot \left( \sum_{k=0}^p \eta^{-k} \|\mathbf{B}^\star\|_{\mathrm{op}}^k \, r^{\frac{p-k}{4}} \right)^2$$

$$+ o\left( \frac{1}{d^2 n_1^2 n_2^2 m_1^2 m_2^2} \right)$$

$$\lesssim \|g\|_{L^2}^2 \cdot \frac{r^p}{\lambda_{\min}^2(\mathbf{H})} \cdot \left( \frac{\iota^{p+2} d^5}{m_2} + \frac{\iota d^3}{\sqrt{m_2}} + \frac{\iota^{p+3/2} d}{\sqrt{n}} + \frac{\iota L r^2 \log^2 d}{d^{1/6}} \right)^2$$

$$+ \frac{\iota^{p+1} \|g\|_{L^2}^2}{m_1} \cdot \left( \sum_{k=0}^p \eta^{-k} \|\mathbf{B}^\star\|_{\mathrm{op}}^k \, r^{\frac{p-k}{4}} \right)^2 .$$

Here $\mathbf{a}^\star$ satisfies

$$\|\mathbf{a}^\star\|_2^2 \leq \sum_{i=1}^{m_1} v(a_i, b_i, \mathbf{w}_i)^2$$

$$\lesssim m_1 \iota^p \|v\|_{L^2}^2$$

$$\lesssim m_1 \iota^p \|g\|_{L^2}^2 \left( \sum_{k=0}^p \eta^{-k} \|\mathbf{B}^\star\|_{\mathrm{op}}^k \, r^{\frac{p-k}{4}} \right)^2 .$$

The proof is complete. □

## D.2 Omitted Proofs in Appendix D.1

### D.2.1 Random Feature Construction of Univariate Polynomials

In this section, before proving Lemmas 26 and 27, we first construct univariate polynomials using the outer activation function $\sigma_1$ and the random features $a$ and $b$ progressively.

**Lemma 28.** *There exists $v_0(a, b)$, supported on $\{\pm 1\} \times [2,3]$, such that for any $|z| \leq 1$*

$$\mathbb{E}_{a,b}[v_0(a,b)\sigma(az+b)] = 1, \quad \sup_{a,b} |v(a,b)| \lesssim 1.$$

*Proof.* Let $v_0(a,b) = 12 \cdot \mathbf{1}_{a=1}(b - \frac{5}{2}) \cdot \frac{\mathbf{1}_{b \in [2,3]}}{\mu(b)}$. Then, since $z + b \geqslant 1$,

$$\mathbb{E}_{a,b}[v_0(a,b)\sigma(az+b)] = 6 \int_2^3 (b - \frac{5}{2})\sigma(z+b)db$$

$$= 6 \int_2^3 (b - \frac{5}{2})(2z + 2b - 1)db$$

$$= z \cdot 6 \int_2^3 (b - \frac{5}{2})db + 6 \int_2^3 (b - \frac{5}{2})(2b - 1)db$$

$$= 1.$$

The proof is complete. □

**Lemma 29.** *There exists $v_1(a, b)$, supported on $\{\pm 1\} \times [2,3]$, such that for any $|z| \leq 1$*

$$\mathbb{E}_{a,b}[v_1(a,b)\sigma(az+b)] = z, \quad \sup_{a,b} |v(a,b)| \lesssim 1.$$

*Proof.* Let $v_0(a,b) = \mathbf{1}_{a=1}(-24b + 61) \cdot \frac{\mathbf{1}_{b \in [2,3]}}{\mu(b)}$. Then, since $z + b \geqslant 1$,

$$
\begin{aligned}
\mathbb{E}_{a,b}[v_1(a,b)\sigma(az + b)] &= \frac{1}{2}\int_2^3 (-24b + 61)\sigma(z + b)db \\
&= \frac{1}{2}\int_2^3 (-24b + 61)(2z + 2b - 1)db \\
&= z\int_2^3 (-24b + 61)db + \frac{1}{2}\int_2^3 (-24b + 61)(2b - 1)db \\
&= z.
\end{aligned}
$$

The proof is complete. $\qquad\square$

**Lemma 30.** *There exists $v_2(a,b)$, supported on $\{\pm 1\} \times [-2, 3]$, such that for any $|z| \leq 1$*

$$
\mathbb{E}_{a,b}[v_2(a,b)\sigma(az + b)] = z^2, \quad \sup_{a,b}|v(a,b)| \lesssim 1.
$$

*Proof.* First, see that

$$
\begin{aligned}
\int_{-2}^2 \sigma(z + b)db &= \int_{-2+z}^{2+z} \sigma(b)db \\
&= \int_{-2+z}^{-1} (-2b - 1)db + \int_{-1}^1 b^2 db + \int_1^{2+z} (2b - 1)db \\
&= [-b^2 - b]_{-2+z}^{-1} + \frac{2}{3} + [b^2 - b]_1^{2+z} \\
&= (z - 2)^2 + (z - 2) + \frac{2}{3} + (z + 2)^2 - (z + 2) \\
&= 2z^2 + \frac{14}{3}.
\end{aligned}
$$

Let $v_2(a,b) = \mathbf{1}_{a=1}\frac{\mathbf{1}_{b \in [-2,2]}}{\mu(b)} - \frac{7}{3}v_0(a,b)$ Then

$$
\begin{aligned}
\mathbb{E}_{a,b}[v_2(a,b)\sigma(az + b)] &= \frac{1}{2}\int_{-2}^2 \sigma(z + b)db - \frac{7}{3} \\
&= z^2 + \frac{7}{3} - \frac{7}{3} \\
&= z^2.
\end{aligned}
$$

The proof is complete. $\qquad\square$

**Lemma 31.** *Let $v(b) = -\frac{1}{2}k(k-1)(k-2)(1-b)^{k-3} \cdot \frac{\mathbf{1}_{b \in [0,1]}}{\mu(b)}$. Then*

$$
\mathbb{E}_b[v_k(b)\sigma(z + b)] = z^k \cdot \mathbf{1}_{z>0} - \frac{k(k-1)}{2}z^2 - kz - 1.
$$

*Proof.* Plugging in $v_k(b)$ and applying integration by parts yields

$$
\begin{aligned}
\mathbb{E}_b[v_k(b)\sigma(z + b)] &= \int_0^1 -\frac{1}{2}k(k-1)(k-2)(1-b)^{k-3}\sigma(z + b)db \\
&= [\frac{1}{2}k(k-1)(1-b)^{k-2}\sigma(z + b)]_0^1 - \int_0^1 \frac{1}{2}k(k-1)(1-b)^{k-2}\sigma'(z + b)db \\
&= -\frac{1}{2}k(k-1)\sigma(z) + [\frac{1}{2}k(1-b)^{k-1}\sigma'(z + b)]_0^1 - \int_0^1 \frac{1}{2}k(1-b)^{k-1}\sigma''(z + b)db \\
&= -\frac{1}{2}k(k-1)\sigma(z) - \frac{1}{2}k\sigma'(z) - \int_0^1 k(1-b)^{k-1}\mathbf{1}_{|z+b|\leq 1}db
\end{aligned}
$$

When $1 \geqslant z > 0$, we have

$$-\int_0^1 k(1-b)^{k-1}\mathbf{1}_{|z+b|\leq 1}db = -\int_0^{1-z} k(1-b)^{k-1}db = [(1-b)^k]_0^{1-z} = z^k - 1.$$

When $-1 \leq z \leq 0$, we have

$$-\int_0^1 k(1-b)^{k-1}\mathbf{1}_{|z+b|\leq 1}db = -\int_0^1 k(1-b)^{k-1}db = -1.$$

Since $z \in [-1,1]$, we have that $\sigma(z) = z^2$ and $\sigma'(z) = 2z$. Therefore for $z \in [-1,1]$

$$\mathbb{E}_b[v_k(b)\sigma(z+b)] = z^k \cdot \mathbf{1}_{z>0} - \frac{k(k-1)}{2}z^2 - kz - 1.$$

The proof is complete. $\qquad\square$

**Lemma 32.** *There exists $v_k(a,b)$, supported on $\{\pm 1\} \times [-2,3]$, such that for any $|z| \leq 1$*

$$\mathbb{E}_{a,b}[v_k(a,b)\sigma(az+b)] = z^k, \quad \sup_{a,b}|v_k(a,b)| \lesssim \text{poly}(k).$$

*Proof.* We focus on $k \geqslant 3$. We have that

$$\mathbb{E}_b[v_k(b)\sigma(z+b)] = z^k \cdot \mathbf{1}_{z>0} - \frac{k(k-1)}{2}z^2 - kz - 1.$$

$$\mathbb{E}_b[v_k(b)\sigma(-z+b)] = (-z)^k \cdot \mathbf{1}_{z<0} - \frac{k(k-1)}{2}z^2 + kz - 1.$$

Therefore if $k$ is even

$$\mathbb{E}_b[v(b)\sigma(z+b) + v(b)\sigma(-z+b)] = z^k - k(k-1)z^2 - 2.$$

Let $v_k(a,b) = 2v_k(b) + k(k-1)v_2(a,b) + 2$. Then

$$\mathbb{E}_{a,b}[v_k(a,b)\sigma(az+b)] = \mathbb{E}_b[v_k(b)\sigma(z+b) + v_k(b)\sigma(z-b)] + k(k-1)z^2 + 2 = z^k.$$

If $k$ is odd,

$$\mathbb{E}_b[v(b)\sigma(z+b) - v(b)\sigma(-z+b)] = z^k - 2kz.$$

Let $v_k(a,b) = 2av_k(b) + 2kv_1(a,b)$. Then

$$\mathbb{E}_{a,b}[v_k(a,b)\sigma(az+b)] = \mathbb{E}_b[v_k(b)\sigma(z+b) - v_k(b)\sigma(z-b)] + 2kz = z^k.$$

The proof is complete. $\qquad\square$

### D.2.2 Proof of Supporting Lemmas in Appendix D.1

*Proof of Lemma 26.* Let's consider a general version of Lemma 26.

**Lemma 33.** *Let $g : \mathbb{R}^r \to \mathbb{R}$ be a degree $p$ polynomial, and let $\mathbf{B}^\star \in \mathbb{R}^{r\times m_2}$. Given a set of vectors $\mathcal{D} = \{\mathbf{z}_1, \mathbf{z}_2, \ldots, \mathbf{z}_n\} \subseteq \mathbb{R}^{m_2}$ that satisfies $\eta\langle\mathbf{w}, \mathbf{z}\rangle \leq 1$ for any $\mathbf{z} \in \mathcal{D}$ with probability at least $1 - 2(n_1 + n_2)\exp(-\iota^2/2)$ over $\mathbf{w} \sim \mathcal{N}(\mathbf{0}_{m_2}, \mathbf{I}_{m_2})$ (uniformly over $\mathcal{D}$). Then, there exists $v : \{\pm 1\} \times \mathbb{R} \times \mathbb{R}^m \to \mathbb{R}$ so that for all $\mathbf{z} \in \mathcal{D}$,*

$$\mathbb{E}_{a,b,\mathbf{w}}[v(a,b,\mathbf{w})\sigma_1(\eta a\langle\mathbf{w}, \mathbf{z}\rangle + b)] = g(\mathbf{B}^\star\mathbf{z}) + o\left(\frac{1}{dn_1 n_2 m_1 m_2}\right), \quad \text{and}$$

$$\|v\|_{L^2} \lesssim \|g\|_{L^2} \sum_{k=0}^p \eta^{-k} \|\mathbf{B}^\star\|_{\text{op}}^k r^{\frac{p-k}{4}}.$$

Thus, according to Proposition 4, we could set the learning rate $\eta = C\iota^{-5}\kappa_2^{-1}m_2^{-1/2}d^6$ for a constant $C > 0$ to ensure $\left|\eta\langle\mathbf{w}, \mathbf{h}^{(1)}(\mathbf{x}')\rangle\right| \leq 1$ for any $\mathbf{x}' \in \mathcal{D}_2$ with high probability on $\mathbf{V}, \mathcal{D}_1$, and probability at least $1 - 2(n_1 + n_2)\exp(-\iota^2/2)$ on $\mathbf{w}$. Thus, taking $\mathcal{D} = \{\mathbf{h}(\mathbf{x})\}_{\mathbf{x}\in\mathcal{D}_2}$ concludes our proof. $\qquad\square$

To prove Lemma 33, we first decompose $g$ into sum of polynomials of different degrees and construct a function $v$ to express these polynomials accordingly.

**Lemma 34.** *Given* $\mathbf{z} \in \mathbb{R}^{m_2}$. *Let* $\mathbf{B}^\star \in \mathbb{R}^{r \times m_2}$ *and* $\mathbf{T}_k \in (\mathbb{R}^r)^{\otimes k}$. *Then, there exists* $v_k : \mathbb{R}^{m_2} \to \mathbb{R}$ *such that*

$$\mathbb{E}_{\mathbf{w}}\left[v_k(\mathbf{w})(\eta\langle\mathbf{w}, \mathbf{z}\rangle)^k\right] = \mathbf{T}_k\left((\mathbf{B}^\star \mathbf{z})^{\otimes k}\right).$$

*Here* $v_k$ *satisfies*

$$\|v_k\|_{L^2} \lesssim \eta^{-k} \|\mathbf{B}^\star\|_{op}^k \|\mathbf{T}_k\|_F \quad and \quad \sup_w |v_k(\mathbf{w})| \lesssim m_2^{k/2}\eta^{-k} \|\mathbf{B}^\star\|_{op}^k \|\mathbf{T}_k\|_F. \qquad (35)$$

*Proof of Lemma 34.* It suffices to solve

$$\mathbb{E}_{\mathbf{w}}[v(\mathbf{w})\mathbf{w}^{\otimes k}] = \eta^{-k}\mathbf{B}^{\star\otimes k}(\mathbf{T}_k),$$

where $\mathbf{B}^{\star\otimes k}(\mathbf{T}_k) \in (\mathbb{R}^{m_2})^{\otimes k}$. This is achieved by setting

$$v(\mathbf{w}) := \eta^{-k}\operatorname{Vec}(\mathbf{w}^{\otimes k})^T \operatorname{Mat}(\mathbb{E}[\mathbf{w}^{\otimes 2k}])^{-1} \operatorname{Vec}(\mathbf{B}^{\star\otimes k}(\mathbf{T}_k)).$$

Then,

$$\|v\|_{L^2}^2 = \eta^{-2k}\operatorname{Vec}(\mathbf{B}^{\star\otimes k}(\mathbf{T}_k))^T \operatorname{Mat}(\mathbb{E}[\mathbf{w}^{\otimes 2k}])^{-1} \operatorname{Vec}(\mathbf{B}^{\star\otimes k}(\mathbf{T}_k)).$$

Since

$$\operatorname{Mat}(\mathbb{E}[\mathbf{w}^{\otimes 2k}]) \succeq k!\Pi_{\operatorname{Sym}^k(\mathbb{R}^{m_2})},$$

we have

$$\|v\|_{L^2}^2 \lesssim \eta^{-2k} \left\|\mathbf{B}^{\star\otimes k}(\mathbf{T}_k)\right\|_F^2 \leq \eta^{-2k} \|\mathbf{B}^\star\|_{op}^{2k} \|\mathbf{T}_k\|_F^2.$$

Finally,

$$\begin{aligned}
\sup_w |v(\mathbf{w})| &= \eta^{-k}\sup_w \left|\operatorname{Vec}(\mathbf{w}^{\otimes k})^T \operatorname{Mat}(\mathbb{E}[\mathbf{w}^{\otimes 2k}])^{-1} \operatorname{Vec}(\mathbf{B}^{\star\otimes k}(\mathbf{T}_k))\right| \\
&\leq \left\|\mathbf{w}^{\otimes k}\right\|_F \left\|\mathbf{B}^{\star\otimes k}(\mathbf{T}_k)\right\|_F \\
&\lesssim m_2^{k/2}\eta^{-k} \|\mathbf{B}^\star\|_{op}^k \|\mathbf{T}_k\|_F.
\end{aligned}$$

The proof is complete. $\qquad\square$

Then we begin our proof of Lemma 33.

*Proof of Lemma 33.* We can write

$$g(\mathbf{z}) = \sum_{k=0}^p \langle\mathbf{T}_k, \mathbf{z}^{\otimes k}\rangle.$$

By Lemma 10, we have $\|\mathbf{T}_k\|_F \lesssim r^{\frac{p-k}{4}} \|g\|_{L^2}$.

Define $v_k(a, b)$ to be the function so that $\mathbb{E}_{a,b}[v_k(a, b)\sigma_1(az + b)] = z^k$, and let $v_k(\mathbf{w})$ be the function where $\mathbb{E}_{\mathbf{w}}[v(\mathbf{w})(\eta\langle\mathbf{w}, \mathbf{z}\rangle)^k] = \langle\mathbf{T}_k, (\mathbf{B}^\star \mathbf{z})^{\otimes k}\rangle$. Next, define

$$v(a, b, \mathbf{w}) = \sum_{k=0}^p v_k(a, b)v_k(\mathbf{w}).$$

Here $v_k(a, b)$ is defined in Lemma 34. Then we have that

$$\|v\|_{L^2} \lesssim \sum_{k=0}^p (\mathbb{E}[v_k(\mathbf{w})^2])^{1/2} \leq \|g\|_{L^2} \sum_{k=0}^p \eta^{-k} \|\mathbf{B}^\star\|_{op}^k r^{\frac{p-k}{4}}.$$

Note that $\|v\|_{L^2} = \mathcal{O}(\text{poly}(m_2, d))$ and $|\sigma_1(\eta a\langle \mathbf{w}, \mathbf{z}\rangle + b)| \leq \eta a\langle \mathbf{w}, \mathbf{z}\rangle + b)$ has polynomial growth. Since we have taken $\iota = C\log(dn_1n_2m_1m_2)$ for some sufficiently large $C > 0$, we know by Cauchy inequality,

$$|\mathbb{E}_{a,b,\mathbf{w}}[v(\mathbf{w})\sigma_1(\eta\langle \mathbf{w}, \mathbf{z}\rangle + b)\mathbf{1}\{\eta\langle \mathbf{w}, \mathbf{z}\rangle > 1\}]| \leq o\left(\frac{1}{dn_1n_2m_1m_2}\right).$$

Thus, we then have that

$$\mathbb{E}_{a,b,\mathbf{w}}[v(a, b, \mathbf{w})\sigma_1(\eta a\langle \mathbf{w}, \mathbf{z}\rangle + b)]$$

$$= \mathbb{E}_{a,b,\mathbf{w}}[v(a, b, \mathbf{w})\sigma_1(\eta a\langle \mathbf{w}, \mathbf{z}\rangle + b) \cdot \mathbf{1}_{|\eta\langle \mathbf{w}, \mathbf{z}\rangle| \leq 1}] + o\left(\frac{1}{dn_1n_2m_1m_2}\right)$$

$$= \sum_{k=0}^{p} \mathbb{E}_{a,b,\mathbf{w}}[v_k(a, b)v_k(\mathbf{w})\sigma_1(\eta a\langle \mathbf{w}, \mathbf{z}\rangle + b) \cdot \mathbf{1}_{|\eta\langle \mathbf{w}, \mathbf{z}\rangle| \leq 1}] + o\left(\frac{1}{dn_1n_2m_1m_2}\right)$$

$$= \sum_{k=0}^{p} \mathbb{E}_{\mathbf{w}}[v_k(\mathbf{w})(\eta\langle \mathbf{w}, \mathbf{z}\rangle)^k \cdot \mathbf{1}_{|\eta\langle \mathbf{w}, \mathbf{z}\rangle| \leq 1}] + o\left(\frac{1}{dn_1n_2m_1m_2}\right)$$

$$= \sum_{k=0}^{p} \mathbb{E}_{\mathbf{w}}[v_k(\mathbf{w})(\eta\langle \mathbf{w}, \mathbf{z}\rangle)^k] + o\left(\frac{1}{dn_1n_2m_1m_2}\right)$$

$$= \sum_{k=0}^{p} \langle \mathbf{T}_k, (\mathbf{B}^\star \mathbf{z})^{\otimes k}\rangle + o\left(\frac{1}{dn_1n_2m_1m_2}\right)$$

$$= g(\mathbf{B}^\star \mathbf{z}) + o\left(\frac{1}{dn_1n_2m_1m_2}\right).$$

The proof is complete. $\qquad\square$

*Proof of Lemma 27.* Fix $\mathbf{x} \in \mathcal{D}_2$. For notation simplicity, we denote $f_v^\infty(\mathbf{x}) = f_{\infty,m_2}(\mathbf{x}; v)$. Consider a truncation radius $R > 0$ to be chosen later and let $E_x$ be the set of $\mathbf{w}$ such that

$$\sup_{a,b}|v(a, b, \mathbf{w})| \leq R \text{ and } \eta\langle \mathbf{w}, \mathbf{h}^{(1)}(\mathbf{x})\rangle \leq 1.$$

By the construction of $v(a, b, \mathbf{w})$ in the proof of Lemma 33, we know it can be seen as a degree-$p$ polynomial of $\mathbf{w}$. Thus, by Lemma 7, by taking $R = C\iota^{p/2}\|v\|_{L^2}$ for some sufficiently large $C > 0$, we can ensure that

$$\Pr[\sup_{a,b}|v(a, b, \mathbf{w})| \leq R] \geq 1 - \exp(-\iota).$$

Moreover, by Proposition 4, conditional on a high probability event on $\mathbf{V}$, $\mathcal{D}_1$ and $\mathcal{D}_2$, by taking $\eta = C\iota^{-5}d^6m_2^{-1}$, we have $\Pr[\eta\langle \mathbf{w}, \mathbf{h}^{(1)}(\mathbf{x})\rangle \leq 1] \geq 1 - 4\exp(-\iota^2/2)$ for a single $\mathbf{x}$. Now consider the random variables

$$Z_i := \mathbf{1}\{\mathbf{w}_i \in E_x\}v(a_i, b_i, \mathbf{w}_i)\sigma_1(\eta a_i\langle \mathbf{w}_i, \mathbf{h}^{(1)}(\mathbf{x})\rangle + b_i), \ i = 1, 2, \ldots, m_1.$$

We directly have that $|Z_i| \lesssim \iota^{p/2}\|v\|_{L^2}$, and with high probability,

$$\frac{1}{m_1}\sum_{i=1}^{m_1} v(a_i, b_i, \mathbf{w}_i)^2 \lesssim \iota^p\|v\|_{L^2}^2.$$

Therefore by Hoeffding inequality, with probability at least $1 - 2\exp(-\iota)$, we have

$$\left|\frac{1}{m_1}\sum_{i=1}^{m_1} \mathbf{1}_{\mathbf{w}_i \in E_x}v(a_i, b_i, \mathbf{w}_i)\sigma_1(\eta a_i\langle \mathbf{w}_i, \mathbf{h}^{(1)}(\mathbf{x})\rangle + b_i)\right.$$

$$\left. - \mathbb{E}[\mathbf{1}_{w \in E_x}v(a, b, \mathbf{w})\sigma_1(\eta a\langle \mathbf{w}, \mathbf{h}^{(1)}(\mathbf{x})\rangle + b)]\right| \lesssim \sqrt{\frac{\iota^{p+1}\|v\|_{L^2}^2}{m_1}}.$$

Similar to the proof of Lemma 33, note that both $v(a, b, \mathbf{w})$ and $|\sigma_1(\eta a \langle \mathbf{w}, \mathbf{z} \rangle + b)|$ has polynomial growth. Since we have taken $\iota = C \log(dn_1n_2m_1m_2)$ for some sufficiently large $C > 0$, we know by Cauchy inequality,

$$
\left| \mathbb{E}[\mathbf{1}_{w \in E_x} v(a, b, \mathbf{w}) \sigma_1(\eta a \langle \mathbf{w}, \mathbf{h}^{(1)}(\mathbf{x}) \rangle + b)] - f_v^\infty(\mathbf{x}) \right|
$$
$$
= \left| \mathbb{E}[\mathbf{1}_{w \notin E_x} v(a, b, \mathbf{w}) \sigma_1(\eta a \langle \mathbf{w}, \mathbf{h}^{(1)}(\mathbf{x}) \rangle + b)] \right|
$$
$$
\leq \mathbb{P}(\mathbf{w} \notin E_x)(\mathbb{E}[v(a, b, \mathbf{w})^2 \sigma_1(\eta a \langle \mathbf{w}, \mathbf{h}^{(1)}(\mathbf{x}) \rangle + b)^2])^{1/2}
$$
$$
\lesssim \exp(-C \log(dm_1m_2n_1n_2))\widetilde{O}(\|v\|_{L^2})
$$
$$
\lesssim \frac{1}{m_1}.
$$

Finally, union bounding over $\mathbf{x} \in \mathcal{D}_2$, we see that

$$
\sup_{\mathbf{x} \in \mathcal{D}_2} \left| \frac{1}{m_1} \sum_{i=1}^{m_1} v(a_i, b_i, \mathbf{w}_i) \sigma_1(\eta a_i \langle \mathbf{w}_i, \mathbf{h}^{(1)}(\mathbf{x}) \rangle + b_i) - f_v^\infty(\mathbf{x}) \right| \lesssim \sqrt{\frac{\iota^{p+1} \|v\|_{L^2}^2}{m_1}} + \frac{1}{m_1}
$$
$$
\lesssim \sqrt{\frac{\iota^{p+1} \|v\|_{L^2}^2}{m_1}}.
$$

The proof is complete. $\qquad\qquad\qquad\qquad\qquad\qquad\qquad\qquad\qquad\qquad\qquad\qquad\qquad\qquad\qquad$ $\square$

# E  GENERALIZATION THEORY

## E.1  FORMAL PROOF OF THEOREM 1

The proof is divided into two parts. The first part of proof formalizes the proof we present in Section 4. The second part presents the generalization theory after we construct $\mathbf{a}^\star$ that gives small $L^2$ error by Proposition 2, with the formal version presented in Proposition 6.

### E.1.1  PART1: ANALYSIS BEFORE FEATURE RECONSTRUCTION

Denote $\mathbf{w}_j = \epsilon^{-1} \mathbf{w}_j^{(0)} \sim \mathcal{N}(0, \mathbf{I}_{m_2})$. Note that for any $\mathbf{x} \in \mathcal{D}_1$ and $j \in [m_1]$, we have

$$
\langle \mathbf{w}_j^{(0)}, \mathbf{h}^{(0)}(\mathbf{x}) \rangle = \langle \epsilon \mathbf{w}_j, \mathbf{h}^{(0)}(\mathbf{x}) \rangle \sim \mathcal{N}\left(0, \epsilon^2 \left\| \mathbf{h}^{(0)}(\mathbf{x}) \right\|_2^2\right).
$$

Since $\left\| \mathbf{h}^{(0)}(\mathbf{x}) \right\|_2^2 = \sum_{k=1}^{m_2} \sigma_2^2(\mathbf{v}_k^\top \mathbf{x}) \leq m_2 C_\sigma^2$. By setting $\epsilon^{-1} = C_\sigma \sqrt{2\iota m_2}$, we know $\langle \mathbf{w}_j^{(0)}, \mathbf{h}^{(0)}(\mathbf{x}) \rangle \leq 1$ with probability at least $1 - 2\exp(-\iota)$. Thus, uniformly bounding over $\mathbf{x} \in \mathcal{D}_1$ and $j \in [m_1]$, we know with high probability over $\mathbf{W}$, we have

$$
\langle \mathbf{w}_j^{(0)}, \mathbf{h}^{(0)}(\mathbf{x}) \rangle \leq 1 \text{ for any } \mathbf{x} \in \mathcal{D}_1, \ j \in [m_1].
$$

Then, according Algorithm 1, after one-step gradient descent on $\mathbf{W}$, we know with high probability, for each $j \in [m_1]$,

$$
\eta_1 \nabla_{\mathbf{w}_j^{(0)}} \mathcal{L}(\theta^{(0)}) = -\eta_1 \frac{a_j^{(0)}}{m_1} \cdot \frac{1}{n_1} \sum_{\mathbf{x} \in \mathcal{D}_1} f^*(\mathbf{x}) \mathbf{h}^{(0)}(\mathbf{x}) \sigma_1'\left(\langle \epsilon \mathbf{w}_j, \mathbf{h}^{(0)}(\mathbf{x}) \rangle\right)
$$
$$
= -\frac{2\epsilon \eta_1}{m_1} a_j^{(0)} \cdot \frac{1}{n_1} \sum_{i=1}^{n} f^*(\mathbf{x}) \mathbf{h}^{(0)}(\mathbf{x}) \mathbf{h}^{(0)}(\mathbf{x})^\top \mathbf{w}_j,
$$

which is a linear transformation on $\mathbf{w}_j$. By taking $\eta_1 = \frac{m_1}{2\epsilon m_2} \cdot \eta$ for some $\eta > 0$ to be chosen later and $\lambda_1 = \eta_1^{-1}$, we have

$$
\mathbf{w}_j^{(1)} = \mathbf{w}_j^{(0)} - \eta_1 \left[ \nabla_{\mathbf{w}_j^{(0)}} \mathcal{L}(\theta^{(0)}) + \lambda_1 \mathbf{w}_j^{(0)} \right]
$$
$$
= -\eta_1 \nabla_{\mathbf{w}_j^{(0)}} \mathcal{L}(\theta^{(0)})
$$
$$
= \frac{\eta a_j^{(0)}}{m_2} \cdot \frac{1}{n_1} \sum_{i=1}^{n} f^*(\mathbf{x}) \mathbf{h}^{(0)}(\mathbf{x}) \mathbf{h}^{(0)}(\mathbf{x})^\top \mathbf{w}_j.
$$

Then for any second-stage training sample $\mathbf{x}' \in \mathcal{D}_2$, the inner-layer neuron becomes

$$\left\langle \mathbf{w}_j^{(1)}, \sigma_2(\mathbf{V}\mathbf{x}') \right\rangle = \frac{\eta a_j^{(0)}}{m_2} \left\langle \frac{1}{n_1} \sum_{i=1}^n f^*(\mathbf{x}) \mathbf{h}^{(0)}(\mathbf{x}) \mathbf{h}^{(0)}(\mathbf{x})^\top \mathbf{w}_j, \mathbf{h}^{(0)}(\mathbf{x}') \right\rangle$$

$$= \eta a_j^{(0)} \cdot \left\langle \mathbf{w}_j, \frac{1}{n_1} \sum_{i=1}^n K_{m_2}^{(0)}(\mathbf{x}, \mathbf{x}') \mathbf{h}^{(0)}(\mathbf{x}) \right\rangle$$

$$= \eta a_j^{(0)} \cdot \langle \mathbf{w}_j, \mathbf{h}^{(1)}(\mathbf{x}') \rangle.$$

Thus, after the first training stage and reinitialization on $\mathbf{b} = \mathbf{b}^{(1)}$, the model becomes the following random-feature model in the second stage:

$$f(\mathbf{x}'; \theta) = \frac{1}{m_1} \sum_{j=1}^{m_1} a_j \sigma_1 \left( \eta a_j^{(0)} \langle \mathbf{w}_j, \mathbf{h}^{(1)}(\mathbf{x}') \rangle + b_j^{(1)} \right).$$

By Proposition 6, we know there exists $\mathbf{a}^\star \in \mathbb{R}^{m_1}$ such that with high probability over $\mathcal{D}_1, \mathcal{D}_2$, $\{\mathbf{w}_i\}_{i=1}^{m_1}$ and $\mathbf{V}$, by taking the parameter $\theta^\star = (\mathbf{a}^\star, \mathbf{W}^{(1)}, \mathbf{b}^{(1)}, \mathbf{V})$, it holds that

$$\hat{\mathcal{L}}_2(\theta^\star) \lesssim \|g\|_{L^2}^2 \cdot \frac{r^p}{\lambda_{\min}(\mathbf{H})} \cdot \left( \frac{\iota^{p+2} d^5}{m_2} + \frac{\iota d^3}{\sqrt{m_2}} + \frac{\iota^{p+3/2} d}{\sqrt{n_1}} + \frac{\iota L r^2 \kappa_1 \log^2 d}{d^{1/6}} \right)^2$$

$$+ \frac{\iota^{p+1} \|g\|_{L^2}^2}{m_1} \cdot \left( \sum_{k=0}^p \eta^{-k} \|\mathbf{B}^\star\|_{\mathrm{op}}^k r^{\frac{p-k}{4}} \right)^2.$$

Here $\mathbf{a}^\star$ satisfies

$$\frac{\|\mathbf{a}^\star\|_2^2}{m_1} \lesssim \iota^p \|g\|_{L^2}^2 \cdot \left( \sum_{k=0}^p \eta^{-k} \|\mathbf{B}^\star\|_{\mathrm{op}}^k r^{\frac{p-k}{4}} \right)^2.$$

The first part of the proof is complete.

### E.1.2  PART2: GENERALIZATION THEORY

Denote the population absolute loss as $\mathcal{L}_1(f, g) = \mathbb{E}_\mathbf{x}[|f(\mathbf{x}) - g(\mathbf{x})|]$. Moreover, we consider a truncated loss function as

$$\ell_\tau(z) = \min(|z|, \tau) \quad \text{and} \quad \mathcal{L}_{1,\tau}(f, g) = \mathbb{E}_\mathbf{x}[\ell_\tau(f(\mathbf{x}) - g(\mathbf{x}))],$$

where $\tau > 0$ is the truncation radius. Moreover, we denote the empirical truncated absolute loss as

$$\hat{\mathcal{L}}_{1,\tau}(f, g) = \frac{1}{n_2} \sum_{\mathbf{x} \in \mathcal{D}_2} \ell_\tau(f(\mathbf{x}) - g(\mathbf{x})).$$

Suppose Algorithm 1 gives rise to a set of parameters $\hat{\theta} = (\hat{\mathbf{a}}, \mathbf{W}^{(1)}, \mathbf{b}^{(1)}, \mathbf{V})$, and we have constructed $\theta^\star = (\mathbf{a}^\star, \mathbf{W}^{(1)}, \mathbf{b}^{(1)}, \mathbf{V})$ that leads to small empirical loss, we decompose the population absolute loss as

$$\mathcal{L}_1(f(\cdot; \hat{\theta}), f^\star) = \underbrace{\hat{\mathcal{L}}_{1,\tau}(f(\cdot; \hat{\theta}), f^\star)}_{L_1} + \underbrace{\mathcal{L}_{1,\tau}(f(\cdot; \hat{\theta}), f^\star) - \hat{\mathcal{L}}_{1,\tau}(f(\cdot; \hat{\theta}), f^\star)}_{L_2}$$

$$+ \underbrace{\mathcal{L}_1(f(\cdot; \hat{\theta}), f^\star) - \mathcal{L}_{1,\tau}(f(\cdot; \hat{\theta}), f^\star)}_{L_3}.$$

Here with a little abuse of notation, we consider $f^\star = g^\star(\mathbf{p})$ for learning the original target function and denote $f^\star = g(\mathbf{p})$ with $g$ being any degree $p$ polynomial for the transfer learning setting. Next, we bound $L_1, L_2$ and $L_3$ respectively.

**Bound $L_1$**  With a little abuse of notation, we denote $\hat{\mathcal{L}}_2(\mathbf{a}) = \hat{\mathcal{L}}_2(\theta)$ for $\theta = (\mathbf{a}, \mathbf{W}^{(1)}, \mathbf{b}^{(1)}, \mathbf{V})$ since we only optimize $\mathbf{a}$ in the second stage. By Proposition 6, we know with high probability, the

empirical $L^2$ loss of $\theta^\star$ is bounded by

$$\hat{\mathcal{L}}_2(\mathbf{a}^\star) = \frac{1}{n_2} \sum_{\mathbf{x} \in \mathcal{D}_2} \left( f(\mathbf{x}; \theta^\star) - f^\star(\mathbf{x}) \right)^2$$

$$\lesssim \|g\|_{L^2}^2 \cdot \frac{r^p}{\lambda_{\min}^2(\mathbf{H})} \cdot \left( \frac{\iota^{p+2} d^5}{m_2} + \frac{\iota d^3}{\sqrt{m_2}} + \frac{\iota^{p+3/2} d}{\sqrt{n_1}} + \frac{\iota L r^2 \kappa_1 \log^2 d}{d^{1/6}} \right)^2$$

$$+ \frac{\iota^{p+1} \|g\|_{L^2}^2}{m_1} \cdot \left( \sum_{k=0}^p \eta^{-k} \|\mathbf{B}^\star\|_{\mathrm{op}}^k \, r^{\frac{p-k}{4}} \right)^2.$$

Here $\mathbf{a}^\star$ satisfies

$$\frac{\|\mathbf{a}^\star\|_2^2}{m_1} \lesssim \iota^p \|g\|_{L^2}^2 \cdot \left( \sum_{k=0}^p \eta^{-k} \|\mathbf{B}^\star\|_{\mathrm{op}}^k \, r^{\frac{p-k}{4}} \right)^2.$$

In the second training stage, let's set the weight decay in the second training stage as

$$\lambda_2 = \lambda = \|\mathbf{a}^\star\|_2^{-2} \|g\|_{L^2}^2 \cdot \left( \frac{r^p}{\lambda_{\min}^2(\mathbf{H})} \cdot \left( \frac{\iota^{p+2} d^5}{m_2} + \frac{\iota d^3}{\sqrt{m_2}} + \frac{\iota^{p+3/2} d}{\sqrt{n_1}} + \frac{\iota L r^2 \kappa_1 \log^2 d}{d^{1/6}} \right)^2 \right.$$

$$\left. + \frac{\iota^{p+1}}{m_1} \cdot \left( \sum_{k=0}^p \eta^{-k} \|\mathbf{B}^\star\|_{\mathrm{op}}^k \, r^{\frac{p-k}{4}} \right)^2 \right)$$

so that the empirical $L^2$ loss is directly bounded by

$$\hat{\mathcal{L}}_2(\mathbf{a}^\star) := \frac{1}{n_2} \sum_{\mathbf{x} \in \mathcal{D}_2} \left( f(\mathbf{x}; \theta^\star) - f^\star(\mathbf{x}) \right)^2 \lesssim \lambda \|\mathbf{a}^\star\|_2^2.$$

We further consider the regularized second-stage training loss to be

$$\hat{\mathcal{L}}_{2,\lambda}(\mathbf{a}) = \frac{1}{n_2} \sum_{\mathbf{x} \in \mathcal{D}_2} \left( f(\mathbf{x}; (\mathbf{a}, \mathbf{W}^{(1)}, \mathbf{b}^{(1)}, \mathbf{V})) - f^\star(\mathbf{x}) \right)^2 + \frac{\lambda}{2} \|\mathbf{a}\|_2^2.$$

Note that this loss is strongly convex, so it has a global minimum $\mathbf{a}^{(\infty)} = \arg\min \hat{\mathcal{L}}_{2,\lambda}(\mathbf{a})$. Thus, we have

$$\mathcal{L}_{2,\lambda}(\mathbf{a}^{(\infty)}) \leq \mathcal{L}_{2,\lambda}(\mathbf{a}^\star) \lesssim \lambda \|\mathbf{a}^\star\|_2^2.$$

Since $\mathcal{L}_{2,\lambda}(\mathbf{a})$ is $\lambda$- strongly convex, and we can write $f(\mathbf{x}; (\mathbf{a}, \mathbf{W}^{(1)}, \mathbf{b}^{(1)}, \mathbf{V})) = \mathbf{a}^\top \Psi(\mathbf{x})$, where $\Psi(\mathbf{x}) = \mathrm{Vec}\left( m_1^{-1} \sigma_1 \left( \eta a_i^{(0)} \langle \mathbf{w}_i, \mathbf{h}^{(1)}(\mathbf{x}) \rangle + b_i^{(1)} \right) \right)$. Therefore, by Lemma 4 and our choice of $\eta$ to ensure $\eta \langle \mathbf{w}_i, \mathbf{h}^{(1)}(\mathbf{x}) \rangle \leq 1$ with high probability, we know with high probability,

$$\lambda_{\max}\left( \nabla_{\mathbf{a}}^2 \hat{\mathcal{L}}_{2,\lambda} \right) \leq \frac{2}{n_2} \sum_{\mathbf{x} \in \mathcal{D}_2} \|\Psi(\mathbf{x})\|_2 \lesssim \frac{1}{m_1}.$$

Thus, $\mathcal{L}_{2,\lambda}(\mathbf{a})$ is $\lambda + \mathcal{O}(\frac{1}{m_1})$- smooth. By choosing the second-stage learning rate $\eta_2 = \Omega(m_1)$, after $T = \widetilde{\mathcal{O}}(\lambda^{-1}) = \mathrm{poly}(d, n, m_1, m_2, \|g\|_{L^2})$ steps, we can reach an iterate $\hat{\mathbf{a}} = \mathbf{a}^{(T)}$ so that

$$\hat{\mathcal{L}}_2(\hat{\mathbf{a}}) \lesssim \hat{\mathcal{L}}_2(\mathbf{a}^\star) \quad \text{and} \quad \|\hat{\mathbf{a}}\|_2 \lesssim \|\mathbf{a}^\star\|_2.$$

Denoting $\hat{\theta} = (\hat{\mathbf{a}}, \mathbf{W}^{(1)}, \mathbf{b}^{(1)}, \mathbf{V})$, it holds that

$$\hat{\mathcal{L}}_{1,\tau}(f(\cdot; \hat{\theta}), f^\star) \leq \frac{1}{n_2} \sum_{\mathbf{x} \in \mathcal{D}_2} \left| f(\mathbf{x}; \hat{\theta}) - f^\star(\mathbf{x}) \right| \leq \sqrt{\mathcal{L}_2(\hat{\mathbf{a}})} \leq \sqrt{\mathcal{L}_2(\mathbf{a}^\star)}.$$

Thus, we have

$$L_1 = \hat{\mathcal{L}}_{1,\tau}(f(\cdot; \hat{\theta}), f^\star) \leq \sqrt{\frac{1}{n_2} \sum_{\mathbf{x} \in \mathcal{D}_2} \left( f(\mathbf{x}; \theta^\star) - f^\star(\mathbf{x}) \right)^2}$$

$$\lesssim \|g\|_{L^2} \cdot \frac{r^{p/2}}{\lambda_{\min}(\mathbf{H})} \cdot \left( \frac{\iota^{p+2} d^5}{m_2} + \frac{\iota d^3}{\sqrt{m_2}} + \frac{\iota^{p+3/2} d}{\sqrt{n}} + \frac{\iota L r^2 \kappa_1 \log^2 d}{d^{1/6}} \right)$$

$$+ \sqrt{\frac{\iota^{p+1} \|g\|_{L^2}}{m_1}} \cdot \left( \sum_{k=0}^p \eta^{-k} \|\mathbf{B}^\star\|_{\mathrm{op}}^k \, r^{\frac{p-k}{4}} \right).$$

Here $\hat{\mathbf{a}}$ satisfies

$$\frac{\|\hat{\mathbf{a}}\|_2^2}{m_1} \lesssim \frac{\|\mathbf{a}^\star\|_2^2}{m_1} \lesssim \iota^p \|g\|_{L^2}^2 \cdot \left( \sum_{k=0}^p \eta^{-k} \|\mathbf{B}^\star\|_{\mathrm{op}}^k \, r^{\frac{p-k}{4}} \right)^2.$$

We assume $\|\mathbf{a}\|_2^2 \leq m_1 B_a^2$, where $B_a$ satisfies

$$B_a^2 \lesssim \iota^p \|g\|_{L^2}^2 \cdot \left( \sum_{k=0}^p \eta^{-k} \|\mathbf{B}^\star\|_{\mathrm{op}}^k \, r^{\frac{p-k}{4}} \right)^2.$$

**Bound $L_2$**  To bound $L_2$, we rely on standard Rademacher complixity analysis. The following lemma provides an upper bound on the Rademacher complixity of the random feature model.

**Lemma 35.** *Let $\mathcal{F} = \{f_\theta : \theta = (\mathbf{a}, \mathbf{W}^{(1)}, \mathbf{b}^{(1)}, \mathbf{V}), \|\mathbf{a}\|_2 \leq \sqrt{m_1}B_a\}$. Recall the empirical Rademacher complexity of $\mathcal{F}$ as*

$$\mathcal{R}_n(\mathcal{F}) = \mathbb{E}_{\sigma \in \{\pm 1\}^n} \left[ \sup_{f \in \mathcal{F}} \frac{1}{n_2} \sum_{i=1}^n \sigma_i f(\mathbf{x}_i) \right],$$

*Here the dataset $\{\mathbf{x}_1, \mathbf{x}_2, \dots, \mathbf{x}_{n_2}\} = \mathcal{D}_2$. Then with high probability, we have*

$$\mathcal{R}_n(\mathcal{F}) \lesssim \frac{B_a}{\sqrt{n_2}}.$$

The proof is provided in Appendix E.2. Since the $\ell_\tau$ is 1-Lipschitz, by standard Rademacher complexity analysis, we have that with high probability that

$$L_2 = \mathbb{E}_{\mathbf{x}} \ell_\tau \Big( f(\mathbf{x}; \hat{\theta}) - f^*(\mathbf{x}) \Big) - \frac{1}{n_2} \sum_{\mathbf{x} \in \mathcal{D}_2} \ell_\tau \Big( f(\mathbf{x}; \hat{\theta}) - f^\star(\mathbf{x}) \Big)$$

$$\lesssim \mathcal{R}_n(\mathcal{F}) + \tau \sqrt{\frac{\iota}{n_2}}$$

$$\lesssim \sqrt{\frac{B_a^2}{n_2}} + \tau \sqrt{\frac{\iota}{n_2}}.$$

**Bound $L_3$**  Finally, we relate the truncated loss $\ell_\tau$ to the $L_1$ population loss.

**Lemma 36.** *By letting $\tau = \Omega(\max(\iota^p, B_a))$, with high probability over $\hat{\theta}$, we have*

$$L_3 = \mathbb{E}_{\mathbf{x}} \left[ \left| f(\mathbf{x}; \hat{\theta}) - f^*(\mathbf{x}) \right| \right] - \mathbb{E}_{\mathbf{x}} \left[ \ell_\tau \Big( f(\mathbf{x}; \hat{\theta}) - f^*(\mathbf{x}) \Big) \right] \leq o \left( \frac{1}{n_1 n_2 m_1 m_2 d} \right).$$

Here we recall that $n = n_1 + n_2$. The proof is provided in Appendix E.2.

**Put the loss together**   By invoking the upper bound of $L_1$, $L_2$ and $L_3$ and plugging the values of $\tau, \eta, \|\mathbf{B}^\star\|_{\mathrm{op}}, \lambda_{\min}(\mathbf{H}), B_a, L$ and $\|g\|_{L^2}$, we have

$$
\mathbb{E}_{\mathbf{x}}\left[\left|f(\mathbf{x};\hat{\theta}) - f^*(\mathbf{x})\right|\right] = L_1 + L_2 + L_3
$$

$$
\lesssim \frac{1}{n_2}\sum_{\mathbf{x}\in\mathcal{D}_2}\ell_\tau(f(\mathbf{x};\theta^\star) - f^\star(\mathbf{x})) + \sqrt{\frac{B_a^2}{n_2}} + \tau\sqrt{\frac{\iota}{n_2}} + o\left(\frac{1}{n_1 n_2 m_1 m_2 d}\right)
$$

$$
\lesssim \|g\|_{L^2} \cdot \frac{r^{p/2}}{\lambda_{\min}(\mathbf{H})} \cdot \left(\frac{\iota^{p+2}d^5}{m_2} + \frac{\iota d^3}{\sqrt{m_2}} + \frac{\iota^{p+3/2}d}{\sqrt{n_1}} + \frac{\iota L r^2 \kappa_1 \log^2 d}{d^{1/6}}\right)
$$

$$
+ \sqrt{\frac{\iota^{p+1}\|g\|_{L^2}}{m_1}} \cdot \left(\sum_{k=0}^{p}\eta^{-k}\|\mathbf{B}^\star\|_{\mathrm{op}}^k r^{\frac{p-k}{4}}\right) + \sqrt{\frac{B_a^2}{n_2}} + \tau\sqrt{\frac{\iota}{n_2}}.
$$

$$
\lesssim \frac{r^{p/2}}{\lambda_{\min}(\mathbf{H})} \cdot \left(\frac{\iota^{p+2}d^5}{m_2} + \frac{\iota d^3}{\sqrt{m_2}} + \frac{\iota^{p+3/2}d}{\sqrt{n_1}} + \frac{\iota L r^2 \kappa_1 \log^2 d}{d^{1/6}}\right)
$$

$$
+ \sqrt{\frac{\iota^{6p+1}r^{p/2}\kappa_2^{2p}(r^{1/4}\vee\lambda_{\min}^{-1}(\mathbf{H}))^p}{m_1}}
$$

$$
+ \sqrt{\frac{\iota^{6p+1}r^{p/2}\kappa_2^{2p}(r^{1/4}\vee\lambda_{\min}^{-1}(\mathbf{H}))^p}{n_2}}
$$

$$
= \widetilde{\mathcal{O}}\left(\sqrt{\frac{r^p\kappa_2^{2p}}{\min(n_2, m_1)}} + \sqrt{\frac{d^6 r^{p+1}}{m_2}} + \sqrt{\frac{d^2 r^{p+1}}{n}} + \frac{r^{p+2}\kappa_1}{d^{1/6}}\right).
$$

The proof is complete.

### E.2   OMITTED PROOFS IN APPENDIX E.1

*Proof of Lemma 35.* Given $\theta = (\mathbf{a}, \mathbf{W}^{(1)}, \mathbf{b}^{(1)}, \mathbf{V})$, since we can write

$$
f_\theta(\mathbf{x}) = \mathbf{a}^\top\Psi(\mathbf{x}), \quad \text{where} \quad \Psi(\mathbf{x}) = \mathrm{Vec}\left(m_1^{-1}\sigma_1\left(\eta a_i^{(0)}\langle\mathbf{w}_i, \mathbf{h}^{(1)}(\mathbf{x})\rangle + b_i^{(1)}\right)\right).
$$

By Proposition 4 and our choice of $\eta$ to ensure $\left|\eta\langle\mathbf{w}_i, \mathbf{h}^{(1)}(\mathbf{x})\rangle\right| \le 1$ with high probability for any $\mathbf{x}\in\mathcal{D}_2$, we obtain that for any $i\in[m_1]$ and $\mathbf{x}\in\mathcal{D}_2$,

$$
\left|\eta a_i^{(0)}\langle\mathbf{w}_i, \mathbf{h}^{(1)}(\mathbf{x})\rangle + b_i^{(1)}\right| \le a_i^{(0)}\left|\eta\langle\mathbf{w}_i, \mathbf{h}^{(1)}(\mathbf{x})\rangle\right| + b_i^{(1)} \lesssim 1.
$$

Thus, $\|\Psi(\mathbf{x})\|_2^2 \le m_1^{-1}$. by the standard linear Rademacher bound, with high probability, the empirical Rademacher complexity is upper bounded by

$$
\mathcal{R}_n(\mathcal{F}) \lesssim \frac{\sqrt{m_1}B_a}{n_2}\sqrt{\sum_{\mathbf{x}\in\mathcal{D}_2}\|\Psi(\mathbf{x})\|_2^2} \le \frac{B_a}{\sqrt{n_2}}.
$$

The proof is complete. □

*Proof of Lemma 36.* We can bound the difference between $\ell_\tau$ and $L_1$ loss by

$$
\mathbb{E}_{\mathbf{x}}\left[\left|f(\mathbf{x};\hat{\theta}) - f^*(\mathbf{x})\right|\right] - \mathbb{E}_{\mathbf{x}}\left[\ell_\tau\left(f(\mathbf{x};\hat{\theta}) - f^*(\mathbf{x})\right)\right]
$$

$$
\le \mathbb{E}_{\mathbf{x}}\left[\left|f(\mathbf{x};\hat{\theta}) - f^*(\mathbf{x})\right|\mathbf{1}\left\{\left|f(\mathbf{x};\hat{\theta}) - f^*(\mathbf{x})\right| \ge \tau\right\}\right]
$$

$$
\le \sqrt{\mathbb{E}_{\mathbf{x}}\left[\left|f(\mathbf{x};\hat{\theta}) - f^*(\mathbf{x})\right|^2\right]\Pr\left[\left|f(\mathbf{x};\hat{\theta}) - f^*(\mathbf{x})\right| \ge \tau\right]}
$$

$$
\lesssim \sqrt{\mathbb{E}_{\mathbf{x}}\left[\left(f(\mathbf{x};\hat{\theta})^2 + f^\star(\mathbf{x})^2\right)\right]\left[\Pr\left[\left|f(\mathbf{x};\hat{\theta})\right| \ge \tau/2\right] + \Pr\left[|f^\star(\mathbf{x})| \ge \tau/2\right]\right]} \tag{36}
$$

Recall that we can write

$$f(\mathbf{x}; \hat{\theta}) = \hat{\mathbf{a}}^\top \Psi(\mathbf{x}), \text{ where } \Psi(\mathbf{x}) = \text{Vec}\left(m_1^{-1}\sigma_1\left(\eta a_i^{(0)}\langle \mathbf{w}_i, \mathbf{h}^{(1)}(\mathbf{x})\rangle + b_i^{(1)}\right)\right).$$

By following the proof of Lemma 35 and applying Proposition 4 for one single sample point $\mathbf{x}$ (instead of the whole set $\mathcal{D}_2$), we know with high probability over $\mathbf{V}$, $\mathbf{w}$ and $\mathcal{D}_1$ (we denote this event by $E_1$), we have for any $i \in [m_1]$,

$$\left|\eta\langle \mathbf{w}_i, \mathbf{h}^{(1)}(\mathbf{x})\rangle\right| \le 1$$

holds with high probability on $\mathbf{x}$. Also, since for any $i \in [m_1]$, $\mathbf{w}_i \sim \mathcal{N}(\mathbf{0}_{m_2}, \mathbf{I}_{m_2})$, we know $\|\mathbf{w}_i\|_2 \lesssim \sqrt{m_2\iota}$ for any $i \in [m_1]$ with high probability. We denote this joint event on $\mathbf{w}_1, \mathbf{w}_2, \dots, \mathbf{w}_{m_1}$ by $E_2$. Thus, conditional on events $E_1$ and $E_2$, we have

$$\left|f(\mathbf{x}; \hat{\theta})\right| \lesssim \frac{\|\hat{\mathbf{a}}\|_2}{\sqrt{m_1}} \text{ with high probability on } \mathbf{x}.$$

We denote this conditional event by $E_{x,1}$. Moreover, since $\eta = C\iota^{-5}m_2^{-1/2}d^6$, we have

$$\left|f(\mathbf{x}; \hat{\theta})\right| \le \frac{\|\hat{\mathbf{a}}\|_2}{m_1} \sum_{j=1}^{m_1}\left(\eta\|\mathbf{w}_j\|_2\left\|\frac{1}{n_2}\sum_{i=1}^{n} K_{m_2}^{(0)}(\mathbf{x}_i, \mathbf{x}')\mathbf{h}^{(0)}(\mathbf{x}_i)\right\|_2 + 3\right)$$

$$\le \frac{\sqrt{m_2\iota}\|\hat{\mathbf{a}}\|_2}{m_1} \sum_{j=1}^{m_1}\frac{\eta}{n}\sum_{i=1}^{n} C_\sigma^2 \cdot \sqrt{m_2}C_\sigma + 3$$

$$\lesssim \sqrt{m_2}d^6\|\hat{\mathbf{a}}\|_2$$

holds for any $\mathbf{x}$. Moreover, since $f^\star(\mathbf{x})$ is a degree-$2p$ polynomial of $\mathbf{x}$, we know by Lemma 8, with probability at least $1 - \exp(-\iota)$, we have $|f^\star| \le C_f\iota^p$ for sufficiently large $C_f > 0$. Besides, we have $\mathbb{E}_\mathbf{x}\left[f^\star(\mathbf{x})^2\right] \lesssim 1$. Altogether, conditional on $E_1$ and $E_2$, by choosing $\tau = C'\max(\iota^p, m_1^{-1/2}\|\hat{\mathbf{a}}\|_2) = \Omega(\max(\iota^p, B_a)$ for some sufficiently large $C$, we have

$$\mathbb{E}_\mathbf{x}\left[\left(f(\mathbf{x}; \hat{\theta})^2 + f^\star(\mathbf{x})^2\right)\right]\left[\text{Pr}\left[\left|f(\mathbf{x}; \hat{\theta})\right| \ge \tau/2\right] + \text{Pr}\left[|f^\star(\mathbf{x})| \ge \tau/2\right]\right]$$

$$\lesssim \left(m_2 d^{12}\|\hat{\mathbf{a}}\|_2^2 + 1\right)(\text{Pr}[\mathbf{x} \notin E_{x,1}] + \text{Pr}[\mathbf{x} \notin E_{x,2}])$$

$$\lesssim o\left(\frac{1}{d^2 m_1^2 m_2^2 n_1^2 n_2^2}\right).$$

The last inequality holds because of the definition of high probability events and the choice of $\iota$ with $\iota = C\log(dm_1m_2n_1n_2)$ for sufficiently large $C$. Plugging the result into (36) concludes our proof. $\square$

