# OpenReview forum: "Learning Hierarchical Polynomials of Multiple Nonlinear Features"
_ICLR.cc/2025/Conference — ICLR 2025 Poster_

### Official Review · Reviewer_Xpr2 · 2024-11-04

**Soundness:** 3
**Presentation:** 3
**Contribution:** 3
**Rating:** 8
**Confidence:** 4

**Summary:**

This work studies the problem of learning functions $f^* : \mathbb{R}^d \to \mathbb{R}$ of the form $f^* = g^* \circ p$ where $p : \mathbb{R}^d \to \mathbb{R}^r$ returns $r$ quadratic features of the input, and $g^* : \mathbb{R}^r \to \mathbb{R}$ is a polynomial of degree $p$. By analyzing a gradient-based algorithm for training a three-layer neural network with specific choices of activation, the authors prove that a sample complexity of $\tilde{O}(d^4)$ is sufficient for learning this function, as opposed to the $\Omega(d^{2p})$ sample complexity lower bound for learning with rotationally invariant kernels.

**Strengths:**

Presenting a guarantee for learning a polynomial of multiple quadratic features using three-layer neural networks is a new direction in the literature which extends beyond the guarantees for learning multi-index models with two-layer neural networks and learning polynomials of a single non-linear feature which existed in prior works. More broadly, this work contributes to finding settings in which networks deeper than two layers are natural for learning certain function classes, which is an area of particular interest for the community.

**Weaknesses:**

* The feature learning regime considered in this paper is limited in the sense that the first layer weights are fixed at random and not trained. Training these weights could lead to learning more complex hierarchies, such as hierarchies of multi-index models, although admittedly challenging for a theoretical analysis.

* The choice of activation for the second layer is rather specific, and choosing a different activation for each layer seems a bit odd when compared with standard three-layer neural networks. However, I believe there might be a way to choose the same activation as my impression is that $\sigma_1$ only has a constant part and no $Q_1$ part, thus one only needs to introduce a fixed bias to the output of the network. If this interpretation is correct this might be worth mentioning in the main text.

* While the statistical model is naturally suitable for three-layer networks, I could not find any lower bounds in this submission which explain that learning this function with a two-layer neural network is provably less efficient (in terms of samples or computation). I am wondering if the authors could use the results of prior works to show a statistical or computational lower bound holding in this setting.

**Questions:**

* What is the main technical challenge in directly generalizing the result of Nichani et al., 2023 to the setting of having multiple non-linear features? Specifically, if $f^* = g \circ p$ where $p = (p_1,...,p_r)$ and each $p_j$ is a degree $k$ polynomial of $x$, and $g$ is a degree $q$ polynomial, is it possible to argue that in the regime of constant $r$, the sample complexity of learning $f^*$ with three-layer neural networks is $d^k$, as opposed to $d^{kq}$ of kernel methods?

* On the discussion of learning linear features, a relevant work may be [1] that considers learning general multi-index models with two-layer neural networks via a noisy gradient descent algorithm. For single-index models, while the authors have included works in the CSQ framework, there are more recent advances on gradient-based learning with two-layer neural networks that go beyond the CSQ framework [2, 3], or consider structured rather than isotropic input covariance [4, 5].

---
[1] A. Mousavi-Hosseini et al. "Learning Multi-Index Models with Neural Networks via Mean-Field Langevin Dynamics". arXiv 2024.

[2] Y. Dandi et al. "The benefits of reusing batches for gradient descent in two-layer networks: Breaking the curse of information and leap exponents". ICML 2024.

[3] J. D. Lee et al. "Neural network learns low-dimensional polynomials with SGD near the information-theoretic limit". arXiv 2024.

[4] A. Mousavi-Hosseini et al. "Gradient-Based Feature Learning under Structured Data". NeurIPS 2023.

[5] J. Ba et al. "Learning in the presence of low-dimensional structure: a spiked random matrix perspective". NeurIPS 2023

---

> ### Author Response · Authors · 2024-11-22
> **Response to Reviewer Xpr2**
>
> We thank the reviewer for the valuable comments and suggestions.
>
> >The feature learning regime considered in this paper is limited in the sense that the first layer weights are fixed at random and not trained. Training these weights could lead to learning more complex hierarchies, such as hierarchies of multi-index models, although admittedly challenging for a theoretical analysis.
> >The choice of activation for the second layer is rather specific, and choosing a different activation for each layer seems a bit odd when compared with standard three-layer neural networks. However, I believe there might be a way to choose the same activation as my impression is that
> >While the statistical model is naturally suitable for three-layer networks, I could not find any lower bounds in this submission which explain that learning this function with a two-layer neural network is provably less efficient (in terms of samples or computation). I am wondering if the authors could use the results of prior works to show a statistical or computational lower bound holding in this setting.
> >What is the main technical challenge in directly generalizing the result of Nichani et al., 2023 to the setting of having multiple non-linear features?
>
> Regarding the issue of assumptions, the optimality of sample complexity bound and comparison with (Nichani et al., 2023) , please see the **general response** for a detailed discussion.
>
> >σ1 only has a constant part and no Q1  part, thus one only needs to introduce a fixed bias to the output of the network. If this interpretation is correct this might be worth mentioning in the main text.
> We thank the reviewer for pointing out this. We will add a discussion of this equivalence between the assumption on $\sigma_1$ and adding a bias term.
> >Specifically, if f∗=g∘pwhere p=(p1,...,pr) and each pj is a degree k polynomial of x, and g is a degree q polynomial, is it possible to argue that in the regime of constant r, the sample complexity of learning f∗ with three-layer neural networks is dk, as opposed to dkq of kernel methods?
>
> We believe that our analytic framework **applies to general nonlinear features**, not limited to quadratic features, such as the universality result (Lemma 1) and the proof idea proposed in Figure 1 and the entire algorithm. As the reviewer proposed, if we consider multiple degree-$q$ features and the target functions also satisfy similar assumptions as Assumptions 1-3 and the inner activation function makes emphasis on the corresponding degree ($q$) of the feature (instead of the quadratic one), our theory could be generalized to these settings with similar analysis. We just use quadratic features as **an example** of our theoretical framework for multiple nonlinear feature learning.
>
> >On the discussion of learning linear features, a relevant work may be [1] that considers learning general multi-index models with two-layer neural networks via a noisy gradient descent algorithm. For single-index models, while the authors have included works in the CSQ framework, there are more recent advances on gradient-based learning with two-layer neural networks that go beyond the CSQ framework [2, 3], or consider structured rather than isotropic input covariance [4, 5].
>
> We thank the reviewer for mentioning the recent works on feature learning. We agree that there are emerging works beyond CSQ framework. We will add a discussion on these works in our revised version, which will come out in one or two days.

---

> > ### Comment · Reviewer_Xpr2 · 2024-12-02
> >
> > Thank you for your detailed response. I think that despite some contrived modifications and assumptions, which are inevitable in this line of work with current tools and techniques, overall this paper has a solid contribution. I have updated my score accordingly.

---

### Official Review · Reviewer_BXau · 2024-11-04

**Soundness:** 4
**Presentation:** 4
**Contribution:** 4
**Rating:** 8
**Confidence:** 4

**Summary:**

The authors prove that three-layer networks can learn nonlinear, multi-index targets more efficiently than kernel methods. Quite a bit of literature already exists on 2- and 3-layer networks learning single-index targets of linear and non-linear features, and also multi-index targets of linear features. To the best of my knowledge this is the first set of results for learning multiple, non-linear features.

The analysis is valid of an algorithm that does a single step on the first layer weights, W, and then completely trains the rest of the network. The authors use this two-stage training methodology to also prove a transfer learning result, where the link function is changed.

Finally the authors provide some empirical evidence that supports their theory. In particular, their approach appears to improve sample complexity over a random feature model. They also support their findings regarding efficient transfer learning and the reconstruction of quadratic features.

**Strengths:**

This is a strong submission in most respects: relevance, rigor, and clarity. I have minor comments and questions below.

**Weaknesses:**

1. Some of the assumptions might be too restrictive for the results to be relevant for practice (eg parts of Assumption 1, and Assumption 4).
2. Algorithm 1 does but a single update on W. Then parameter a is trained on multiple steps. While this style of analysis is common in the area, there already exist articles that include analysis of many-step gradient descent for all parameters.

**Questions:**

1. The “balanced features” part of Assumption 1, where the operator norm of Ai satisfies a particular bound. This bound essentially tells us that each feature cannot depend on a sparse subset of inputs. Can you discuss the implications of that assumption and pertinence to real distributions?
2. Similarly, can you discuss the implications of Assumption 4, especially the Gegenbauer expansion? Activation functions used in practice are not even, and many of them are actually odd.
3. Do you expect that the d^4 dependence on dimension is necessary for quadratic features?

---

> ### Author Response · Authors · 2024-11-22
> **Response to Reviewer BXau**
>
> We thank the reviewer for the valuable comments and suggestions. We would appreciate it if you could champion our paper during the discussions!
> >Some of the assumptions might be too restrictive for the results to be relevant for practice (eg parts of Assumption 1, and Assumption 4).
> >The “balanced features” part of Assumption 1, where the operator norm of Ai satisfies a particular bound. This bound essentially tells us that each feature cannot depend on a sparse subset of inputs. Can you discuss the implications of that assumption and pertinence to real distributions?
> >Similarly, can you discuss the implications of Assumption 4, especially the Gegenbauer expansion? Activation functions used in practice are not even, and many of them are actually odd.
> >Do you expect that the d^4 dependence on dimension is necessary for quadratic features?
>
> Regarding the issue of assumptions and the optimality of sample complexity bound (Weakness 1 and Questions 1-3), please see our **general response** for a detailed discussion.
>
> >Algorithm 1 does but a single update on W. Then parameter a is trained on multiple steps. While this style of analysis is common in the area, there already exist articles that include analysis of many-step gradient descent for all parameters.
>
> We agree that there exist works that analyze many-step gradient descent, but when analyzing hierarchical targets of multiple non-linear features, this could be quite challenging and requires further assumptions or modification on the network architecture like [Wang, 2023]. We believe this is a good future direction to analyze multi-step GD on the weights.

---

### Official Review · Reviewer_okDB · 2024-11-04

**Soundness:** 3
**Presentation:** 2
**Contribution:** 2
**Rating:** 6
**Confidence:** 2

**Summary:**

In this work, the authors studies theoretically the setting of using three-layer neural networks to learn a type of "multi-index quadratic models" under a uniform data distribution on the sphere. With suitable activation functions and layer-wise training, the model is proved to be able to learn both the output function and the quadratic features with a sample complexity that scales as $d^4$.

**Strengths:**

The type of model studied in this paper generalizes the one considered by Nichani et al. (2023) by allowing multiple quadratic features instead of just one in the first layer. To prove the learning guarantee, the authors introduced a nice technical result on the distribution of multiple nonlinear functions of Gaussian inputs, which is interesting mathematically and novel to my knowledge. The theoretical results appear to be sound though they have not been thoroughly checked.

**Weaknesses:**

**Beyond the mathematical and technical**: It could be argued that the series of assumptions on the layer-wise learning algorithm, the activation functions, the data distribution, etc. are quite specific and restrictive, and the contribution of this work (relative to e.g. Nichani et al., 2023 and Wang et al., 2023, which share much of a common analytical framework) resides more in the improved mathematical results than advancements in new learning theory insights.

**Some confusion regarding the proof of Theorem 1**: The authors' description of the second stage in the proof of Theorem 1 gives the impression that it breaks down into two parts: 1) showing the existence of an $a^*$ achieving small empirical loss, and 2) using Rademacher complexity analysis to bound the generalization error of $a^*$. But they alone wouldn't be sufficient since we know only the existence of the $a^*$ but not its learnability by the proposed algorithm. And indeed, the actual proof steps taken by the authors is different (and correctly so), where they use the strong convexity of the loss function to show that an alternative solution $\hat{a}$ can be learned by gradient descent and achieves low empirical loss as well as generalization error. Even though $a^*$ still makes its appearance in the proof argument in Section D.1.2, I suspect that it is not truly necessary and only complicates the proof. If this assessment is correct, it would imply that Proposition 2 is redundant for the proof of Theorem 1, which should be commented on in order to avoid confusion.

Minor typos:
1) In Section 1.1 on Page 2, "outer layer can achieves" --> "outer layer can achieve".
2) In Section 4.1 on Page 7, "trained presentations representations" --> "trained representations".
3) The links referencing Proposition 6 are misplaced.

**Questions:**

See above.

---

> ### Author Response · Authors · 2024-11-22
> **Response to Reviewer okDB**
>
> We thank the reviewer for the valuable comments and suggestions.
> >Beyond the mathematical and technical: It could be argued that the series of assumptions on the layer-wise learning algorithm, the activation functions, the data distribution, etc. are quite specific and restrictive, and the contribution of this work (relative to e.g. Nichani et al., 2023 and Wang et al., 2023, which share much of a common analytical framework) resides more in the improved mathematical results than advancements in new learning theory insights.
>
> Regarding the assumptions and comparison with previous works， please see our **general response**.
>
> >Some confusion regarding the proof of Theorem 1: The authors' description of the second stage in the proof of Theorem 1 gives the impression that it breaks down into two parts: 1) showing the existence of an achieving small empirical loss, and 2) using Rademacher complexity analysis to bound the generalization error of . But they alone wouldn't be sufficient since we know only the existence of the  but not its learnability by the proposed algorithm. And indeed, the actual proof steps taken by the authors is different (and correctly so), where they use the strong convexity of the loss function to show that an alternative solution  can be learned by gradient descent and achieves low empirical loss as well as generalization error. Even though still makes its appearance in the proof argument in Section D.1.2, I suspect that it is not truly necessary and only complicates the proof. If this assessment is correct, it would imply that Proposition 2 is redundant for the proof of Theorem 1, which should be commented on in order to avoid confusion
>
> We respectfully disagree with the reviewer’s opinion that Proposition 2 is redundant for the proof of Theorem 1. To prove Theorem 1, we need to first construct an $a^{\star}$ that has both **a) low norm** and **b) low empirical error**. This implies that if we do GD on the regularized loss with appropriate choice of $\lambda$, we will converge to an $\hat a$ that has both low norm (the same scale as $a^{\star}$) and low empirical error. To conclude, we show that all the weights $a$ with low norm have a small generalization gap, which implies that the $\hat a$ we have learned has small test error. Thus, Proposition 2 is **necessary for concluding Theorem 1**. We will rewrite this part to make it clearer.
>
> >Minor typos
>
> We thank the reviewer for pointing out the typos. We will fix all the typos and make notations consistent for a better readability.

---

### Official Review · Reviewer_bUVE · 2024-11-04

**Soundness:** 3
**Presentation:** 3
**Contribution:** 3
**Rating:** 6
**Confidence:** 4

**Summary:**

The paper studies hierarchical feature learning in fully-connected neural networks when trained with gradient descent. In particular, the authors consider a nonlinear (quadratic) multi-index model with $f^*(x) = g^*(x^\top A_1 x, \ldots, x^\top A_r x)$ for $x \in \mathcal{S}^{d-1}$ and $r\ll d$. Under approximate orthogonality, well-balancedness conditions of the quadratic features, and additional well-conditioned assumptions on the link function, the authors show that a layer-wise gradient descent method on a three-layer neural network (first layer is never trained, and activations are carefully chosen) can learn $f^*$ with roughly $\tilde{O}(d^4 + d^2r^{p+1})$ (ignoring $\kappa$) to irreducible error $O(r^{p+2}/d^{1/6})$, and approximately recover the $r$-dimensional subspace to allow for transfer learning to a different link function.

**Strengths:**

- The paper studies an interesting model of feature learning, generalizing prior work that consider either linear multi-index settings or non-linear single-index models.
- The authors show that gradient descent recovers the subspace which allows them to give transfer learning results for any link function.
- To prove the results, the authors prove a new universality theorem for multiple-dimensions, generalizing previous known results. Though the proof seems to follow from existing tools in high dimensional probability, it seems novel.

**Weaknesses:**

- The paper makes several assumptions and tweaks that seem to be done to make the proofs easier: layer-wise training, removing lower order terms in activation functions, taking almost orthonormal equally important features, etc. Most of these are standard is deep learning theory, but the activation assumption seems more artificial, and tailored to make the analysis easier. Some assumptions (Assumptions 2 and 3) can potentially be relaxed if the goal is only getting low loss and not subspace recovery.
- Compared to prior work (Nichani et al. 2022) that was for general hierarchal functions, this paper is narrow in its scope and relies heavily on tools already available in this prior work, while requiring additional assumptions. I do understand that the prior work did not look at multiple quadratic features, but limiting to quadratic seems like a limitation of the techniques.
- The experiments are limited to following the theoretical assumptions precisely. This is not very insightful. If the experiments are to remain synthetic, then it would be worthwhile seeing how the scaling with different parameters in the bounds shows up in experiments. Also, how much can you relax the assumptions, and still get the same empirical results. It would also be worthwhile to come up with a real-world dataset where this quadratic model makes sense, and gains of three vs two layers can be shown.

**Questions:**

- Why did the authors consider only uniform over the sphere compared to Gaussian, which is more commonly studied in this literature? Do the results extend to more general distributions?
- There is no lower bound on the optimality of the dependence on $d$ and $\kappa$ in the bound. It should be possible to get $d^2$ potentially. What dependence on $\kappa$ is optimal?
- Figure 1 is not very easy to understand. Might be helpful to add more explanation to the caption and label the different blocks/terms. It took me a while to parse it and I did not find it very helpful to understand the proof.
- The irreducible error term of $O(r^{p+2}/d^{1/6})$ is not discussed. This is much larger than the $\frac{\kappa^{1/3}}{d^{1/6}}$ term in Niching et al. (2022) and requires $r$ to be much smaller than $d$. Is this necessary?

---

> ### Author Response · Authors · 2024-11-22
> **Response to Reviewer bUVE**
>
> We thank the reviewer for the valuable comments and suggestions.
> >The paper makes several assumptions and tweaks that seem to be done to make the proofs easier: layer-wise training, removing lower order terms in activation functions, taking almost orthonormal equally important features, etc. Most of these are standard is deep learning theory, but the activation assumption seems more artificial, and tailored to make the analysis easier. Some assumptions (Assumptions 2 and 3) can potentially be relaxed if the goal is only getting low loss and not subspace recovery.
>
> We acknowledge that these assumptions serve to simplify technicalities, but more importantly, better illustrate the multiple feature learning process without compromising the essential nature of feature learning. Please see our **general response** on the assumptions. Moreover, we agree that Assumptions 2 and 3 can potentially be relaxed to some extent , such as a larger (but still near constant) tolerance factor of $\kappa_1$ and $\kappa_2$. However, we disagree that we can achieve low loss without  subspace recovery. Only when we **fully capture the subspace** spanned by all the features, we can **efficiently approximate and learn** any polynomials building upon these features. Otherwise, we may suffer from a much larger sample complexity like $O(d^{2p})$ in kernel methods.
>
> > Compared to prior work (Nichani et al. 2022) that was for general hierarchal functions, this paper is narrow in its scope and relies heavily on tools already available in this prior work, while requiring additional assumptions. I do understand that the prior work did not look at multiple quadratic features, but limiting to quadratic seems like a limitation of the techniques.
>
> We acknowledge that we do not provide a general hierarchical theoretical result like Theorem 1 in (Nichani et al. 2023), but we think this kind of result does not provide information and reveal the essence of feature learning until it is applied in **specific cases** to show that the components involved in the error term like $Kf^{\star}$ in (Nichani et al. 2023) can be much smaller compared with fix-feature methods. Moreover, the specific features considered in (Nichani et al. 2023) are also limited in quadratic features. Thus, in terms of end-to-end results, we do not restrict our scope.  Please also see our **general response** for a detailed discussion.
>
> Moreover, we remark that most of our techniques are **not limited in quadratic features**, such as the universality result (Lemma 1), the proof idea proposed in Figure 1 and the entire algorithm. If target functions of high-degree features also satisfy similar assumptions as Assumption 1-3 and the inner activation function makes emphasis on the corresponding degree of the feature, our theory could be generalized to these settings with similar analysis. We just use quadratic features as an example of our theoretical framework of multiple nonlinear feature learning.
>
> >The experiments are limited to following the theoretical assumptions precisely. This is not very insightful. If the experiments are to remain synthetic, then it would be worthwhile seeing how the scaling with different parameters in the bounds shows up in experiments. Also, how much can you relax the assumptions, and still get the same empirical results. It would also be worthwhile to come up with a real-world dataset where this quadratic model makes sense, and gains of three vs two layers can be shown.
>
> We thank the reviewer for pointing out the limitations of our experiments. We plan to conduct more thorough experiments to relax the assumptions and see the networks’ performance, which we will release before the end of the discussion.
>
> >Why did the authors consider only uniform over the sphere compared to Gaussian, which is more commonly studied in this literature? Do the results extend to more general distributions?
>
> The uniform sphere distribution and Gaussian distribution are **quite similar in high-dimensional** case ($d \rightarrow \infty$), they all represent good isotropic distributions. Compared with Gaussian variables, the uniform over the sphere distribution has neater properties (some orthogonality) on **kernel expansion**  in the $L^2$ space, as introduced in Section 2.3. Using Gaussian distribution for analysis may only differ an $o_d(1)$ term compared with  the uniform sphere distribution in most of the cases. The **essence of the feature learning is the same**, not decided by what type of isotropic data we use.

---

> > ### Comment · Reviewer_bUVE · 2024-11-27
> >
> > I thank the authors for their detailed response, and for clarifying my questions. I still have the following concerns:
> > - It is standard in learning theory problems to achieve low loss without subspace recovery. If the weight matrix is ill-conditioned, you may be able to get low loss without ever recovering all the directions. This allows for relaxed assumptions on the features, at the cost of making the analysis more complex.
> > - The assumption on the activation being tailored to the underlying feature structure (quadratic) is still rather strong imo. As the authors point out, this could be modified for different feature structures, but I believe you may need a different assumption for different features. A more generic assumption on the activation, that works for a broad class of features would make the results in the paper stronger.
> > - The experimental results are still lacking, so it is hard for me to predict how they will differ from the proposed bounds.
> > - Sample complexity dependence is sub-optimal. As the authors point out, this is perhaps unavoidable by their one-step gradient techniques, but that's not a satisfying response given that prior works have improved this in similar settings.
> > - To clarify on irreducible term, I agree with the assumption that $r < d$ and you would need $r^p$ samples, but your work requires $r < d^{1/6p}$ where $d$ is the dimension. I don't see why this is necessary here.
> >
> > While I agree with the authors that the results are more general than prior work and involve new tools, I still feel that the work lacks technically novel insight on feature learning with neural networks. Either relaxing the assumptions, or improving the sample complexity would make this paper stronger. Further emphasizing what new insights we get from this analysis, would be helpful.

---

> > > ### Author Response · Authors · 2024-11-28
> > > **Response part 1**
> > >
> > > We thank the reviewer for the response.
> > > > It is standard in learning theory problems to achieve low loss without subspace recovery. If the weight matrix is ill-conditioned, you may be able to get low loss without ever recovering all the directions. This allows for relaxed assumptions on the features, at the cost of making the analysis more complex.
> > >
> > > We remark that out  asssumptions of nondegenerate expected Hessian does not lose generality and the relaxation on this assumption will actually make us repeat the process of feature learning. This is because we have already considered a feature learning case (from $x\in \mathbb{R}^d$ to a low-dimensional feature subspace $p\in \mathbb{R}^{r}$ ). If we consider a ill-conditioned (say rank $r' \le r$) matrix, we may consider another feature learning process to extract the $r'$ directions.
> > >
> > > >The assumption on the activation being tailored to the underlying feature structure (quadratic) is still rather strong imo. As the authors point out, this could be modified for different feature structures, but I believe you may need a different assumption for different features. A more generic assumption on the activation, that works for a broad class of features would make the results in the paper stronger.
> > >
> > > We have argued in the general response that Assumption 4 mentioned by the reviewer to be not realistic **can actually be relaxed** to a general one (with constant and linear component). We make assumption 4 just for a cleaner and centered learned weights to simplify our analysis, and the fundamental intuition is that the **quadratic component  $Q_2$ in the activation function plays the dominant role**. We give an informal proof intuition (summarized from Appendix C.1 ). The first gradient step turn the weights into the form
> > > $$ h^{(1)}(x',v) \approx \sum_{i,j \ge 2}\frac{c_i^2c_j}{d^{i}} \mathbb{E}_x[f^{\star}(x) Q_i(x\cdot x') Q_j(x \cdot v) ],$$
> > > where  the component $i=j=2$ will recover the quadratic features by our universality theory:
> > > $$\mathbb{E}_x[f^{\star}(x) Q_2(x\cdot x') Q_2(x \cdot v) ] \approx d^{-4} \mathbf{p}(v)^{\top} \mathbf{H} \mathbf{p}(x'),$$
> > > where $\mathbf{p} \in \mathbb{R}^{r}$ is the $r$ quadratic features and $ \mathbf{H}$ is the expected Hessian.
> > >
> > > (This is also why we neeed $d^4$ samples, because we are extracting the degree $2+2=4$ component from the targets using the empirical gradients). We  have proven in the Appendix C.2 and C.3 that the $(i,j)=(2,2)$ term is the **dominant one** under our assumptions on the targerts.
> > > Let's consider the case we relax the assumption on the activation to add back the constant and linear term ($i\le 1$ or $j \le 1$ ) .
> > > 1. The magnitude of all the terms with $i+j >4$ diminishes to $o_d(1)$ compared with the term of $(i,j)=(2,2)$
> > > 2. For the case $i+j$ is odd, the expectation is zero because $f$ has only even-degree components under our assumptions.
> > > 3. The term of $(i,j)=(0,0)$ is also $o_d(1)$ compared with the term of $(i,j)=(2,2)$ because we have asssumed (in Assumption 2 and 3) the information exponent and leap index to be basically $4$.
> > > 4. The terms of $(i,j)=(1,3)$, $(3,1)$  have the same magnitude of $(i,j)=(2,2)$, but after we apply the linear product $B^{\star}$ in Proposition, these two terms will cancel by the orthongonality of degree-$j$ term of $ Q_j(x \cdot v) $ and the degree-2 terms in $B^{\star}$. This is because we simply set each row of $B^{\star}$ to be proportional $ \mathbf{H}^{-1}\mathbf{p}(v)$ which only has degree $2$ terms. Thus, we still have $$B^{\star} h^{(1)} (x')\approx  \mathbf{H}^{-1}  \mathbb{E}_{v}[\mathbf{p}(v) \cdot d^{-4} \mathbf{p}(v)^{\top} \mathbf{H} \mathbf{p}(x')] \propto \mathbf{p}(x')$$
> > > 5. For $(i,j)=(2,0)$, $(4,0)$, $(0,4)$ and $(0,2)$, they introduce terms either independent of $v$ or $x'$, which leads to a **common bias** among the weights. (The bias of $(0,2)$ and $(2,0)$ might be even dominated by the magnitude of $(2,2)$ as discussed in case 3 ) Thus, we only need to **debias the weights** to remove these parts. This is also equivalent to  introducing a fixed bias to the inner layer of this network as pointed out by Reviewer Xpr2.
> > >
> > > Thus, our assumption on inner activation does not hurt the essence of feature learning and can be relaxed. Also, these analytic strategies can naturally apply to high-degree features (not restricted to quadratic ones). We've conducted **additional experiments** where we use ReLU as the inner activation for comparison and we also apply our methods to high-degree features. [(This link)](https://ibb.co/f4K5qCR) We find that our methods indeed applies for ReLU activation and high-degree features.

---

> ### Author Response · Authors · 2024-11-22
> **Response part 2**
>
> >There is no lower bound on the optimality of the dependence on d and κ in the bound. It should be possible to get d^2 potentially. What dependence on κ is optimal?
>
> Regarding the optimal dependence ofn $d$, we remark that $O(d^4)$ should be optimal when considering learning our targets with one gradient step. Please see our general response for a detailed discussion. For the dependence on $\kappa$, we mainly consider balanced features cases, i.e., $\kappa$ is small and treated as nearly a constant, which is required in our analytic framework since it leads the distribution of the joint distribution of the features to resemble Gaussian distribution (Lemma 1). This allows us to compute the learned weights out approximately. When $\kappa$ is large, the distribution of the features could be quite different and this case is beyond our scope of analysis.
>
> >Figure 1 is not very easy to understand. Might be helpful to add more explanation to the caption and label the different blocks/terms. It took me a while to parse it and I did not find it very helpful to understand the proof.
>
> We apologize for any confusion in the figure. We draw the figure for a visual interpretation of our proof idea presented below Proposition 1: using the **universality result** and the **second-order information of the link function**. For a better understanding of this graph, we will add a detailed explanation on how each block of terms are derived and what roles they are playing in the feature learning in our revised version. To better understand our proof idea, we recommend the readers to go over Appendix B.1, where we present an asymptotic analysis of the learned feature.
>
> >The irreducible error term of O(rp+2/d1/6)  is not discussed. This is much larger than the κ1/3d1/6  term in Niching et al. (2022) and requires r to be much smaller than d Is this necessary?
>
> A: We remark that **all the previous results studying nonlinear features have an irreducible term** because of the use of approximation results similar to central limit theorem on the nonlinear features to calculate the learned weights after GD.
> In general, the **dependence on $r$ is expected** and necessary when learning polynomials of $r$ features. It resembles the results of learning an arbitrary degree $p$ polynomial in $r$-dimensional space, which requires $\Omega(r^p)$ samples for efficient learning. Since $r$ represents the number of features, the case $r$ is much smaller than $d$ characterizes a **typical case of feature learning**, that is, the feature space is much smaller than the ambient space. if $r$ is not smaller or even bigger than $d$, the target function may be so complex that it may be only an arbitrary degree $2p$ polynomial, where feature learning is less meaningful. In this case, some generalized kernel methods like **quadratic NTK** [Chen, 2020] may be a more efficient analytic tool.
>
> We appreciate the suggestions on the presentation of our work. We will emphasize the generality of our work and revise the confusing sentences for better presentation. We would be grateful if the reviewer could reconsider the evaluation based on our contributions. If you have any further questions, please do not hesitate to comment.
>
>
> [Chen, 2020] Minshuo Chen, Yu Bai, Jason D. Lee, Tuo Zhao, Huan Wang, Caiming Xiong, Richard Socher Towards Understanding Hierarchical Learning: Benefits of Neural Representations

---

> ### Author Response · Authors · 2024-11-28
> **Response part 2**
>
> > The experimental results are still lacking, so it is hard for me to predict how they will differ from the proposed bounds.
>
> We have conducted new experiments to verify the generality of our analytic framework under different activations, targets and relaxations of assumptions. In [this link](https://ibb.co/d47bLQ4), we relax assumption 1 on how balanced the features are, i.e., the magnitude of $||A_i||$, as shown in the graph we relax the assumptions on features from fully balanced ones to extermely imbalanced ones that has spiked variance only in some directions. We see our reconstruction method in Proposition 1 actually applies to all the features, but the approximation accuracy seems to first decrease and then increase as the features get less balanced. We guess this is because the extremely imbalanced feature just reduces to multi-index model, which also has a clear sparse structure on the target.
>
> We have also relaxed Assumption 2 from **multiple quadratic features** to **multiple degree $q$ features** and  Assumption 4 from $\sigma_2=Q_q$ to $\sigma_2={\rm ReLU}$ . The results are shown in  [this link](https://ibb.co/f4K5qCR), where we find
> our method indeed applies for either ReLU activation or high-degree features.
>
> > Sample complexity dependence is sub-optimal. As the authors point out, this is perhaps unavoidable by their one-step gradient techniques, but that's not a satisfying response given that prior works have improved this in similar settings.
>
> We understand the reviewer's criticism about the suboptimality of $O(d^4)$, but learning **multiple nonlinear** features with $O(d^c)$ ($c$ is a constant  not dependent on $p$) was an open problem until our work. We believe our results offer **sufficient contributions** as the first work addressing such a broad and complex feature class. To reach $O(d^2)$ sample complexity, we may need to analyze multiple steps of GD (or SGD) to break this $O(d^4)$ gap, which is too hard to analyze.
>
> > To clarify on irreducible term ....
>
> We remark that the irreducible error term of $O(r^{2p}/d^{1/6})$ is a result of **error propagation** as we have discussed under proposition 2 because the error will be amplified when we are learning degree $p$ polynomials of features rather than learning the features themselves. The error of $O(r^{p/2}/d^{1/6})$ in proposition 1 is also a result of error propagation due to the complexity of extracting the features from a degree $p$ polynomial (which is very **non-smooth**, $O(r^{(p-1)/2})-$Lipschitz on average) of that feature. Actually, our result in Proposition 1 completely applies to any link function $g$ rather than polynomials.  Please see **Proposition 3 in Appendix C.1**, where we actually provide a general analysis on **$L$-Lipschitz link function $g$**. If the link function is smooth enough (e.g., $g$ is $O(1)-$Lipschitz) and has nondegenerate expected Hessian, the error can be reduced to $O(r^{2}/d^{1/6})$.
>
> We would be grateful if the reviewer could reconsider the evaluation based on our contributions, our discussion on the **relaxation of assumptions** and **our additional experiments** shown above. If you have any further questions, please do not hesitate to comment.

---

> > ### Author Response · Authors · 2024-12-02
> > **Thank you for your time and effort in reviewing our paper.**
> >
> > Dear Reviewer bUVE,
> >
> > Thank you for your valuable feedback on our paper. We believe our revisions have fully addressed your additional concerns by **relaxing the assumptions** and **doing additional experiments** as you ask. Please see the new response above.
> >
> > If you have any additional questions, we're available for discussion until tomorrow. Otherwise, we would greatly appreciate it if you could reconsider your score based on our revisions.
> >
> > Your insights have been instrumental in enhancing our paper. Thank you again for your thorough review.
> >
> > Best regards,
> >
> > The Authors

---

> > > ### Comment · Reviewer_bUVE · 2024-12-02
> > >
> > > Thanks for addressing several of my concerns, in particular, relaxing the assumption on the activation and adding in new experiments. I will update my score accordingly. I still feel that given the one-step analysis, it is still limited in new insights on feature learning beyond generalizing results to new more complex settings. Nevertheless, I'm happy to see this be accepted.

---

### Official Review · Reviewer_jyeu · 2024-11-10

**Soundness:** 4
**Presentation:** 4
**Contribution:** 2
**Rating:** 6
**Confidence:** 4

**Summary:**

The authors study layerwise training on a three layer neural network trained on a more general target function with a more involved "hierarchical feature" structure. They show sample complexity improvements over the classic kernel methods.

**Strengths:**

- Feature learning is one of the most fundamental problems in deep learning theory.

- The proof seem to be correct. Nice proof sketches are provided in the main text.

**Weaknesses:**

- I don't find the results of this paper particularly interesting. I would have easily guessed that these results should be obtainable for the case where r \neq 1, after looking at Nichani et al., 2023; Wang et al., 2023. See also my other comments below:

- The results of the paper are not tight; i.e., the sample complexity of O(d^4). Essentially, one would expect a sample complexity of $O(d^2)$, because there are $O(d^2)$ parameters to be learned in this setting. In particular, a more interesting study will consider $n \asymp d^2$ and provide a precise characterization of the limiting test error. In the single index case, this is similar to the benefit show by Ba et al. 2022 (and others) over the initial results of Damian et al 2022.

- In assumption 2, it is assumed that the Hessian has eigenvalues that are bounded below by a certain quantity. Can this be translated into a condition on the target function in terms of some quantity like information-exponent or leap-index? Right now, the assumption seems very artificial.

- What do the authors mean by " Moreover, when the entries of Ai are sampled i.i.d., the assumption is satisfied with high probability by Wigner’s semicircle law" in line 157?

- In the layer-wise training, the inner most layer is not trained and is kept at initialization; i.e., random features.

- Assumption 4 on the activation function is completely unrealistic and is only an artifact of the proof.

- What is the role of the update on the bias. Some prior work in the area consider neural networks without bias (e.g., the papers I list below). Can the authors comment on this please?

- The discussion of prior work on "Learning Linear Features" is very limited. There are many more results on this topic, studying feature in this case with detail. The authors should consider citing and discussing these papers (alongside any other paper on the topic that I might have missed here).

 [1] Dandi et al., The benefits of reusing batches for gradient descent in two-layer networks: Breaking the curse of information and leap exponents

 [2] Moniri et al., A Theory of Non-Linear Feature Learning with One Gradient Step in Two-Layer Neural Networks

 [3] Cui et al., Asymptotics of feature learning in two-layer networks after one gradient-step

 [4] Dandi et al., A Random Matrix Theory Perspective on the Spectrum of Learned Features and Asymptotic Generalization Capabilities

 [5] Wang et al, Nonlinear spiked covariance matrices and signal propagation in deep neural networks

**Questions:**

please see the weakness section.

---

> ### Author Response · Authors · 2024-11-22
> **Response to Reviewer jyeu**
>
> We thank the reviewer for the valuable comments and suggestions.
> > I don't find the results of this paper particularly interesting. I would have easily guessed that these results should be obtainable for the case where r \neq 1, after looking at (Nichani et al., 2023); (Wang et al., 2023). See also my other comments below:
>
> We understand the reviewer’s impression that our paper resembles (Nichani et al., 2023; Wang et al., 2023), but we remark that the **targets of interest**, **neural network parametrizations**, **intuitions** behind the results and **mathematical strategies differ significantly** from past works. Please see the **general response part 1** for a detailed discussion on this.
>
> >The results of the paper are not tight; i.e., the sample complexity of O(d^4). Essentially, one would expect a sample complexity of O(d2), because there are O(d2) parameters to be learned in this setting. In particular, a more interesting study will consider n≍d2 and provide a precise characterization of the limiting test error. In the single index case, this is similar to the benefit show by Ba et al. 2022 (and others) over the initial results of Damian et al 2022.
>
> A:We acknowledge that $\tilde{O}(d^4)$ is suboptimal under the information theoretical pointview, but it is near optimal under our algorithm framework which only uses one gradient step for feature learning. Please see the **general response part 2** for the detailed discussion of the sample complexity. Moreover, as the reviewer proposed, one can achieve tighter results like (Ba et al. 2022) in the single-index case, but our model considers multiple and nonlinear features, which are much more complex to analyze. We make the first step on analyzing this “generalized multi-index” model and provide a general analytic framework for learning multiple nonlinear features,
>
> >In assumption 2, it is assumed that the Hessian has eigenvalues that are bounded below by a certain quantity. Can this be translated into a condition on the target function in terms of some quantity like information-exponent or leap-index? Right now, the assumption seems very artificial.
>
> We remark that the **information-exponent is 4**. We make Assumptions 2-3 to ensure a small $\mathcal{P}_{0}(f^{\star})$ and $\mathcal{P}_2(f^{\star})$ and a non-degenerate $\mathcal{P}_4(f^{\star})$. These assumptions ensure the target function adequately depends on all  r features, and prevent degenerate cases, which are also considered in previous works, such as Assumption 2 in [Damian et al 2022] (where the information-exponent is 2 because they consider linear features). For the same reason, the leap-index is **also basically 4** in our setting (jumping from degree 0 to degree  4).
>
> >What do the authors mean by " Moreover, when the entries of Ai are sampled i.i.d., the assumption is satisfied with high probability by Wigner’s semicircle law" in line 157?
>
>  Here we want to give an example of $A_i$ that satisfies the assumption. For example, when the entries of $A_i$ follow i.i.d distributions with zero mean, right scaling of variance, and a bounded fourth moment,  such as $[A_i]_{j,k}\sim N(0,1/d^2)$, then we have $||A_i||_F \approx 1$ and $||A_i||_2 \lesssim \frac{1}{\sqrt{d}}$ by standard random matrix theory.
>
> >In the layer-wise training, the inner most layer is not trained and is kept at initialization; i.e., random features.
>
> The random feature embedding is commonly assumed in previous works (such as [Nichani et al., 2023]; [Wang et al., 2023], [Safran. et. al]) for multi-layer neural networks, which simplifies the analysis for the training process.
> >Assumption 4 on the activation function is completely unrealistic and is only an artifact of the proof.
>
> Assumption 4 on the activation function does not hurt the essence of feature learning, as long as the inner activation emphasizes the crucial quadratic component $Q_2$ in capturing quadratic features. Please see our **general response** for detailed discussions.

---

> > ### Author Response · Authors · 2024-11-22
> > **Response to Reviewer jyeu part 2**
> >
> > >What is the role of the update on the bias. Some prior work in the area consider neural networks without bias (e.g., the papers I list below). Can the authors comment on this please?
> >
> > The bias term is **necessary for expressing arbitrary polynomials** of the features. It along with the trainable weights $a$ allows for the outer layer to construct any polynomial approximator. If there is no bias term, we cannot derive a universal approximation of the polynomial features. The prior works which don't have bias term only focus on recovering the subspace and not generalization.
> > >The discussion of prior work on "Learning Linear Features" is very limited. There are many more results on this topic, studying feature in this case with detail. The authors should consider citing and discussing these papers (alongside any other paper on the topic that I might have missed here).
> >
> > We thank the reviewer for providing the literatures and they are indeed highly related to our work. We will cite these works and add a detailed discussion on them. However, they lie in the scope of learning linear features, while our work considers learning non-linear ones. Our work may not be optimal when we compare our algorithms and analysis to the most recent ones for multi-index models, but it is the first step to investigate multiple nonlinear features.
> >
> > We appreciate the suggestions on the presentation of our work. We will emphasize the generality of our assumptions and methods and add more discussions to compare our work with previous works. We would be grateful if the reviewer could reconsider the evaluation based on our contributions. If you have any further questions, please do not hesitate to comment.
> >
> > [Safran. et. al] Itay Safran, Jason D. Lee, Optimization-Based Separations for Neural Networks

---

> ### Comment · Reviewer_jyeu · 2024-11-26
>
> I thank the authors for their thorough response. While I agree (in general) with the *general response part 1*, I still think:
>
> 1. the setting being studied here is not novel enough.
> 2. there is room for improvement of the sample complexity.
> 3. the assumptions of the paper are unrealistic.
>
> I will be willing to recommend the acceptance of the paper at a future time if: (1) the assumptions I discussed in my review are relaxed, *or* (2) the gradient update is tweaked to give a better sample complexity.
>
> I will maintain my score for now.

---

> ### Author Response · Authors · 2024-11-28
> **Response**
>
> We thank the reviewer for the response.
>
> >the setting being studied here is not novel enough.
>
> We emphasize again that we consider the setting of learning **multiple nonlinear** features, which is **not considered in any previous works** (previous works either consider multiple linear feautures or single nonlinear feature). This setting also includes **much broader function classes** and is more interesting (and also harder) to consider. Although our algorithm framework resembles previous works, our **targets of interest**, **neural network parametrizations**, **intuitions** behind the results and **mathematical strategies**  differ significantly, as we have discussed in the general response.
>
> >there is room for improvement of the sample complexity.
>
>
> It is our targets of interests and algorithm framework that determine the $O(d^4)$ sample complexity. As discussed in our general response, the information exponent and leap index is basically $4$, which means that $O(d^4)$ might be neccessary for efficient recovery of the feature subspace under our algorithm framework. We understand the reviewer's criticism about the suboptimality of $O(d^4)$, but learning **multiple nonlinear** features with $O(d^c)$ ($c$ is a constant  not dependent on $p$) was an open problem until our work. We believe our results offer **sufficient contributions** as the first work addressing such a broad and complex feature class. To reach $O(d^2)$ sample complexity, we may need to analyze multiple steps of GD (or SGD) to break this $O(d^4)$ gap, which is too hard to analyze.
>
> >The assumptions of the paper are unrealistic.
>
> We have argued in the general response that Assumption 4 mentioned by the reviewer to be not realistic **can actually be relaxed**. We make assumption 4 just for a cleaner and centered learned weights to simplify our analysis, and the fundamental intuition is that the **quadratic component  $Q_2$ in the activation function plays the dominant role**. We give an informal proof intuition (summarized from Appendix C.1 ). The first gradient step turn the weights into the form
> $$ h^{(1)}(x',v) \approx \sum_{i,j \ge 2}\frac{c_i^2c_j}{d^{i}} \mathbb{E}_x[f^{\star}(x) Q_i(x\cdot x') Q_j(x \cdot v) ],$$
> where  the component $i=j=2$ will recover the quadratic features by our universality theory:
> $$\mathbb{E}_x[f^{\star}(x) Q_2(x\cdot x') Q_2(x \cdot v) ] \approx d^{-4} \mathbf{p}(v)^{\top} \mathbf{H} \mathbf{p}(x'),$$
> where $\mathbf{p} \in \mathbb{R}^{r}$ is the $r$ quadratic features and $ \mathbf{H}$ is the expected Hessian.
>
> (This is also why we neeed $d^4$ samples, because we are extracting the degree $2+2=4$ component from the targets using the empirical gradients). We  have proven in the Appendix C.2 and C.3 that the $(i,j)=(2,2)$ term is the **dominant one** under our assumptions on the targerts.
> Let's consider the case we relax the assumption on the activation to add back the constant and linear term ($i\le 1$ or $j \le 1$ ) .
> 1. The magnitude of all the terms with $i+j >4$ diminishes to $o_d(1)$ compared with the term of $(i,j)=(2,2)$
> 2. For the case $i+j$ is odd, the expectation is zero because $f$ has only even-degree components under our assumptions.
> 3. The term of $(i,j)=(0,0)$ is also $o_d(1)$ compared with the term of $(i,j)=(2,2)$ because we have asssumed (in Assumption 2 and 3) the information exponent and leap index to be basically $4$.
> 4. The terms of $(i,j)=(1,3)$, $(3,1)$  have the same magnitude of $(i,j)=(2,2)$, but after we apply the linear product $B^{\star}$ in Proposition, these two terms will cancel by the orthongonality of degree-$j$ term of $ Q_j(x \cdot v) $ and the degree-2 terms in $B^{\star}$. This is because we simply set each row of $B^{\star}$ to be proportional $ \mathbf{H}^{-1}\mathbf{p}(v)$ which only has degree $2$ terms. Thus, we still have $$B^{\star} h^{(1)} (x')\approx  \mathbf{H}^{-1}  \mathbb{E}_{v}[\mathbf{p}(v) \cdot d^{-4} \mathbf{p}(v)^{\top} \mathbf{H} \mathbf{p}(x')] \propto \mathbf{p}(x')$$
> 5. For $(i,j)=(2,0)$, $(4,0)$, $(0,4)$ and $(0,2)$, they introduce terms either independent of $v$ or $x'$, which leads to a **common bias** among the weights. (The bias of $(0,2)$ and $(2,0)$ might be even dominated by the magnitude of $(2,2)$ as discussed in case 3 ) Thus, we only need to **debias the weights** to remove these parts. This is also equivalent to  introducing a fixed bias to the inner layer of this network as pointed out by Reviewer Xpr2.
>
> All in all, as shown above, our assumption on inner activation does not hurt the essence of feature learning and can be relaxed. Also, these analytic strategies can naturally apply to high-degree features (not restricted to quadratic ones). We've conducted **additional experiments** where we use ReLU as the inner activation for comparison and we also apply our methods to high-degree features. [(This link)](https://ibb.co/f4K5qCR) We find that our methods indeed apply for ReLU activation and high-degree features.

---

> > ### Author Response · Authors · 2024-11-28
> >
> > We would be grateful if the reviewer could reconsider the evaluation based on our contributions and our discussion on the relaxation of assumptions above. If you have any further questions, please do not hesitate to comment.

---

> > > ### Author Response · Authors · 2024-12-02
> > > **Thank you for your time and effort in reviewing our paper.**
> > >
> > > Dear Reviewer jyeu,
> > >
> > > Thank you for your valuable feedback on our paper. We believe our revisions have fully addressed your additional concerns by **relaxing the assumptions** as you ask. Please see the new response above.
> > >
> > > If you have any additional questions, we're available for discussion until tomorrow. Otherwise, we would greatly appreciate it if you could reconsider your score based on our revisions.
> > >
> > > Your insights have been instrumental in enhancing our paper. Thank you again for your thorough review.
> > >
> > > Best regards,
> > >
> > > The Authors

---

> > > > ### Comment · Reviewer_jyeu · 2024-12-02
> > > >
> > > > Thanks for providing the detailed response. After reading the paper one more time, the other reviews, and also the response provided by the authors, I decided to increase my score to 6 and lean towards acceptance (because of the arguments provided by the authors during the discussion period regarding the activation function assumption).

---

### Author Response · Authors · 2024-11-22
**General Response to All Reviewers Part 1**

We thank all reviewers for the detailed comments and feedback. We’d first like to address comments that were shared by a few of the reviewers.

**1.Comparison with  (Nichani et al., 2023) & (Wang et al., 2023)**

While our work seems similar to these two previous works in the algorithm framework, we point our that the **targets of interests**, the **parametrizations of neural networks**, **intuitions behind the results** and the corresponding **mathematical strategies** are of great difference compared to the past works, not an extension to either (Nichani et al., 2023) or (Wang et al., 2023). The theory of (Nichani et al., 2023) or (Wang et al., 2023) cannot be generalized to our work with an “easy guess”.

**a) Target of interests**: We consider learning hierarchical polynomials
 of multiple quadratic features, while  (Nichani et al., 2023) considers polynomials of single quadratic feature and   (Wang et al., 2023) consider broader feature class but still limited to a  single feature. Our function class contains  **multiple** non-linear features, which have not been considered in any previous work. This generalized multi-index target is much more interesting and also **inherently harder** to analyze compared with these two works. Moreover, our assumptions on the target functions, particularly the link function $g$ are quite different and even contrary to the previous two works. While  (Nichani et al., 2023) & (Wang et al., 2023) assumes the link function $g$ has non-zero linear component, we rather assume in Assumption 2 & 3 $g$ has a **non-degenerate expected Hessian** and a nearly **zero linear component**. This is the first **essential difference between single-feature and multi-feature** learning, because  our assumptions ensure that the network genuinely learns to represent and distinguish all $r$ features, while assumptions proposed by  (Nichani et al., 2023) & (Wang et al., 2023)  represent a degenerate case that neural network may only learn the dominant linear combinations of the  $r$ features. We’ve also provided examples and counterexamples in Remark 1 to elaborate on this difference.


**b) Parametrizations of neural networks**: Different from (Nichani et al., 2023) and (Wang et al., 2023), both which initialize the second-layer weights (i.e., $\mathbf{W}$ in our paper) at zeros, we initialize $\mathbf{W}$ as **independent Gaussians**. This **random initialization** instead of a deterministic one is another essential difference between multiple feature learning and single feature learning, because we require the weights to **span the full space**, which enables the learned weights (after one step of GD) to **capture the multiple features in all directions** instead of converging to a specific direction like (Nichani et al., 2023) and (Wang et al., 2023), i.e., single feature learning.

**c) Intuition and mathematical strategies**: The intuition of (Nichani et al., 2023) and (Wang et al., 2023) is that the gradient descent on the zeroed weights $W$ of  (either one step or multiple steps) make them to concentrate to the single target feature (scaled by a constant). On the contrary under our assumptions on the targets and proper initialization on the weights of neural network，our intuition is that with sufficient number of training samples, gradient step on the randomly initialized weights $W$, i.e., Stage 1,  makes them **fully recover and span the $r-$dimensional space** of all the non-linear features, as discussed in Proposition 1 and Figure 1. **Subspace recovery is completely different from and also significantly harder than single feature recovery**.


To prove this statement, we propose a **novel universality theory** (Lemma 1) that relates the joint distribution of multiple nonlinear features to a multivariate Gaussian, which enables us to approximately compute the learned weights. This technique of multivariate Gaussian approximation is not considered in any previous work. More specifically, to handle learning multiple features, the proof of Proposition 1 is different and also more complex than similar results in (Nichani et al., 2023) and (Wang et al., 2023) for learning a single-feature, because we are focusing on the calculation of second-order terms of the link function, i.e. $\mathbb{E}]\nabla^2 g(p)]$, while their works mainly consider the first-order term  $\mathbb{E}]\nabla g(p)]$. Our computation requires more techniques in high dimensional probability and random matrix theory.

---

> ### Author Response · Authors · 2024-11-22
> **General Response Part 2**
>
> **2. Discussion on the sample complexity of  $\tilde{O}(d^4)$**
>
> We remark that $\tilde{O}(d^4)$ is a **meaningful** and also **reasonable** sample complexity bound in our algorithm framework. Firstly, any **polynomial sample complexity bound** (i.e., ${O}(d^{c})$) is a significant improvement upon the bounds of $\Omega(d^{2p})$ achieved by kernel methods when learning a target of multiple quadratic features. Secondly, our $\tilde{O}(d^4)$ sample complexity is **nearly optimal** when considering feature learning with **one gradient step**. As discussed in our work, we utilize the  second-order information of the link function $g$, which is dominated by the degree 4 component of the entire function, i.e. $\mathcal{P}_4(f)$. Under our assumptions on the target the “information exponent” is approximately 4 and for an efficient concentration of $\mathcal{P}_4(f)$, [Dandi et a, 2022] shows that one needs $\Omega(d^4)$ samples. This substantiates the near optimality of our results. Regarding the intuitive bound of ${O}(d^2)$ proposed by the reviewers, we think this information optimal result could be achieved when we consider different algorithms that utilize the samples more thoroughly and analyze the model asymptotically such as in the regime of $n\propto d^2$. However, this would require analyzing multiple steps of GD (or other methods) on the weights, which is quite challenging and could be a great future extension of our work.
>
>
>
>
> **3. Discussion on the proposed assumptions**
>
> We understand the reviewers’ concern that the four assumptions may oversimplify the model and restrict the scope of this paper, but we remark that these assumptions are all for technical simplicity, while still demonstrating the key  essence of feature learning. To be specific, Assumption 1 of orthonormal quadratic features can be attained via linear transformation on the features. The assumption on the operator norm of the feature matrix represents a **typical case of features**, where the feature has **balanced emphasis** on each direction of the ambient space. The assumption on the expected Hessian in Assumption 2 and the assumption of a **diminished** $\mathcal{P}_2{f}$ are **necessary for distinguishing all the $r$ features**, which is also similarly assumed in previous works of multi-index models [Damian,2022]. A degenerate expected Hessian may fail to span all the directions spanned by the features, and a large $\mathcal{P}_2{f}$, that is, a dominant linear part of the quadratic features in the target,  prevents the first step of GD on the weights from detecting the Hessian because the learned weights will concentrate to the direction of that linear part.
>
>
> For the assumption on the inner activation function, we remark that the most important component is its quadratic component $Q_2$, which plays the role of capturing the quadratic features, while other components all lead to noises or biases in the learned weights. Thus, we make assumption 4 just for a cleaner and centered learned weights to simplify our analysis, the fundamental intuition is that the **quadratic component plays the dominant role**. Through a more complicated argument, for any inner activation function with a nondegenerate quadratic component, we can also recover the features through one gradient step on the weights.
>
> We thank all the reviewers again for the suggestions on our work. We will add a detailed discussion on these three issues in our revised version, which will release in one or two days.
>
> [Dandi et. al, 2022] Yatin Dandi, Florent Krzakala, Bruno Loureiro, Luca Pesce, Ludovic Stephan, "How Two-Layer Neural Networks Learn, One (Giant) Step at a Time"
>
> [Damian,2022] Alexandru Damian, Jason Lee, Mahdi Soltanolkotabi, “Neural networks can learn representations with gradient descent”

---

### Author Response · Authors · 2024-11-25
**Rebuttal Revision of our paper comes out**

Dear reviewers,

Thank you again to all the reviewers for your detailed feedback on our submission. We have uploaded the revised version of our paper, in which we have added detailed **comparison with previous works** (Nichani et al., 2023) & (Wang et al., 2023), **more recent related works** and **explanations on our proposed assumptions**. Due to the page limit, we have moved the original Section 5 (Numerical Experiments) to Appendix A. All the modifications in the main page are **marked in red** for the reviewers’ convenience.

Given that the discussion period ends soon, we would like to know whether our rebuttal has addressed your concerns, and whether any more questions of yours remain. Please do let us know, we are happy to answer any additional questions.

Best,

The Authors

---

### Meta-Review · Area_Chair_fiMJ · 2024-12-21

**Metareview:**

This paper considers the problem of learning of hierarchical polynomials of multiple nonlinear features using three-layer neural networks. Authors consider a layer-wise training procedure and prove recovery of the space spanned by the nonlinear features, and learning the target non-linearity on top. The overall sample complexity of the procedure is d^4 with polynomial-time iteration.


This paper was reviewed by five expert reviewers with the following Scores/Confidence: 8/4, 6/4, 6/4, 8/4, 6/2. I think the paper is studying an interesting topic and the results are relevant to ICLR community. The following concerns were brought up by the reviewers:

- The algorithm in this paper deviates from deep learning practice in many parts. Although the methodology is clearly explained, these should be emphasized in the introduction and throughout the main text.


- While I agree with the authors that d^4 sample complexity is reasonable, I suggest they include discussions on any room for improvements. This is a point that was raised by multiple reviewers and reader could also benefit.

- several typos and ambiguous statements were pointed by the reviewers. These should be carefully addressed.


Authors should carefully go over reviewers' suggestions and address any remaining concerns in their final revision. Based on the reviewers' suggestion, as well as my own assessment of the paper, I recommend including this paper as 'spotlight' to the ICLR 2025 program.

**Additional Comments On Reviewer Discussion:**

Reviewer questions are thoroughly answered by the authors. The revision provides color-coded updates.

---

### Decision · Program_Chairs · 2025-01-22

Accept (Poster)